# Deciphering the spatial landscape and plasticity of immunosuppressive fibroblasts in breast cancer

Hugo Croizer[1,2,7], Rana Mhaidly[1,2,7], Yann Kieffer [1,2,7], Geraldine Gentric [1,2], Lounes Djerroudi[1,2,3], Renaud Leclere [3], Floriane Pelon[1,2], Catherine Robley[1,2], Mylene Bohec [4,5], Arnaud Meng[1,2], Didier Meseure [3], Emanuela Romano [6], Sylvain Baulande [4,5], Agathe Peltier [1,2], Anne Vincent-Salomon [3] & Fatima Mechta-Grigoriou [1,2] ✉

Although heterogeneity of FAP+ Cancer-Associated Fibroblasts (CAF) has been described in breast cancer, their plasticity and spatial distribution remain poorly understood. Here, we analyze trajectory inference, deconvolute spatial transcriptomics at single-cell level and perform functional assays to generate a high-resolution integrated map of breast cancer (BC), with a focus on inflammatory and myofibroblastic (iCAF/myCAF) FAP+ CAF clusters. We identify 10 spatially-organized FAP+ CAF-related cellular niches, called EcoCellTypes, which are differentially localized within tumors. Consistent with their spatial organization, cancer cells drive the transition of detoxification-associated iCAF (Detox-iCAF) towards immunosuppressive extracellular matrix (ECM)-producing myCAF (ECM-myCAF) via a DPP4- and YAP-dependent mechanism. In turn, ECM-myCAF polarize TREM2+ macrophages, regulatory NK and T cells to induce immunosuppressive EcoCellTypes, while Detox-iCAF are associated with FOLR2+ macrophages in an immuno-protective EcoCellType. FAP+ CAF subpopulations accumulate differently according to the invasive BC status and predict invasive recurrence of ductal carcinoma in situ (DCIS), which could help in identifying low-risk DCIS patients eligible for therapeutic de-escalation.

Breast cancer (BC) is one of the most frequent cancers in women and a major cause of death in western countries, despite recent improvements in its earlier detection and the development of effective therapies. BC is a heterogeneous disease classified by histological analysis into three main subtypes exhibiting distinct prognoses: luminal (Lum), HER2, and triple-negative (TN). It is important to note that there is still no biomarker to predict BC progression from ductal carcinoma in situ (DCIS) to invasive BC (IBC). It is now well established that the tumor micro-environment (TME) plays a key role in tumor growth and progression. The TME is composed of cancer-associated fibroblasts (CAF), infiltrating immune cells, endothelial cells and pericytes embedded in extracellular matrix (ECM), involved in numerous steps of tumor growth and metastatic spread. CAF constitute one of the most abundant TME components in solid tumors. While underestimated for a long time, CAF heterogeneity is now well-recognized. Indeed, several CAF populations have been recently uncovered in human BC by

[1]Institut Curie, Stress and Cancer Laboratory, Equipe Labélisée par la Ligue Nationale Contre le Cancer, PSL Research University, 26, Rue d'Ulm, F-75248 Paris, France. [2]Inserm, U830, 26, Rue d'Ulm, F-75005 Paris, France. [3]Department of Diagnostic and Theragnostic Medicine, Institut Curie Hospital Group, 26, Rue d'Ulm, F-75248 Paris, France. [4]Institut Curie, PSL Research University, ICGex Next-Generation Sequencing Platform, 75005 Paris, France. [5]Institut Curie, PSL Research University, Single Cell Initiative, 75005 Paris, France. [6]Department of Medical Oncology, Center for Cancer Immunotherapy, Institut Curie, 26, Rue d'Ulm, F-75248 Paris, France. [7]These authors contributed equally: Hugo Croizer, Rana Mhaidly, Yann Kieffer. ✉e-mail: fatima.mechta-grigoriou@curie.fr

combining the study of multiple CAF markers including Fibroblast Activation Protein (FAP), Smooth Muscle-α Actin (SMA) and Integrin β1 (CD29), among others[1–16]. We previously identified four CAF populations, referred to as CAF-S1 to CAF-S4 in BC[17–19]. The myofibroblastic CAF-S1 population (FAP^Pos CD29^Med SMA^Med-High) and the perivascular-like CAF-S4 (FAP^Neg CD29^High SMA^High, also highlighted by others as cancer-associated perivascular-like or CAP)[20] are detected in tumors and enriched in TN BC. The existence of these different CAF and CAP populations was confirmed by numerous methods, including single cell analysis, and demonstrated in other cancer types and in various species[1–16].

Myofibroblastic CAF populations are pro-metastatic, and their content has been associated with BC progression[9,18,19,21–28]. In particular, the myofibroblastic CAF-S1 and the perivascular-like CAP (CAF-S4) populations enhance tumor invasion through complementary mechanisms by acting on tumor cells and the surrounding ECM, respectively[18,19]. In addition, FAP+ CAF (or CAF-S1) fibroblasts have also been associated with an immunosuppressive environment in various tumor types[3,4,10,17,29–35]. Indeed, FAP+ CAF promote immunosuppression through multiple mechanisms: they attract CD4+ CD25+ T lymphocytes, enhance their survival and promote their differentiation into FOXP3+ regulatory T lymphocytes (Tregs), while they simultaneously inhibit CD8+ T cell cytotoxicity[4,10,17,35–39]. Moreover, FAP+ CAF have been shown to contribute to immunotherapy resistance in mouse and human cancers[3,6,8,10,40–43]. As FAP+ CAF exhibited both pro-metastatic and immunosuppressive activities, we previously hypothesized that this population could itself be heterogenous and we generated one of the most resolutive single-cell RNA sequencing data (scRNA-seq) of the FAP+ CAF population in BC[10]. We found that the FAP+ CAF population is composed of 8 cellular clusters, including three inflammatory (iCAF) and five myofibroblastic (myCAF) clusters[10]; iCAF and myCAF having been previously identified in pancreatic ductal adenocarcinoma (PDAC)[1,5,6,44]. Based on differentially expressed genes as previously described in detail[10], the three iCAF clusters are characterized by detoxification pathway (Detox-iCAF), interleukin-signaling pathway (IL-iCAF) and IFNγ-mediated response (IFNγ-iCAF). IFNγ-iCAF express the CD74 antigen and are potentially reminiscent of the antigen-presenting CAF (ap-CAF) identified in PDAC[6,45]. The five myCAF clusters are characterized by a high expression of genes coding ECM proteins (ECM-myCAF), TGFβ signaling pathway (TGFβ-myCAF), wound healing (Wound-myCAF), IFNαβ-mediated response (IFNαβ-myCAF) and acto-myosin pathway (acto-myCAF)[10]. ECM-myCAF and TGFβ-myCAF clusters accumulate in BC enriched in PD-1+, CTLA-4+, and TIGIT+ CD4+ T lymphocytes, themselves enriched in Tregs. Importantly, ECM-myCAF, TGFβ-myCAF and Wound-myCAF clusters are associated with primary resistance to immunotherapies in melanoma and non-small cell lung cancer patients[10].

Although the role of FAP+ CAF clusters in metastatic spread, immunosuppression and resistance to immunotherapies is now well-established, their spatial localization, plasticity and reciprocal crosstalk with surrounding cells remain unanswered questions. Efforts to understand the cellular organization in BC has revealed co-occurring cell types in bulk-RNAseq dataset[13], yet these studies lacked spatial context. Moreover, multiplex imaging techniques remain limited by the number of stained proteins and cannot identify diverse cell states. Spatial transcriptomics recently provided a new means to fulfill this lack[46]. As first examples, several recent studies demonstrated that immune cells are not randomly distributed in tumors but organized into niches, which facilitate their functions and can predict response to immunotherapies and patient prognosis[44,47–52].

Here, we combine single-cell trajectory inference, deconvolution of spatial transcriptomics data and functional assays using primary FAP+ CAF isolated from BC patients to uncover the plasticity and spatial organization of these fibroblasts with other cell types, thus addressing the limitations of previous studies. We unravel FAP+ CAF plasticity and crosstalk with both cancer and immune cells, and we identify 10 spatially-organized FAP+ CAF cluster-related cellular modules referred to as EcoCellTypes (ECT), which are composed of specific FAP+ CAF clusters and precisely localized within tumors. Immuno-suppressive and immuno-permissive ECT comprise specific FAP+ CAF clusters and immune populations located at various distances from tumor aggregates and blood vessels. Consistent with the spatial organization of these FAP+ CAF cluster-related ECT, we observe that cancer cells promote the differentiation of the Detox-iCAF cluster into Wound-myCAF and ECM-myCAF clusters through DPP4- and YAP1-dependent mechanisms both in vitro and in vivo. Furthermore, our study reveals that FAP+ CAF clusters play a key role in the organization of ECT. Indeed, Detox-iCAF, IL-iCAF and IFNγ-iCAF recruit monocytes and induce a FOLR2+ tumor-associated macrophage (TAM) phenotype, while ECM-myCAF and TGFβ-myCAF promote TREM2+ TAM and NKG2A+ regulatory NK phenotypes. As CAF have recently been associated with invasion in Ductal Carcinoma in Situ (DCIS)[53], we go a step further by analyzing the role of these different FAP+ CAF clusters in BC progression. By using in-house and public cohorts of BC patients with DCIS, we show that the FAP+ Detox-iCAF cluster content significantly decreases from DCIS to IBC, consistent with their transition into ECM-myCAF. Importantly, we also observe that low Detox-iCAF and high TGFβ-myCAF content in DCIS at diagnosis predicts the recurrence of DCIS into IBC, independently of nuclear grade or molecular subtype, thereby revealing a prognostic factor for recurrence based on specific FAP+ CAF clusters. These findings will help in identifying DCIS patients with low-risk of progression, who might benefit from therapeutic de-escalation.

## Results

### FAP+ CAF plasticity revealed by in silico analysis and functional assays

We previously identified 8 clusters of FAP+ CAF (CAF-S1) in breast cancer (BC) by scRNA-seq[10] (Supplementary Fig. 1A), but the origin of this FAP+ CAF heterogeneity in BC is poorly understood. We hypothesized that one particular FAP+ CAF cluster might be the reservoir of the others. In line with this hypothesis, we found that part of the Detox-iCAF cluster retained *PI16* (*Peptidase Inhibitor 16*) expression (Supplementary Fig. 1B) which has recently been identified as a marker of universal fibroblasts[42,54]. We validated these observations using an independent public scRNA-seq datasets from BC and healthy mammoplasties[55,56]. Label transfer[57] enabled us to confirm the existence of the different FAP+ CAF clusters in BC, as well as 2 fibroblast clusters from healthy mammoplasties (Fig. 1A). Interestingly, the Detox-iCAF cluster formed a transcriptional continuum with *PI16*+ universal fibroblasts (Fig. 1B), indicating that the FAP+ Detox-iCAF cluster might be the reservoir of the other FAP+ CAF clusters. As Detox-iCAF showed transcriptional similarities with universal fibroblasts, we sought to confirm that the Detox-iCAF cluster was transcriptionally different from normal fibroblasts. Differential analysis between fibroblasts from healthy mammary tissue and Detox-iCAF confirmed that Detox-iCAF showed a large number of up-regulated genes compared to normal fibroblasts, such as FAP (Fig. 1C), consistent with the CAF-S1 isolation method based on FAP marker[10,17–19]. To account for sample size inflation in scRNA-seq data, we confirmed this result at sample level after pseudo-bulk reconstruction (Supplementary Fig. 1C, see also "Methods", #"Differential analysis between fibroblasts from healthy tissue and Detox-iCAF"). To investigate how FAP+ CAF cluster diversity emerged, we applied several trajectory inference methods on the FAP+ CAF-enriched scRNA-seq dataset isolated from BC, setting the Detox-iCAF cluster as the root of the trajectories. PAGA tree[58] revealed transitions from the Detox-iCAF to IL-iCAF and IFNγ-iCAF clusters and a direct trajectory from the Detox-iCAF to the ECM-myCAF cluster, which in turn gave rise to the TGFβ-myCAF cluster (Fig. 1D). The trajectory inference method STREAM[59] uncovered similar

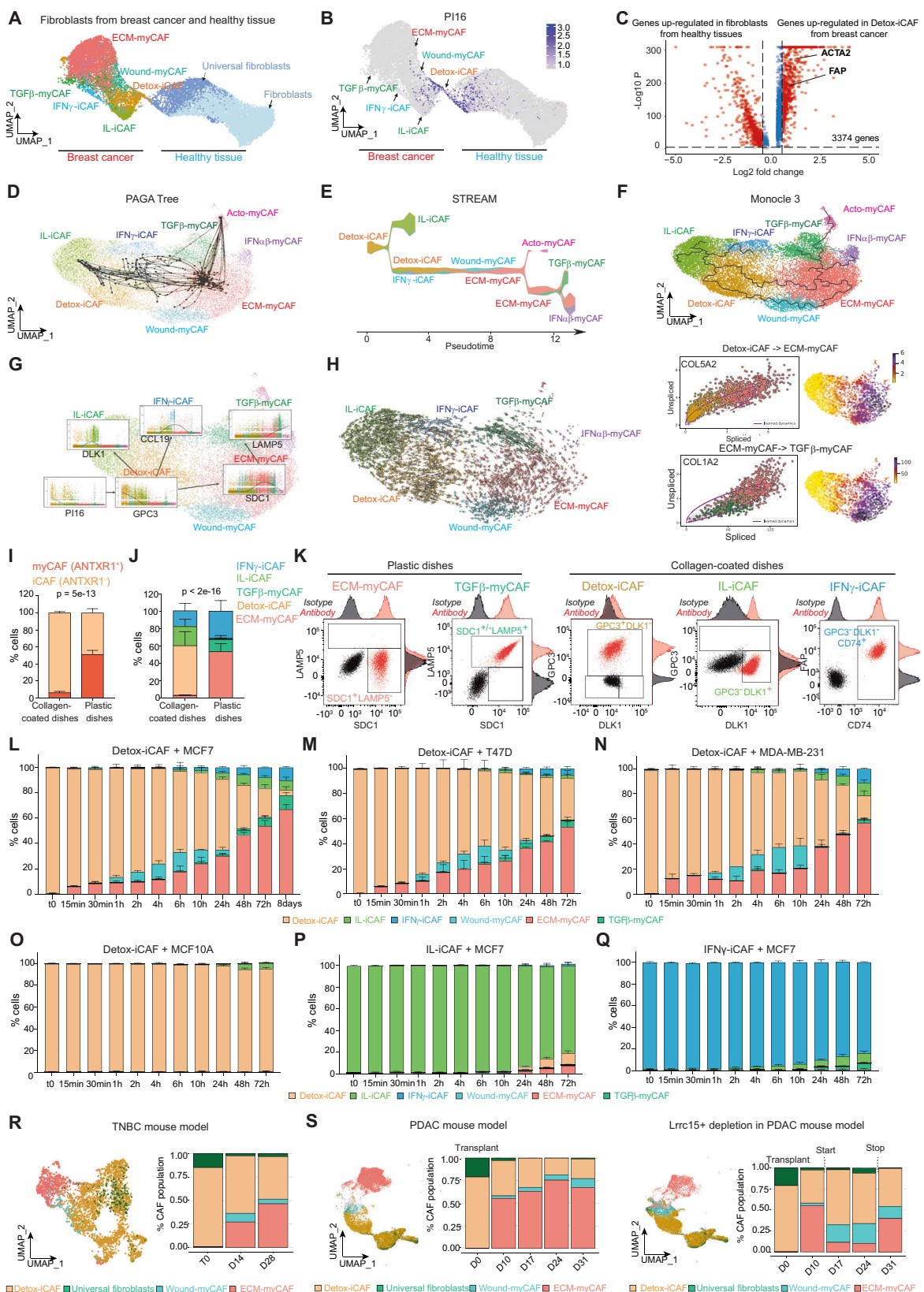

trajectories between the different FAP+ CAF clusters (Fig. 1E). In addition, STREAM detected an indirect transition between Detox-iCAF and ECM-myCAF through the Wound-myCAF cluster, as well as a trajectory from ECM-myCAF to IFNαβ-myCAF (Fig. 1E). Monocle3[60,61] recapitulated PAGA tree and STREAM by detecting both the direct trajectory between Detox-iCAF and ECM-myCAF clusters, and the

indirect transition through the Wound-myCAF cluster (Fig. 1F). From Monocle3 trajectories, we computed the pseudotime by rooting the trajectory in the Detox-iCAF (Supplementary Fig. 1D). This revealed an early loss of *PI16* expression, quickly followed by a progressive loss of *GPC3* expression in Detox-iCAF and the gradual upregulation of *DLK1* in IL-iCAF and *CCL19* in IFNγ-iCAF (Fig. 1G), GPC3, DLK1 and CCL19

**Fig. 1 | In silico analysis of FAP+ CAF plasticity in human breast cancer. A** UMAP combining FAP+ CAF from BC[55] (Left) and fibroblasts from healthy breast tissues[56] (Right, *n* = 15,667 cells), colored by cluster identity. **B** *PI16* gene expression. **C** Differential gene expression between Detox-iCAF and normal fibroblasts. *p* values from two-sided Wilcoxon rank sum test. In red, genes with adjusted *p* value < 0.05. Trajectory inferences on BC FAP+ CAF scRNA-seq dataset from ref. 10 inferred by PAGA tree (10 random downsampling) (**D**), STREAM trajectory showing cluster proportion along pseudotime (**E**) and Monocle 3 (**F**). **G** Expression of FAP+ CAF cluster markers according to Monocle 3 pseudotime and colored according to cluster identity. **H** Left, velocities from scVelo on the FAP+ CAF UMAP. Right, spliced/unspliced phase portrait and expression on UMAP for *COL5A2* (Top) and *COL1A2* (Bottom) genes. **I** Percentages of iCAF (ANTXR1−) and myCAF (ANTXR1+) clusters among FAP+ CAF cultured on collagen-coated or plastic dishes (*n* = 6 independent experiments). *p* value from two-sided Fisher's Exact test. **J** Same as (**I**) for FAP+ CAF cluster identity using specific markers by flow cytometry (*n* = 6).

**K** Flow cytometry plots showing FAP+ CAF cluster-specific surface markers in sorted primary FAP+ CAF. **L** Percentage of each FAP+ CAF cluster among FAP+ CAF (flow cytometry data) after co-culture of Detox-iCAF with MCF7. Timepoints indicate the duration of co-culture (*n* = 3 independent experiments). **M−O** Same as (**L**) for co-culture of Detox-iCAF with T47D (**M**), MDA-MB-231 (**N**) and MCF10A (**O**). **P, Q** Same as (**L**) for co-culture of IL-iCAF (**P**) or IFNγ-iCAF (**Q**) with MCF7. All data are mean ± SEM. **R** Left, UMAP of fibroblasts from scRNA-seq data following TN BC cell injection from ref. 9 colored by cell identity (*n* = 3363 fibroblasts). Right, quantification at 0, 14 and 28 days after tumor implantation. **S** Same as (**R**) for fibroblasts from scRNA-seq data following subcutaneous injection of PDAC cell line in WT mice (*n* = 23,675 fibroblasts) (Left) and in *Lrrc15*-diphteria toxin receptor knock-in mice (*n* = 21,306 cells) (Right) from ref. 42. Quantification after tumor implantation at 0, 10, 17, 24 and 31 days. Source data are provided as a Source Data file.

being specific markers of Detox-iCAF, IL-iCAF and IFNγ-iCAF clusters, respectively[10]. In addition, *SDC1* and *LAMP5* expression (defined as specific markers of ECM-myCAF and TGFβ-myCAF, respectively[10]) were sequentially upregulated in the ECM-myCAF and then in the TGFβ-myCAF clusters (Fig. 1G). To validate the directionality of the trajectories, we leveraged the splicing information (Fig. 1H) by using the RNA velocity method scVelo[62]. RNA velocity analyses confirmed the directionality of the transitions, including Detox-iCAF to ECM-myCAF and ECM-myCAF to TGFβ-myCAF (Fig. 1H, Left). Moreover, RNA velocity revealed the dynamics of gene expression patterns, such as the induction of *COL5A2* in the Detox-iCAF to reach stable expression in the ECM-myCAF and repression of *COL1A2* in the transition from ECM-myCAF to TGFβ-myCAF (Fig. 1H, Right).

We next performed functional assays to validate FAP+ CAF cluster trajectories identified in silico, particularly the transitions from Detox-iCAF to Wound-myCAF and ECM-myCAF, as these two last clusters are indicative of immunotherapy resistance[10]. To do so, we established primary cultures of the different FAP+ CAF clusters (see "Methods", #"Isolation and culture of primary FAP+ CAF clusters"). We confirmed that these fibroblasts were all positive for FAP, with a higher percentage of ANTXR1+ FAP+ myCAF clusters when expanded on plastic dishes and of ANTXR1− FAP+ iCAF clusters on collagen-coated dishes (Fig. 1I and Supplementary Fig. 1E, F), as observed in PDAC[5,63]. Using specific FAP+ CAF cluster markers[10], we found that primary FAP+ CAF expanded on plastic dishes were composed on average of 54% ECM-myCAF (ANTXR1+ SDC1+ LAMP5−), 14% TGFβ-myCAF (ANTXR1+ SDC1+/− LAMP5+) and 32% IFNγ-iCAF (ANTXR1− DLK1− GPC3− CD74+). Meanwhile, FAP+ CAF cultured on collagen-coated dishes were 57% Detox-iCAF (ANTXR1− DLK1+/− GPC3+), 22% IL-iCAF (ANTXR1− GPC3− DLK1+) and 18% IFNγ-iCAF (ANTXR1− DLK1− GPC3− CD74+) (Fig. 1J and Supplementary Fig. 1E, F). We next isolated pure FAP+ CAF clusters in vitro. By using the aforementioned specific markers, we sorted ECM-myCAF and TGFβ-myCAF clusters and cultured them on plastic dishes on the one hand, and we isolated Detox-iCAF, IL-iCAF and IFNγ-CAF clusters and kept them on collagen-coated dishes on the other hand (Fig. 1K). We confirmed by flow cytometry (Supplementary Fig. 1G) that these different primary FAP+ CAF clusters exhibited the same profiles for each marker as those used to characterize them in BC patient samples (Supplementary Fig. 1H). We also performed bulk RNA sequencing from each CAF population and validated that the transcriptomic profiles of the in vitro sorted CAF-S1 clusters are similar to those of the subpopulations originally identified in patients[10] (Supplementary Fig. 1I).

We took advantage of these different primary FAP+ CAF clusters in culture to analyze the mechanisms driving their plasticity and guiding their identity. Considering that the Detox-iCAF cluster was the root of FAP+ CAF cluster trajectories, we hypothesized that the Detox-iCAF cluster could give rise to other FAP+ CAF clusters in presence of cancer cells. We thus co-cultured Detox-iCAF in presence of the

luminal BC cell line MCF7 and tested the impact of this co-culture on Detox-iCAF phenotype at different timepoints (Fig. 1L and Supplementary Fig. 2A). We observed an immediate increase in the percentage of ECM-myCAF at early timepoints of the co-culture (15−30 min), followed by an increase in the content of Wound-myCAF (starting at 1 h). ECM-myCAF and Wound-myCAF clusters kept increasing gradually at later timepoints. These functional assays are consistent with the two trajectories identified in silico, *i.e.* the direct trajectory from Detox-iCAF to ECM-myCAF and the indirect path going through Wound-myCAF. At later phases of the kinetic (from 10 to 72 h), the level of IFNγ-iCAF, IL-iCAF and TGFβ-myCAF also increased, once again confirming the trajectories identified by in silico approaches (Fig. 1L and Supplementary Fig. 2A). Finally, at the latest kinetic timepoints (48 h, 72 h and 8 days), the ECM-myCAF cluster accumulated the most at the expense of the Wound-myCAF and Detox-iCAF clusters (Fig. 1L and Supplementary Fig. 2A). Moreover, co-culture of Detox-iCAF with two alternative breast cancer cell lines (T47D and MDA-MB-231) confirmed transitions from Detox-iCAF toward ECM-myCAF in presence of cancer cells (Fig. 1M, N and Supplementary Fig. 2B, C). Importantly, maintenance of Detox-iCAF alone (Supplementary Fig. 2D) or co-culture with non-tumoral breast epithelial cells (MCF10A) (Fig. 1O and Supplementary Fig. 2E) did not induce Wound-myCAF or ECM-myCAF, showing that the presence of cancer cells is required to induce these clusters. Moreover, consistent with the trajectories, neither IL-iCAF nor IFNγ-iCAF co-cultured with MCF7 were efficiently converted into Wound-myCAF or ECM-myCAF clusters (Fig. 1P, Q and Supplementary Fig. 2F, G). To confirm these observations in vivo, we leveraged previously published scRNA-seq datasets, which examined changes in TME composition following tumor implantation in mice[9,42] (Fig. 1R, S). In these mouse models, BC[9] and PDAC[42] cancer cells were transplanted, followed by sampling and scRNA-seq at different timepoints after grafting. We isolated fibroblasts from the two datasets and annotated the different clusters using label transfer, thereby identifying Detox-iCAF, Wound-myCAF and ECM-myCAF as the main FAP+ CAF clusters in these BC and PDAC mouse models (Supplementary Fig. 2H). At the time of cancer cell transplantation, we detected a high content of the Detox-iCAF cluster in the two datasets (Fig. 1R, S). Interestingly, at later timepoints, we observed a gradual loss of Detox-iCAF in favor of ECM-myCAF, and at a lesser extent, of Wound-myCAF in both BC- and PDAC-bearing mice (Fig. 1R, S, Left). As LRRC15 was also identified as a specific marker of ECM-myCAF in human BC[10] (Supplementary Fig. 2I), we took advantage of the recently published *Lrrc15*-diphteria toxin receptor knock-in PDAC mouse model[54] to investigate how FAP+ CAF cluster composition evolved after ECM-myCAF depletion in vivo. In agreement with the trajectories, selective ablation of ECM-myCAF led to a shift in the proportions of CAF-S1 clusters in favor of Detox-iCAF and Wound-myCAF clusters (Fig. 1S, Right). Moreover, when depletion was halted, we observed a rapid resurgence of ECM-myCAF with a diminution of both Detox-iCAF and Wound-myCAF (Fig. 1S,

Right). Thus, consistent with in silico trajectories among FAP+ CAF clusters, these findings show that cancer cells promote a switch from a detoxification-associated inflammatory pathway (Detox-iCAF) to a myofibroblastic signature (ECM-myCAF) in FAP+ CAF both in vitro and in vivo.

## Cancer cells convert Detox-iCAF into ECM-myCAF through a DPP4- and YAP1-dependent mechanism

We next sought to elucidate the molecular mechanisms involved in FAP+ CAF cluster plasticity, focusing on the direct transition from Detox-iCAF to ECM-myCAF, and the indirect path through Wound-myCAF. We first performed a differential analysis of the genes expressed by FAP+ CAF isolated from the direct transition versus all other FAP+ CAF from human BC single cell RNAseq data (Supplementary Fig. 2J). Using this approach, we identified *DPP4* (Dipeptidyl Peptidase 4) as the main up-regulated gene in FAP+ CAF in the direct transition between Detox-iCAF and ECM-myCAF (Fig. 2A, B). In the two scRNA-seq datasets from TNBC and PDAC mouse models, *DPP4* expression was also specifically upregulated in Detox-iCAF but progressively declined in the Wound-myCAF and ECM-myCAF clusters (Fig. 2C). Previous studies in other pathologies have demonstrated that inhibition of DPP4 can lead to reduced pathological fibrosis, which suggests that DPP4 may play a crucial role in myofibroblast formation[64,65], and highlight the pertinence of DPP4 in the direct transition from Detox-iCAF to ECM-myCAF. Regarding the indirect transition, we identified 7 YAP1-TEAD (Yes-Associated Protein–Transcriptional Enhanced Associate Domain)-target genes among the top-50 up-regulated genes in the indirect pathway via the Wound-myCAF (Supplementary Fig. 2J). Moreover, by computing transcription factor activity scores using Dorothea[66], we confirmed that TEAD activity was specifically increased in FAP+ CAF undergoing the indirect transition in human BC (Fig. 2D, E), as well as in the Wound-myCAF in TNBC and PDAC mouse models (Fig. 2F).

We next tested if DPP4- and TEAD-dependent molecular pathways were involved in the generation of ECM-myCAF from Detox-iCAF by performing functional assays in vitro. We first analyzed the impact of *DPP4* silencing (Fig. 2G) on FAP+ CAF cluster plasticity (Fig. 2H, I). *DPP4* silencing prevented the increase in ECM-myCAF content at early timepoints (15 and 30 min) and delayed the transition from Detox-iCAF to ECM-myCAF after 1 h to 2 h of co-culture (Fig. 2H, I). In contrast, *DPP4* inactivation had no impact on the transition toward Wound-myCAF (Fig. 2H, I), confirming that DPP4 might be involved in the direct transition between Detox-iCAF and ECM-myCAF but not in the indirect path through Wound-myCAF. We next tested the impact of the TEAD transcription factors on the indirect transition. As there are different members of the TEAD family, we inactivated the TEAD co-activator YAP1. While *YAP1* silencing (Fig. 2J) had no impact on the increase of ECM-myCAF at early timepoints of the co-culture (15 and 30 min), it almost completely abolished the transition from Detox-iCAF to Wound-myCAF detected after 1 h of co-culture under control conditions (Fig. 2K, L). Moreover, at 10 h and 24 h timepoints, the percentages of ECM-myCAF cells were higher upon *YAP1* silencing compared to controls (Fig. 2K, L), suggesting that the direct transition from Detox-iCAF to ECM-myCAF clusters might compensate *YAP1* silencing. Therefore, we tested the hypothesis of a compensatory mechanism between the two trajectories. Specifically, we investigated whether *DPP4* silencing caused an increase in YAP1-dependent indirect transition, and conversely, whether *YAP1* silencing leads to increased DPP4-dependent direct transition. Interestingly, inactivation of both *DPP4* and *YAP1* blocked both the direct transition from Detox-iCAF to ECM-myCAF and the indirect path through the Wound-myCAF cluster (Fig. 2M–O). Moreover, consistent with this compensatory mechanism, YAP1 protein level remained unchanged upon DPP4 inactivation and DPP4 protein level was not affected by YAP1 inactivation throughout the kinetics of co-culture with cancer cells (Fig. 2P),

suggesting that ECM-myCAF can be generated by two different trajectories and that one can replace the other when one is inactivated. Taken together, these results highlight the crucial role of DPP4 and YAP1/TEAD in driving the emergence of ECM-myCAF from Detox-iCAF through two independent mechanisms, shedding light on the molecular interactions underlying cancer immune escape.

Based on recent data showing that the TGFβ2/TGFBR2 axis drives LRRC15+ ECM-myCAF differentiation in PDAC mouse models[42], we wondered if this pathway could be involved in CAF-S1 cluster plasticity. We first observed that TGFβ2 stimulation, validated by SMAD family member 2 (SMAD2) phosphorylation, gradually increased the ECM-myCAF content (Supplementary Fig. 2K–M), suggesting that TGFβ2 is sufficient to promote the transition from Detox-iCAF to ECM-myCAF. Reciprocally, silencing of TGFBRII in Detox-iCAF prevented the transition from Detox-iCAF to ECM-myCAF in the presence of cancer cells (Fig. 2Q, R), confirming that TGFBR2 is necessary for this transition. Collectively, these data highlight the role of the TGFβ2/TGFBR2-dependent pathway in the emergence of ECM-myCAF from Detox-iCAF. Finally, we evaluated the impact of conditioned medium (CM) derived from MCF7 cancer cells on Detox-iCAF and observed that CM promoted the transition from Detox-iCAF to Wound-myCAF (Supplementary Fig. 2N, O). The pattern of induced clusters differed between TGFβ2 and CM stimulation, suggesting that CM contains other secreted signaling molecules in addition to TGFβ2 which promote this specific transition and that the TGFβ2-mediated effect requires direct contact between cancer cells and Detox-iCAF.

## Spatial organization of FAP+ CAF clusters in breast cancer

Our next goal was to gain insights into FAP+ CAF cluster plasticity based on their spatial organization in BC and interactions with surrounding cells. Thus, we performed spatial transcriptomics on 4 luminal (Lum) and 3 triple negative (TN) BC (Supplementary Fig. 3A). Three sections were isolated from the tumor bed and 4 at the invasive margin, based on pathological annotations (Fig. 3A). On average, we sequenced 2391 spots per section and captured 3896 genes per spot (Supplementary Fig. 3B). We supplemented this dataset by analyzing 10 additional publicly available BC sections[13] covering 47,830 spatial regions. Morphological annotations enabled us to differentiate tumors, invasive margins and peritumors (Fig. 3A). In the tumor compartment, pathologists distinguished cancer cells, intra-tumor stroma, as well as normal lobules and ducts. In peritumors, pathologists identified normal lobules and ducts containing normal epithelial and myoepithelial cells surrounded by basement membrane and paleal stroma, as well as interlobular conjunctive tissue and lymphocyte aggregates (Fig. 3A). For further analysis, we transferred the digital pathological annotations on the spatial data by computing the area of each pathological annotation covered by each spot (Supplementary Fig. 3C–E). We then mapped and estimated the abundance of the different cell types in each 55 μm spot by applying the deconvolution method cell2location[67]. As input to cell2location, we computed a matrix of reference cell types by generating and annotating a high-resolution BC cellular atlas that we built from newly generated BC scRNA-seq data and publicly available datasets[35,55,56] (Supplementary Fig. 4A, B). This BC atlas comprised 73,426 high-quality cells and encompassed 39 different cell types and states, thereby representing a comprehensive BC cellular landscape (Fig. 3B and Supplementary Fig. 4C). We defined the identity of each cell type by label transfer (for FAP+ CAF[10] and CAP clusters, Supplementary Fig. 4D) and marker-based annotations (Supplementary Fig. 4E). We also used Copy Number Variation (CNV) profiles to confirm cancer cell identity (Supplementary Fig. 4F). To ensure that our BC atlas was complete, we transferred our high-resolution annotations on other independent published BC scRNA-seq datasets[13,35] and confirmed that all BC cell types and states were covered (Supplementary Fig. 4G).

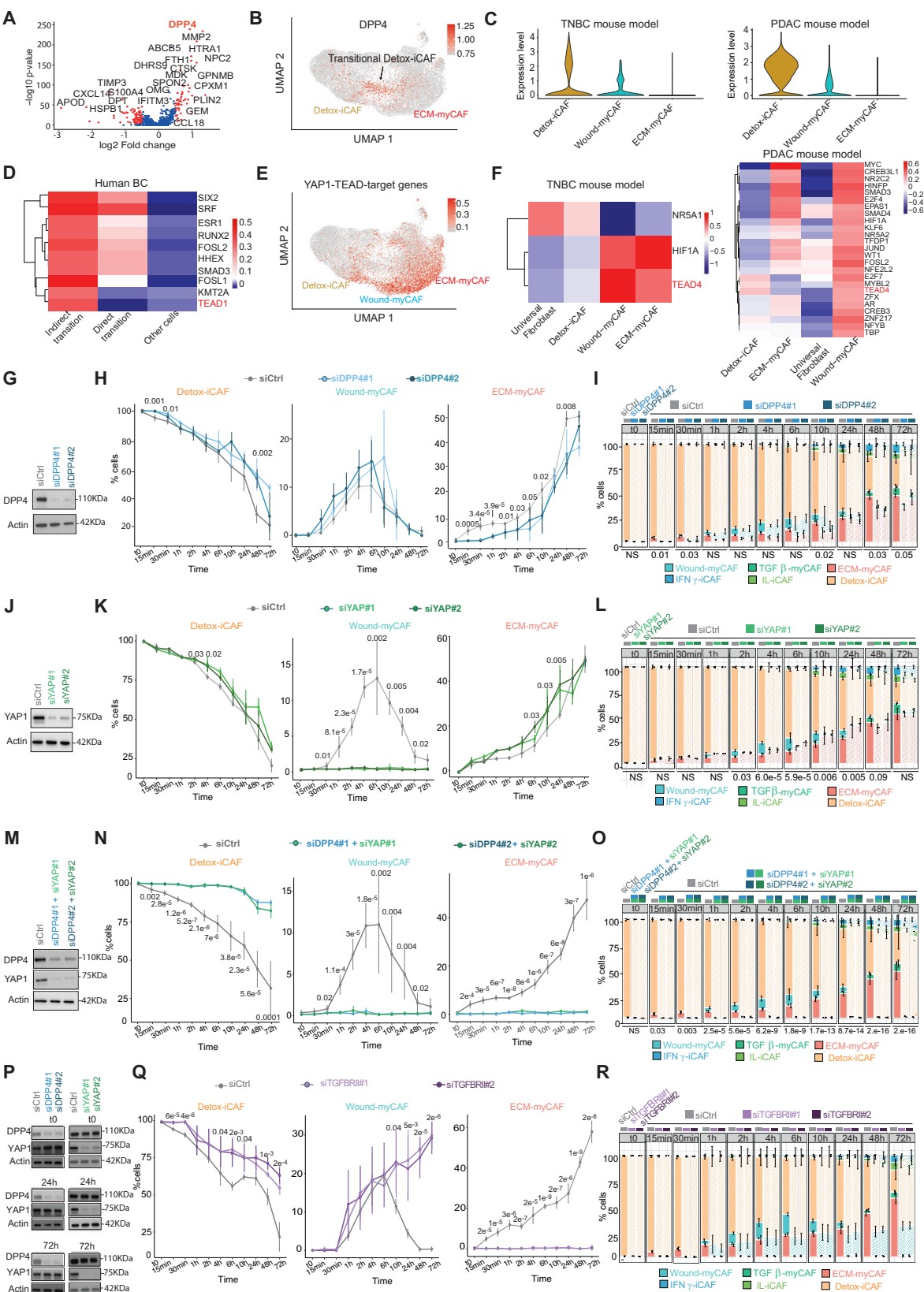

We next deconvoluted spatial transcriptomics data from the 17 BC sections at single cell-like resolution by applying cell2location[67] using our scRNA-seq BC atlas as reference (Fig. 3C and Supplementary Fig. 5A, B for representative deconvoluted sections). Several cell types identified by deconvolution were first morphologically confirmed by pathologists (Supplementary Fig. 6A–C). We also tested RCTD and

SpatialDWLS, two other top-performing deconvolution methods[68–70], both of which produced results entirely consistent with cell2location (Supplementary Fig. 6D, E). Interestingly, the deconvolution of BC spatial datasets revealed the specific localizations of different cell types and states, including FAP+ CAF clusters (Fig. 3D and Supplementary Fig. 7). We analyzed the enrichment of each cell state within

**Fig. 2 | Identification of DPP4- and YAP-1-dependent transitions of Detox-iCAF into ECM-myCAF. A** Volcano plot showing differential gene expression in FAP+ CAF from the direct transition (red cells in Supplementary Fig. 2J) compared to other CAF from the BC scRNA-seq dataset[10]. *p* values from two-sided Wilcoxon rank sum test. In red, genes with adjusted *p* value < 0.05. **B** *DPP4* expression in a subset of Detox-iCAF (transitional Detox-iCAF). **C** *DDP4* expression in FAP+ CAF clusters from scRNA-seq data of BC (Left) and PDAC (Right) mouse models[9,42]. **D** Topmost variable transcription factors in CAF in direct/indirect transition and in other CAF. **E** Expression of TEAD/YAP1-target genes in FAP+ CAF clusters. **F** Same as (**D**) for scRNA-seq data from ref. 9 (Left) and ref. 42 (Right). **G** Representative western blot showing DPP4 silencing in Detox-iCAF at the beginning of the co-culture (*t*0) from three independent experiments. Actin is internal control for protein loading. **H** Percentages of Detox-iCAF, Wound-myCAF and ECM-myCAF clusters among FAP+ CAF after co-culture of MCF7 with Detox-iCAF silenced (siDPP4) or not (siCtrl) for DPP4 (*n* = 3 independent experiments). *p* values from two-sided Student's *t* test. **I** Same as (**H**) showing the fraction of each FAP+ CAF cluster with/without DPP4 silencing (*n* = 3). *p* values from two-sided Fisher's Exact test. **J** Same as (**G**) for YAP1 silencing. **K** Same as (**H**) after co-culture of MCF7 with Detox-iCAF silenced (siYAP1) or not (siCtrl) for YAP1 (*n* = 3). **L** Same as (**I**) with/without YAP1 silencing (*n* = 3). **M** Same as (**G**) showing DPP4 and YAP1 silencing. **N** Same as (**H**) after co-culture of MCF7 with Detox-iCAF silenced (siDPP4/siYAP1) or not (siCtrl) for both DPP4 and YAP1 (*n* = 3). **O** Same as (**I**) with/without DPP4 and YAP1 silencing (*n* = 3). **P** Western blots showing DPP4 and YAP1 protein levels in Detox-iCAF silenced either for DPP4 or YAP1 at 3 timepoints of the co-culture with MCF7. **Q** Same as (**H**) with Detox-iCAF silenced (siTGFBRII) or not (siCtrl) for TGFBRII (*n* = 3). **R** Same as (**I**) with/without TGFBRII silencing (*n* = 3). All data are mean ± SEM. Source data and exact *p* values are provided as a Source Data file.

different pathological compartments and found that Detox-iCAF and IL-iCAF clusters were predominantly detected in the peritumor conjunctive tissue, with an enrichment of Detox-iCAF in peritumors and IL-iCAF in close proximity with normal ducts and lobules (Fig. 3D). Lymphocyte aggregates were found to be specifically enriched in Detox-iCAF and IFNγ-iCAF (Fig. 3D). In contrast, the ECM-myCAF cluster was the most abundant stromal subpopulation detected in the tumor bed (Fig. 3D and Supplementary Fig. 7). TGFβ-myCAF were detected within the tumor compartment but also found frequently adjacent to normal lobules (Fig. 3C and Supplementary Fig. 5B, Black arrowheads) and at the invasive margin (Fig. 3C). Interestingly, Wound-myCAF were predominantly observed in large nests of intra-tumor stroma (Fig. 3C and Supplementary Fig. 5A, Black arrows), while tumor cell-containing areas were specifically enriched in ECM-myCAF and IFNαβ-myCAF (Fig. 3C, D). Within the tumor bed, we were able to distinguish these FAP+ CAF clusters from cancer cells by detecting genomic alterations. We visualized large-scale copy number variation (CNV) in situ by applying the InferCNV algorithm on tissue sections and then confirmed that CNV were detected in cells identified as cancer cells by the deconvolution method (Supplementary Fig. 8A). This allowed us to validate that cells detected in the intra-tumor stroma did not show any genomic rearrangement and were indeed CAF and not epithelial tumor cells which underwent epithelial-to-mesenchymal transition.

Deconvolution of BC sections also showed that FAP+ CAF clusters were located in proximity to other cell types. Similarly to ECM-myCAF, IFNαβ-myCAF and TGFβ-myCAF clusters, TREM2+ TAM and SPP1+ TAM cells were detected within the tumor bed (Fig. 3C, D and Supplementary Fig. 7). FOLR2+ TAM were either detected in the Wound-myCAF-enriched intra-tumor stroma or retained at the invasive margin (Fig. 3C, D). Contractile-CAP predominated in peritumors, while Ag-CAP were prevalent within tumors (Fig. 3C, D and Supplementary Fig. 7). Endothelial cells were also differentially distributed: Angio-EC were significantly enriched in tumor nests and intratumor stroma, while ap-EC were notably scarce in tumors but particularly enriched in lymphocyte aggregates (Fig. 3D). Interestingly, we found that tumor-cell enriched areas and intratumor stroma were enriched in GZMH+ CD8+ T cells, while lymphocytes aggregates showed an enrichment in precursor GZMK+ CD8+ T lymphocytes (Fig. 3D). In conclusion, spatial deconvolution of the BC sections revealed specific patterns in the localization and organization of stromal and immune cells, highlighting a structured organization of the TME components which delineates histological regions within the BC sections.

## Unsupervised analysis reveals shared spatial cellular compositions across patients

As we observed specific localization of each cell type and state in each patient, we next sought to define cellular compositions conserved across patients. Firstly we computed the closest distances separating the different cell types in the 17 BC sections and unraveled a precise distribution of the TME components shared across patients (Fig. 4A and Supplementary Fig. 8B). ECM-myCAF, IFNαβ-myCAF, TGFβ-myCAF, SPP1+ TAM, TREM2+ TAM, regulatory lymphoid cells (FOXP3+ CD4+ Treg and NKG2A+ NKreg) and Angio-EC were detected in close vicinity to cancer cells, while Wound-myCAF, IL-iCAF, Detox-iCAF, FOLR2+ TAM, precursor T lymphocytes (SELL+ CD4+ and XCL1+ CD8+ T cells) and ap-EC were observed farther away from cancer cells (Fig. 4A and Supplementary Fig. 8B). This prompted us to identify areas with similar cell type enrichment across patients in an unsupervised manner (Fig. 4B, C). In brief, we applied Leiden clustering[71] on a batch-corrected K-nearest neighbors' graph[72] built on the deconvolution output of the 17 sections to identify communities of spots that shared a similar cell type composition across patients. Using this unsupervised approach, we identified 11 spatial niches that could be visualized directly on sections (Fig. 4B and Supplementary Fig. 8C). These niches consisted of spots sharing similar cell type enrichment but exhibiting distinct spatial organization within the tissue. We wondered if our BC spatial dataset could be used as a reference to project the cellular niches on new BC sections. To do so, we retrieved 14 publicly available BC sections[73] analyzed by spatial transcriptomics but lacking niche annotations. After deconvolution, we built a latent embedding using scANVI with all sections as input and used the label transfer capacities of scANVI to evaluate the detection of the niches in the new sections (see "Methods", #"Niche reference mapping"). This approach revealed that we were able to re-identify the different niches in new BC sections and that the spatial organization of the mapped niches closely mirrored the original niches (Supplementary Fig. 8D, E). Thus, this analysis showed that our breast cancer spatial dataset can serve as reference in future studies to map BC cellular niches on new data. We next aimed to identify unique patterns of cell types that preferentially coexisted within specific niches. To do so, we applied hierarchical clustering on the mean abundance of cell types per niche. This allowed us to identify 10 co-localization patterns within the niches that were shared across sections (Fig. 4C). We refer to these co-localization patterns as EcoCellTypes (ECT) which stands for ecosystem of cell types (see #"Methods", #"Niches and ECT identification") (Fig. 4C). ECT represented cell types and states co-existing across BC patients within the same tumor area (e.g. each area being a niche). Interestingly, some ECTs were found together in specific niches but not others, highlighting the interest of this approach to discover niche-specific colocalization pattern. Thus, each ECT contains a specific composition of cell types or states localized in close proximity, and we identified specific ECT enriched in different FAP+ CAF clusters. The first ECTs (ECT1-6) were predominantly observed at distance from cancer cells. The "Detox-iCAF-enriched immuno-protective ECT4" highlighted the spatial co-occurrence of Detox-iCAF, ap-EC, Mo-DC, and FOLR2+ TAM, characterized by their localization in the peritumoral zone but also within the tumor bed, forming an intra-tumoral peritumor-like stroma. The "IL-iCAF-enriched stroma ECT5", composed of IL-iCAF, Adipo-EC, and Contractile-CAP, encompassed cell

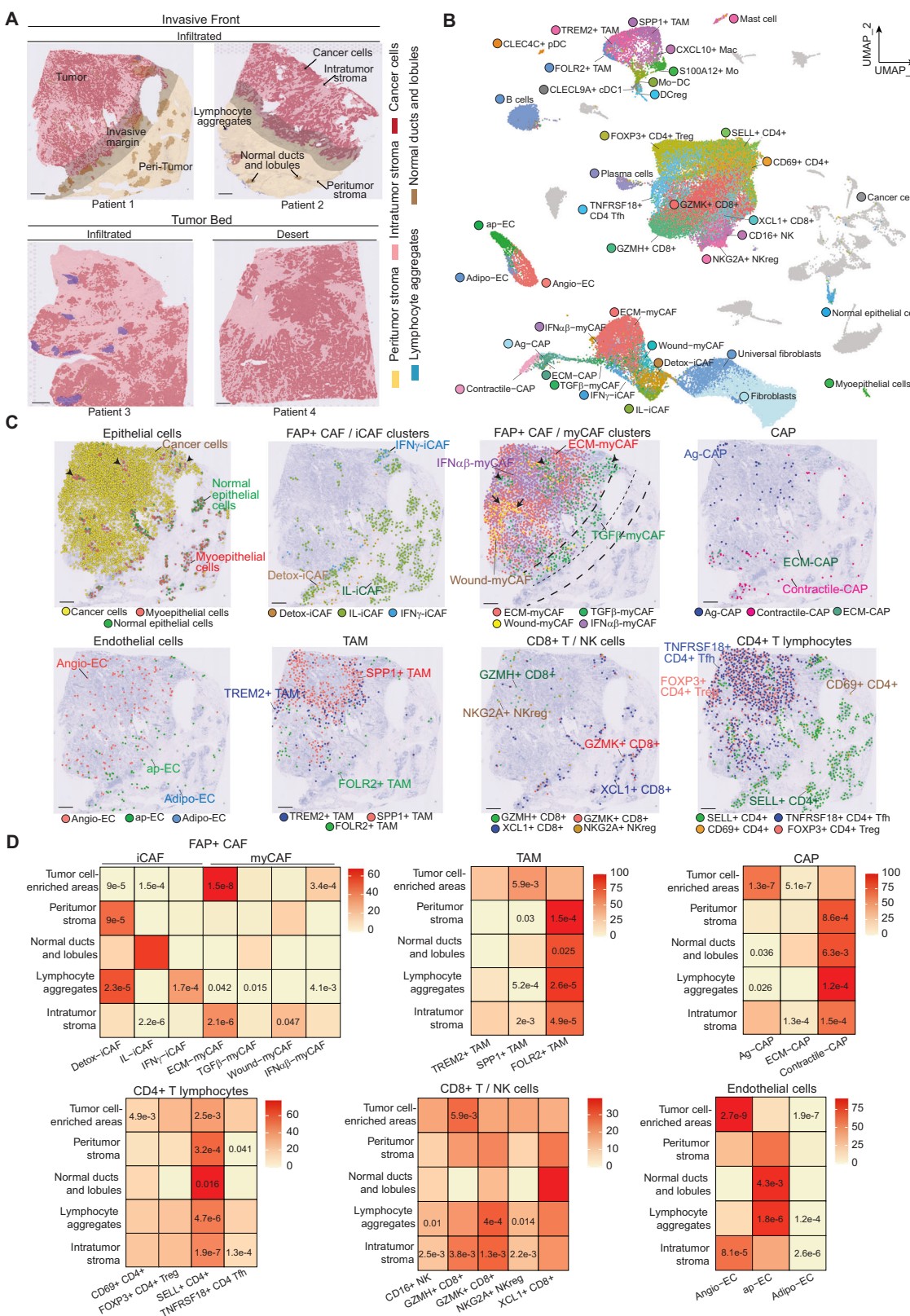

types associated with the "Normal epithelial structures ECT6", formed by normal epithelial and myoepithelial cells. Both ECT1 and ECT2 were enriched in immune cells. Indeed, the "IFNγ-iCAF-enriched immuno-suppressive ECT1" contained immuno-suppressive cell types including FOXP3+ Treg or NKG2A+ NKreg, while the "Precursor immune cell ECT2" contained more precursor or cytotoxic cell states like SELL+ CD4+ and GZMK+ CD8+ T lymphocytes. ECT7-10 gathered cell types strongly associated with cancer cells. The "IFNαβ-myCAF-enriched cancer cell ECT7" encompassed cancer cells, IFNαβ-myCAF, and SPP1+ TAM. Interestingly, the "ECM- and TGFβ-myCAF-enriched ECT9" also spatially overlapped with ECT7, revealing the proximity of cancer cells with ECM-myCAF, TGFβ-myCAF, Angio-EC, and Ag-CAP. Finally, the "Wound-myCAF-enriched intratumoral stroma ECT10" brought together Wound-myCAF, ECM-CAP, and TREM2+ TAM.

**Fig. 3 | Spatial organization of breast cancer microenvironment. A** H&E images of representative Lum BC sections processed by Visium and annotated by pathologists. Tumor areas are colored in red, with cancer epithelial cells in dark red and intra-tumor stroma in light red. Invasive margins are highlighted in gray. Normal peritumor tissues include normal ducts and lobules in brown and peritumor stroma in yellow. T lymphocyte aggregates are in blue. (*N* = 17 sections in total). Scale bars = 500 μm. **B** UMAP of 73,426 cells from 43 patients (34 BC patients and 9 healthy donors) encompassing 39 different cell types and states and composing a comprehensive BC cellular atlas. **C** Deconvolution at single cell-like resolution based on the cellular atlas showed in (**B**) on a representative BC section (see also Supplementary Fig. 5 for deconvolution of additional BC sections). Each dot shows one single cell and the different colors represent distinct cell types and states. Black arrowheads indicate normal lobules and ducts (Panel epithelial cells) and their co-localization with TGFβ-myCAF (Panel FAP+ CAF/myCAF clusters). Dashed lines delineate the invasive margin. Scale bars = 500 μm. **D** Heatmap of the median proportion of each cell state among the corresponding cell type within each pathological compartment (Tumor cell-enriched areas; Intratumor stroma; Peritumor stroma; Lymphocyte aggregates and Normal ducts and lobules), as shown in (**A**). Exact *p* values are shown for significant (*p* < 0.05) enrichment (red and orange) or depletion (yellow) (% indicated on scale bars) of a cell state compared to the others within a particular pathological annotation. *p* values from two-sided Wilcoxon (one versus all) rank sum test. Source data are provided as a Source Data file.

To investigate if the composition of these ECT differed between BC molecular subtypes, we performed deconvolution of transcriptomic data from the METABRIC cohort (487 Lum A, 368 Lum B, 193 HER2 and 186 Basal-like TN BC) using BayesPrism[74]. In terms of immune components, we found that Lum A BC exhibited a lower fraction of immuno-suppressive cells (ECT1) compared to Lum B, HER2, and Basal-like TN BC (Fig. 4D). Conversely, Lum A BC displayed a higher proportion of naive CD4 T cells, as well as precursor and cytotoxic CD8+ T lymphocytes (which constituted the ECT2) compared to the other BC subtypes. We also observed significant differences in the stromal compartment: Lum A BC contained more ECT5 (composed of IL-iCAF, Adipo-EC, contractile-CAP) and ECT9 (ECM-myCAF, TGFβ-myCAF, Ag-CAP) compared to Lum B, HER2 and Basal-like TN BC. In contrast, ECT10 (Wound-myCAF, ECM-CAP and TREM2+ TAM) was enriched in Lum B and HER2 and particularly abundant in Basal-like TN BC, while ECT4 (Detox-iCAF, FOLR2+ TAM) accumulated in Lum A (Fig. 4D). Remarkably, ECT composition in BC allowed us to stratify patients into 4 subgroups (C1 to C4) with different overall survival (Fig. 4E, F). Patients in the C2 subgroup, with an enrichment in ECT1 (immuno-suppressive), ECT3 (plasma cells) and ECT10 (TREM2+ TAM, Wound-myCAF), were mainly composed of Basal-like TN (31.5%), Lum B (31.7%) and HER2 (23%) BC subtypes and showed the worst survival, as expected. In contrast, patients in the C3 subgroup—enriched in ECT2 (immune precursors, effectors), ECT4 (Detox, FOLR2+ TAM) and ECT5 (IL-iCAF, adipo-EC)—were mainly composed of the Lum A subtype (69.6%) and showed the best overall survival. The C1 and C4 subgroups were mainly composed of Luminal patients (Lum A and Lum B). While their 5-year survival was good and comparable to that of the C3 subgroup, the prognosis of the C1 and C4 subgroups fell after 5 years. Compared to the C3 subgroup, these survival differences could be explained in part by the C4 subgroup's enrichment in Lum B BC and ECT6-7 (Epithelial cells, IFNαβ-myCAF, SPP1+ TAM) and the C1 subgroup's abundance of ECT9 (Angio-EC, ECM-myCAF, TGFβ-myCAF) (Fig. 4E, F). In conclusion, ECT were associated with the overall survival of BC patients, which is linked to ECT-enrichment in BC molecular subtypes. To understand the reciprocal crosstalk between each cell type in these distinct ECT, we applied the CellChat method[75] on the BC atlas to infer intercellular communications. Global ligand-receptor (L-R) analysis of all cell types revealed that FAP+ CAF clusters were the main senders of L-R signals in the TME (Fig. 5A), suggesting they could play a key role in the ECT organization. Moreover, the strength of the signals sent and received by the FAP+ CAF increased progressively from universal fibroblasts to iCAF and reached a maximum with ECM-myCAF, TGFβ-myCAF, and IFNαβ-myCAF (Fig. 5A), following similar trajectories as the one described above (Fig. 1). In addition, the number of interactions found between cancer cells and FAP+ CAF clusters by CellChat reflected their spatial proximity in the TME. Indeed, ECM-myCAF and IFNαβ-myCAF displayed the highest numbers of signals sent to and received from cancer cells, while IL-iCAF and Detox-iCAF showed the least (Supplementary Fig. 8F). One of the strongest interactions detected between FAP+ CAF clusters and the other cell types was the collagen-dependent signaling pathway, in particular in the myCAF clusters (Supplementary Fig. 8G). This is

consistent with the high expression of ECM proteins in these FAP+ CAF clusters and the role of YAP1 in the transition of Detox-iCAF into Wound- and ECM-myCAF clusters. In summary, we defined 10 ECT characterizing the spatial co-occurrence of different cell types or cell states in BC, and their reciprocal interactions.

We next validated the composition of immuno-protective and immuno-suppressive ECT by testing the correlative link between cellular populations located in close proximity. Part of the protective ECT4 was composed of Detox-iCAF, monocytes and FOLR2+ TAM, and the immuno-suppressive ECT (ECT9 and ECT10) of ECM-myCAF, Wound-myCAF, TGFβ-myCAF and TREM2+ TAM. As anticipated from ECT composition, using flow cytometry from BC samples (Prospective cohort, Supplementary Table 1), we observed that Detox-iCAF showed a negative correlation with ECM-myCAF, TGFβ-myCAF, and Wound-myCAF and a positive correlation with IL-iCAF and IFNγ-iCAF in BC patients (Fig. 5B). Similarly, consistent with ECT9 and ECT10, we confirmed a positive correlation between ECM-myCAF, TGFβ-myCAF and Wound-myCAF. We also validated cell co-occurrence in the immuno-suppressive niche by demonstrating that these myCAF clusters were positively correlated with TREM2+ TAM and negatively correlated with FOLR2+ TAM (Fig. 5C). Similarly, Detox-iCAF content showed a positive correlation with FOLR2+ TAM, consistent with the immuno-protective ECT4, and a negative correlation with TREM2+ TAM proportion (ECT10) (Fig. 5C). To investigate if FAP+ CAF clusters actively modulated the identity of myeloid cells or only attracted them to form the ECT, we performed Transwell assays to test monocyte attraction toward different FAP+ CAF clusters. Our results demonstrated that Detox-iCAF, IL-iCAF, and IFNγ-iCAF enhanced monocyte migration, while ECM-myCAF and TGFβ-myCAF did not (Fig. 5D). Upon co-culture of FAP+ CAF clusters with CD14+ monocytes, we observed an increase in the proportion of CD16+ cells among CD14+ monocytes (Fig. 5E and Supplementary Fig. 9A). In addition, ECM-myCAF and TGFβ-myCAF strongly increased the proportion of TREM2+ macrophages among total CD14+ CD16+ myeloid cells (Fig. 5F and Supplementary Fig. 9A), while Detox-iCAF, IL-iCAF, and IFNγ-iCAF increased the content of FOLR2+ macrophages (Fig. 5G and Supplementary Fig. 9A). These findings suggest that ECM-myCAF and TGFβ-myCAF play an active role in modulating the identity of myeloid subtypes to form the immuno-suppressive niche, while Detox-iCAF attract monocytes and induce a FOLR2+ phenotype to form the immuno-protective niche. We also tested the impact of CAF-S1 clusters on the differentiation of CD4+ CD25+ FOXP3+ T lymphocytes in vitro. We observed that ECM-myCAF and TGFβ-myCAF increased the percentages of FOXP3+ T cells among the CD4+ CD25+ population, while iCAF clusters had no impact (Fig. 5H and Supplementary Fig. 9B). Moreover, ECM-myCAF and TGFβ-myCAF significantly increased the percentages of both PD-1+ and CTLA4+ T cells among FOXP3+ T lymphocytes (Fig. 5I, J and Supplementary Fig. 9B). Given this impact on monocytes and T cells, we next wondered whether some CAF-S1 clusters might also affect NK cell phenotype. By co-cultivating FAP+ CAF clusters with NK cells, we showed that ECM-myCAF and TGFβ-myCAF reduced perforin and granzyme B levels and significantly increased

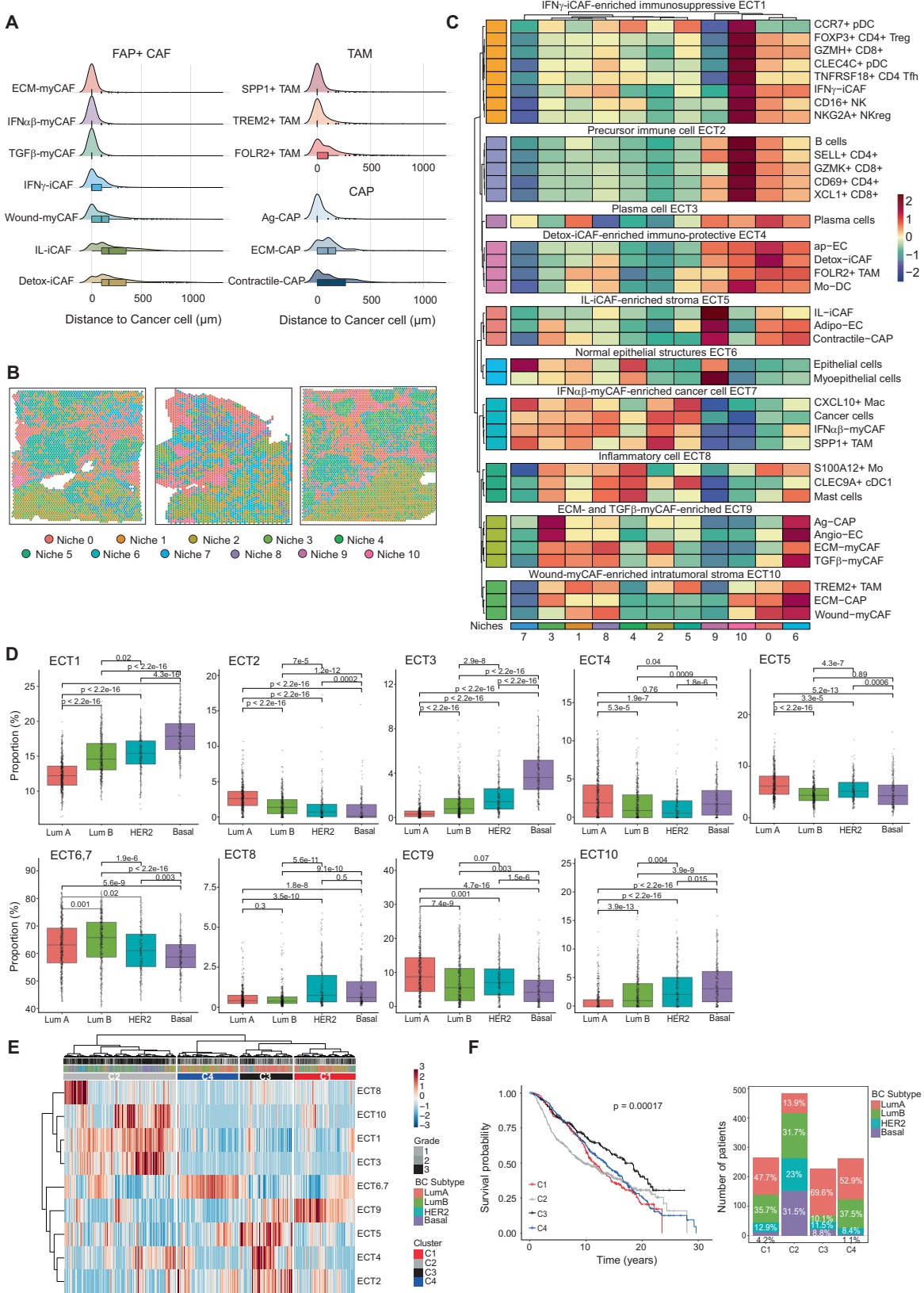

the fraction of CD56^high NK cells and their surface NKG2A levels, thereby decreasing NK cell cytotoxicity (Fig. 5K–N and Supplementary Fig. 9C). In conclusion, FAP+ CAF clusters are differentially associated with specific in situ ECT by attracting and modulating the identity of immune cells in human BC.

## Specific FAP+ CAF clusters are associated with breast cancer invasiveness

We next wondered if BC cellular composition could be associated with invasive properties. To address this question, we analyzed a series of Ductal Carcinoma in Situ (DCIS) lesions, micro-invasive DCIS (MI-DCIS)

**Fig. 4 | Identification of shared spatial cellular compositions across patients, called EcoCellTypes. A** Distribution of FAP+ CAF, CAP and TAM populations according to the distance to cancer cells (in μm). Distances are computed between the closest cancer cell identified by deconvolution and FAP+ CAF clusters, CAP and TAM in the 17 sections. Cell types are ranked based on their median distance to cancer cells. **B** Spatial distribution of 11 cellular niches in 3 representative BC patients. **C** Heatmap showing the mean cell type composition per niche identified on the 17 sections. Values are centered and scaled per cell type and state. Hierarchical clustering in rows defines 10 different groups of co-occurring cell types in the cellular niches referred to as EcoCelltypes (ECT). **D–F** Data from the METABRIC cohort (N = 1234 BC patients). **D** Proportions of each ECT in Lum A (N = 487), Lum B (N = 368), HER2 (N = 193) and Basal-like TN (N = 186) BC subtypes. p values from Mann–Whitney test. **E** Heatmap and clustering of all BC samples (columns) from the METABRIC cohort showing 4 subgroups of patients (C1 = 263, C2 = 483, C3 = 227, C4 = 261) with different ECT enrichments. **F** Left, Kaplan–Meier curves showing overall survival of the 4 BC patient subgroups (C1–C4) stratified in the heatmap. p value from Log-rank test. Right, Distribution of the BC molecular subtypes within the four subgroups (C1–C4) of BC patients. In all boxplot the center line, box limits and whiskers indicate the median, upper and lower quartiles and 1.5 × interquartile range. Source data are provided as a Source Data file.

BC (i.e. DCIS lesions with invasive foci not exceeding 1 mm) and Invasive Breast Cancer (IBC) (Supplementary Table 2 for detailed description of the INVADE cohort). We performed bulk RNA sequencing (RNA-seq) on these lesions and applied BayesPrism[74] to identify the different cellular populations comprising the tumors. We determined cellular proportions by deconvoluting the bulk RNA-seq using the BC cell atlas we built in this study (shown in Fig. 3B) as reference. Firstly, we evaluated the global cell type diversity of each tumor by computing the Shannon index and observed that IBC were more heterogenous than DCIS and MI-DCIS (Fig. 6A). By analyzing the proportion of epithelial, fibroblastic, immune and endothelial cells in each sample, we observed that IBC contained more fibroblasts compared with DCIS and MI-DCIS samples (Fig. 6B). This observation was independent of the tumor cellularity between the BC invasive types (Supplementary Fig. 10A) and consistent with a recent analysis of DCIS and IBC lesions[53]. We next determined the proportion of each cell type and cell state identified in our BC atlas by deconvolution and observed that cell type composition was distinct according to BC invasiveness (Fig. 6C, D and Supplementary Fig. 10B, C). In particular, we observed a significant accumulation of universal fibroblasts (identified in healthy tissues) in DCIS (Supplementary Fig. 10C), while the significant increase in fibroblasts in IBC was mainly due to FAP+ CAF (Fig. 6C). Regarding the FAP+ CAF, pre-invasive tumors contained a higher proportion of iCAF clusters compared to invasive lesions, with DCIS accumulating more Detox-iCAF and MI-DCIS more IL-iCAF (Fig. 6D and Supplementary Fig. 10C). The three invasive BC stages also exhibited distinct myCAF contents, with higher proportion of TGFβ-myCAF in DCIS and elevated levels of both ECM-myCAF and IFNαβ-myCAF in IBC (Fig. 6D and Supplementary Fig. 10C). In comparison to DCIS and MI-DCIS, IBC showed an accumulation in CD8+ T lymphocytes and myeloid cells (particularly in TREM2+ and SPP1+ TAM rather than FOLR2+ TAM) (Fig. 6C, D) but lower numbers of CAP (Fig. 6C). Within the CAP, IBC was associated with an increase in Ag-CAP and a decrease in the ECM-CAP fraction (Fig. 6D and Supplementary Fig. 10C). This shift in cell type composition observed from DCIS to IBC was in agreement with trajectory inferences and spatial analysis described above in our study.

We next performed spatial Ligand-Receptor analysis using the SpaTalk method[76]. We identified TGFBR2 among the top-10 receptors expressed by Detox-iCAF which can drive key interactions with cancer cells, consistent with the functional assays shown above and recent data showing that the TGFβ2/TGFBR2 axis acts in the emergence of LRRC15+ ECM-myCAF in PDAC mouse models[42]. Importantly, tumor cells at the invasive margin, communicated with Detox-iCAF through the TGFβ2/TGFBR2 ligand-receptor pair (Fig. 6E). In turn, upon TGFBR2 stimulation, Detox-iCAF increased YAP1 activity, as identified by the ligand-receptor transcription-factor knowledge-graph-based approach implemented in SpaTalk (Fig. 6F) (see also "Methods", #"Ligand-receptor analysis"), consistent with the role of YAP1/TEAD in FAP+ CAF plasticity we described above. To provide insights on the fact that we detected fewer ECM-myCAF and more Detox-iCAF in DCIS, we analyzed publicly available spatial transcriptomic data from a DCIS section (Fig. 6G, H). The analysis revealed that the localization of ECM-myCAF, TGFβ-myCAF and IFNαβ-myCAF was mainly restricted to the periphery of the ducts around tumor cells. In contrast, Wound-myCAF were found in the stroma between tumor nests and far from cancer cells, suggesting that the direct contact with cancer cells was not necessary for their induction (Fig. 6H). In conclusion, our findings demonstrate that there is a significant increase in FAP+ CAF, particularly ECM-myCAF, in IBC compared to DCIS.

## Detox-iCAF and TGFβ-myCAF proportions predict progression from DCIS to IBC

We next wondered whether the content in specific cell types in DCIS could be associated with risk of progression. As the composition of FAP+ CAF clusters is different in DCIS compared to IBC, we wondered whether their levels in DCIS at diagnosis could predict DCIS recurrence and progression to IBC. To address this question, we studied bulk-RNAseq data from the TBCRC 038 cohort[77], including 216 patients diagnosed with DCIS matched on grade and age at diagnosis. Among them, 121 showed either a DCIS recurrence (n = 66) or an IBC recurrence (n = 55). Interestingly, the FAP+ CAF population displayed a significant difference in composition between patients with and without recurrences, while no significant change was observed in other stromal and immune cell populations (Fig. 7A and Supplementary Fig. 10D). Indeed, patients with a subsequent IBC recurrence had markedly less Detox-iCAF and an increased proportion of TGFβ-myCAF (Fig. 7A). Even with a lower number of patients, we could still validate this result in the 18 DCIS patients from the INVADE cohort, where 13 patients showed no recurrence and 5 patients progressed (Fig. 7B). We next sought to evaluate the prognostic value of Detox-iCAF and TGFβ-myCAF content in DCIS patients. Median stratification of patients based on the Detox-iCAF content significantly predicted any type of recurrences, either DCIS or IBC (Fig. 7C, Left). Importantly, multivariate Cox regression analysis showed that the prognostic value of Detox-iCAF on recurrence was independent of PAM50 classification (Fig. 7C, Right). As the TBCRC 038 cohort was initially matched on the grade[77], the prognostic impact of Detox-iCAF was not confounded by grade. When focusing on IBC recurrence, we confirmed that DCIS patients enriched in Detox-iCAF harbored a lower risk of invasive recurrence (Fig. 7D, Left). In addition, patients enriched in TGFβ-myCAF had a higher risk of IBC recurrence (Fig. 7D, Right). We then hypothesized that DCIS patients harboring both high levels of Detox-iCAF and low levels of TGFβ-myCAF at diagnosis may have a reduced risk of IBC recurrence. Indeed, when considering these two factors in combination, we observed a more robust prognostic value than when considering them separately. This analysis allowed us to identify a subgroup of DCIS patients with a potentially lower risk of invasive progression (Fig. 7E). Our findings thus suggest that FAP+ CAF heterogeneity might be an important factor in DCIS progression and provide preliminary insights for addressing the issue of overtreatment in DCIS, as patients with a low-risk of progression might benefit from therapeutic de-escalation.

## Discussion

Our study provides insights into the spatial organization of the BC TME with a focus on FAP+ CAF diversity, plasticity and their interactions with surrounding cells. Here, we provide a comprehensive map of the

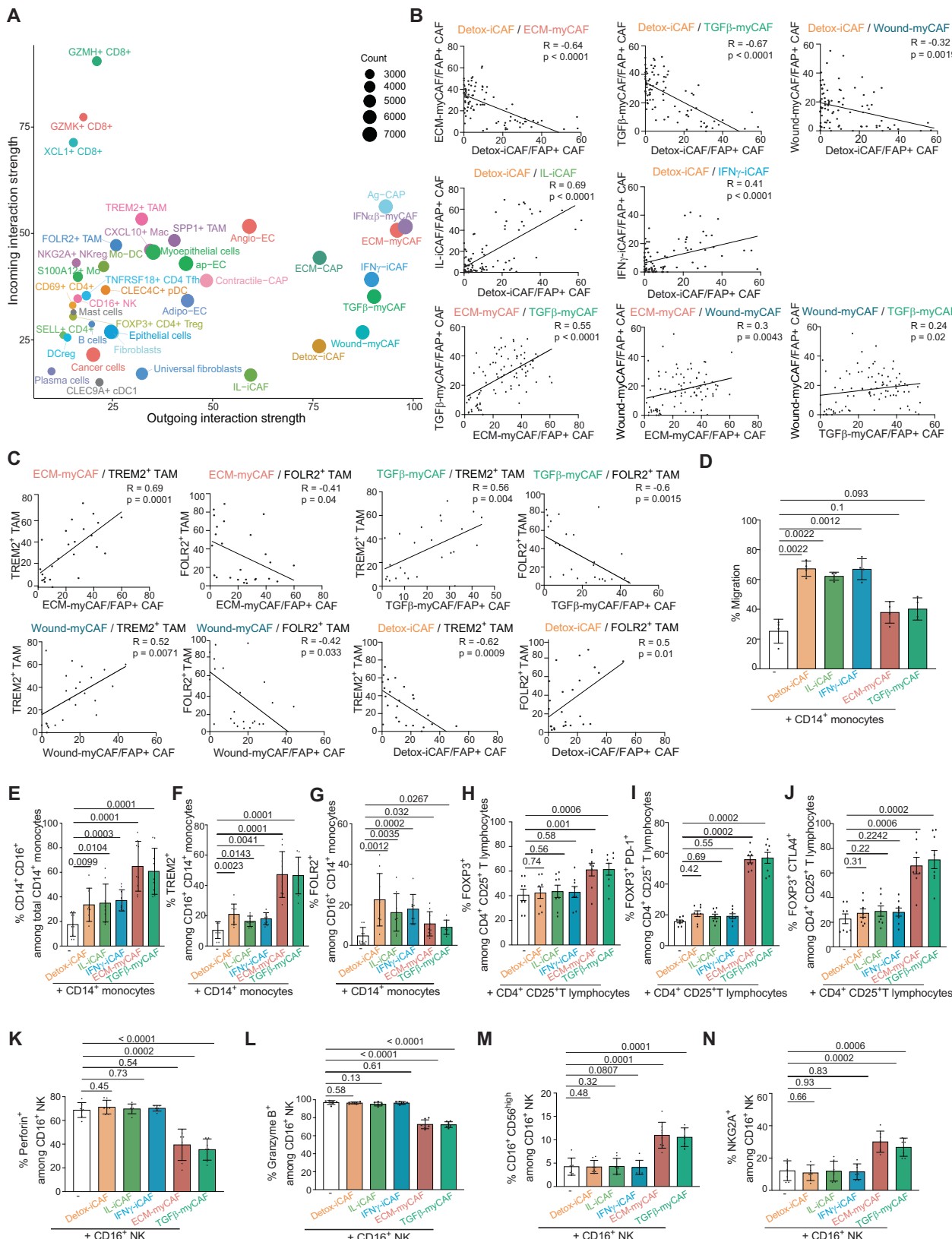

different FAP+ CAF clusters, by integrating spatial transcriptomics with a comprehensive cell atlas based on scRNA-seq data and estimating cell-type and cell-state compositions at single cell level. Based on both deconvoluted data and histological annotations, we identified the spatial organization of different functional states of FAP+ CAF populations, T lymphocytes and TAM, all of which accumulate differentially within the tumor (in proximity of cancer cells), the intratumor stroma and the peritumor space. By this way, we delineate FAP+ CAF cluster localization and identify 10 CAF-related EcoCellTypes (ECT). Recently, two co-existing sub-tumor microenvironments have been characterized in PDAC[78], which display differences in their immune infiltration and iCAF/myCAF accumulation. Our study increases the resolution of

**Fig. 5 | Interactions between FAP+ CAF clusters and immune cells in BC.**
**A** CellChat dominant sender and receiver plot showing incoming and outgoing interaction strength for each cell type identified in the BC atlas (73,426 cells from 43 patients). The size of each circle corresponds to the total number of significant interactions, colored per cell type. **B** Scatter plots with linear regression lines showing correlations in the content of FAP+ CAF clusters quantified by flow cytometry in BC ($N = 87$ patients). $p$ values from two-sided Pearson's correlation test. **C** Same as (**B**) analyzing correlations between the content in ECM-myCAF, TGFβ-myCAF, Wound-myCAF and Detox-iCAF with TREM2+ or FOLR2+ macrophages in BC ($N = 25$). **D** Bar plot showing the percentages (%) of migration of CD14+ monocytes after 6 h of transwell co-culture with FAP+ CAF clusters. Data are mean ± SEM ($n = 4$ independent experiments). $p$ values from two-sided Student's $t$ test. **E** % of CD14+ CD16+ myeloid cells among total CD14+ monocytes after 24 h of co-culture with FAP+ CAF clusters (Detailed gating strategy in Supplementary Fig. 9A). Data are mean ± SEM ($n = 9$). $p$ values from two-sided Student's $t$ test. **F** Same as (**E**) for

TREM2+ macrophages. $p$ values from two-sided Student's $t$ test. **G** Same as (**E**) for FOLR2+ macrophages. $p$ values from two-sided Mann–Whitney test. **H** % of FOXP3+ regulatory T cells among CD4+ CD25+ T lymphocytes after 16 h of co-culture with FAP+ CAF clusters (Detailed gating strategy in Supplementary Fig. 9B). Data are mean ± SEM ($n = 8$). $p$ values from two-sided Mann–Whitney test. **I** Same as (**H**) for the % of PD-1+ FOXP3+ T lymphocytes. $p$ values from two-sided Mann–Whitney test. **J** Same as (**H**) for the % of CTLA-4+ FOXP3+ T lymphocytes. $p$ values from two-sided Mann–Whitney test. **K** % of Perforin+ among total CD16+ NK cells after 24 h of co-culture with FAP+ CAF clusters (Detailed gating strategy in Supplementary Fig. 9C). Data are mean ± SEM ($n = 7$). $p$ values from two-sided Student's $t$ test. **L** Same as (**K**) for Granzyme B+ NK cells. $p$ values from two-sided Student's $t$ test. **M** Same as (**K**) for CD16+ CD56[high] NK cells. $p$ values from two-sided Student's $t$ test. **N** Same as (**K**) for NKG2A+ NK cells. $p$ values from two-sided Student's $t$ test. Source data are provided as a Source Data file.

FAP+ CAF populations, and describes the spatial organization of the entire TME in BC by concomitantly investigating the distribution of 39 different cell types and states, including the 7 FAP+ CAF clusters. Our results also expand on data from a recent study, which identified 9 groups of co-occurring cell types using a bulk RNA-seq dataset[13]. Indeed, we demonstrate that the spatial proximity of different FAP+ CAF clusters is a critical determinant of the TME, with co-occurring cell types directly influencing each other's identity and thus shaping distinct ECT (see Model, Fig. 8). While the FAP+ myCAF clusters—namely ECM-myCAF, TGFβ-myCAF and IFNαβ-myCAF—are observed in close proximity to cancer cells, the FAP+ Detox-iCAF cluster is detected around blood vessels. This specific spatial distribution suggests an adventitial origin of Detox-iCAF, with their plasticity driven by interactions with cancer cells. Here, we show that the FAP+ Detox-iCAF cluster can serve as a reservoir able to give rise to all the other FAP+ CAF clusters. A recent pseudotime trajectory analysis in BC defined a mesenchymal stem cell state characterized by *ALDH1A1* expression[13], a gene also highly expressed in Detox-iCAF. *ALDH1A1* expression decreases while that of *COL1A1* increases when cells transition toward a myofibroblast-like state[13]. This is in agreement with the trajectories we identified between Detox-iCAF and ECM-myCAF clusters. The specific spatial distribution of FAP+ CAF clusters suggests that stroma can shape the intratumor architecture, as recently shown[78–80]. We confirmed these observations and shed light on YAP1 and DPP4-dependent mechanisms. While myofibroblast differentiation has been shown to involve YAP1 activation in various fibrotic models[81–83], its role in CAF plasticity is much less known in BC. Here, we found that the Detox-iCAF cluster can give rise to the activated myCAF state in the presence of cancer cells through two main paths: an indirect transition mediated by the YAP1-signaling pathway passing through the Wound-myCAF cluster and a DPP4-dependent direct transition between Detox-iCAF and ECM-myCAF clusters (see Model, Fig. 8). DPP4 has also been recently implicated in the transition from normal fibroblasts to iCAF in mice[84]. Moreover, genetic evidence in PDAC mouse models showed that iCAF can be converted into LRRC15+ myCAF (LRRC15 being a specific marker of ECM-myCAF[10]) by up-regulating the TGFBR2-dependent signaling pathway[5,42]. Consistent with these findings, YAP- and TGFβ-signaling pathways can act simultaneously to promote a cellular transition of DPP4+ adipocyte progenitors toward DPP4− SMA+ myofibroblasts[85,86]. Moreover, we have provided here a spatial resolution of these interactions by using SpaTalk and showing the TGFβ-mediated crosstalk between Detox-iCAF and cancer cells at the invasive margin leading to YAP1/TEAD activation. In line with previous studies showing that 3-dimensional culture of mouse PDAC models can promote conversion between iCAF and myCAF[5], we observed that collagen-coated plates increase the proportion of iCAF. As we found that YAP1 is a key player in the indirect transition from Detox-iCAF to Wound-myCAF and subsequently to ECM-myCAF, we can hypothesize that the slight reduction of the stiffness in collagen-

coated dishes might favor iCAF maintenance in culture. Thus, these studies, as well as our results, reveal that FAP+ CAF populations can convert into one another depending on the spatial and biological context.

iCAF and myCAF clusters have been shown to be negatively correlated in patient samples and spatially segregated in tumors across human cancer[1,5,10,13,49,77,87,88]. Our current work goes a step further in the resolution of FAP+ CAF populations by identifying the spatial distribution of the different FAP+ iCAF and myCAF clusters, and their proximity to other cell types. Our functional analysis revealed that Detox-iCAF, primarily located in the interlobular stroma, can give rise to ECM-myCAF through two transitions, one direct and one indirect via the Wound-myCAF cluster. Notably, ECM-myCAF are systematically close to cancer cells, while Wound-myCAF are located further away. This suggests that juxtacrine interactions between Detox-iCAF and cancer cells may be required for the direct transition, while paracrine signaling could induce the larger area of Wound-myCAF. This is supported by our experiments showing that cancer cells-conditioned media induce Wound-myCAF from Detox-iCAF but not ECM-myCAF, while co-culture with cancer cells induces both. Observations in DCIS support this hypothesis, as we identified ECM-myCAF directly adjacent to the tumor nests and Wound-myCAF farther away. Specific FAP+ CAF clusters and immune cells are spatially organized in tumors, offering insights into FAP+ CAF-mediated immunoregulation. We uncovered the crosstalk between different FAP+ CAF clusters and immune cells subtypes, which modulates immune cell identity and states, highlighting immuno-permissive and immunosuppressive ECT in BC. We previously demonstrated that TGFβ-myCAF can be induced from ECM-myCAF through interactions with T lymphocytes[10]. Interestingly, we found that TGFβ-myCAF are enriched in immune infiltrated sites in BC, including the invasive margin, around intra-tumoral blood vessels, and around intra-tumoral lobules. We also identify ECT enriched in Detox-iCAF, FOLR2+ TAM and ap-EC. FOLR2+ TAM are known to be close to peritumoral blood vessels in BC[89]. Consistent with these observations, Detox-iCAF attract monocytes and induce a FOLR2+ TAM program. Detox-iCAF are localized close to ap-EC-enriched blood vessels, which express markers of tumor-associated high endothelial venule (TA-HEV), suggesting that Detox-iCAF may play a role in the crosstalk with ap-EC and TA-HEV maturation. In contrast to Detox-iCAF, IL-iCAF are primarily associated with normal breast epithelial structures and found in the intralobular stroma enriched in Adipo-EC, while IFNγ-iCAF are associated with lymphocyte aggregates and enriched in the tumor bed. We observed that IL-iCAF are not able to differentiate into ECM-myCAF, suggesting that myofibroblasts observed in BC might not be primarily derived from the intralobular stroma. As the intralobular stroma is not present in mice[90], this may explain why IL-iCAF are not detected in scRNA-seq data from mouse models. Moreover, ECM-myCAF, IFNαβ-myCAF and TGFβ-myCAF accumulate within the invasive compartment, with ECM-myCAF and IFNαβ-myCAF in the close

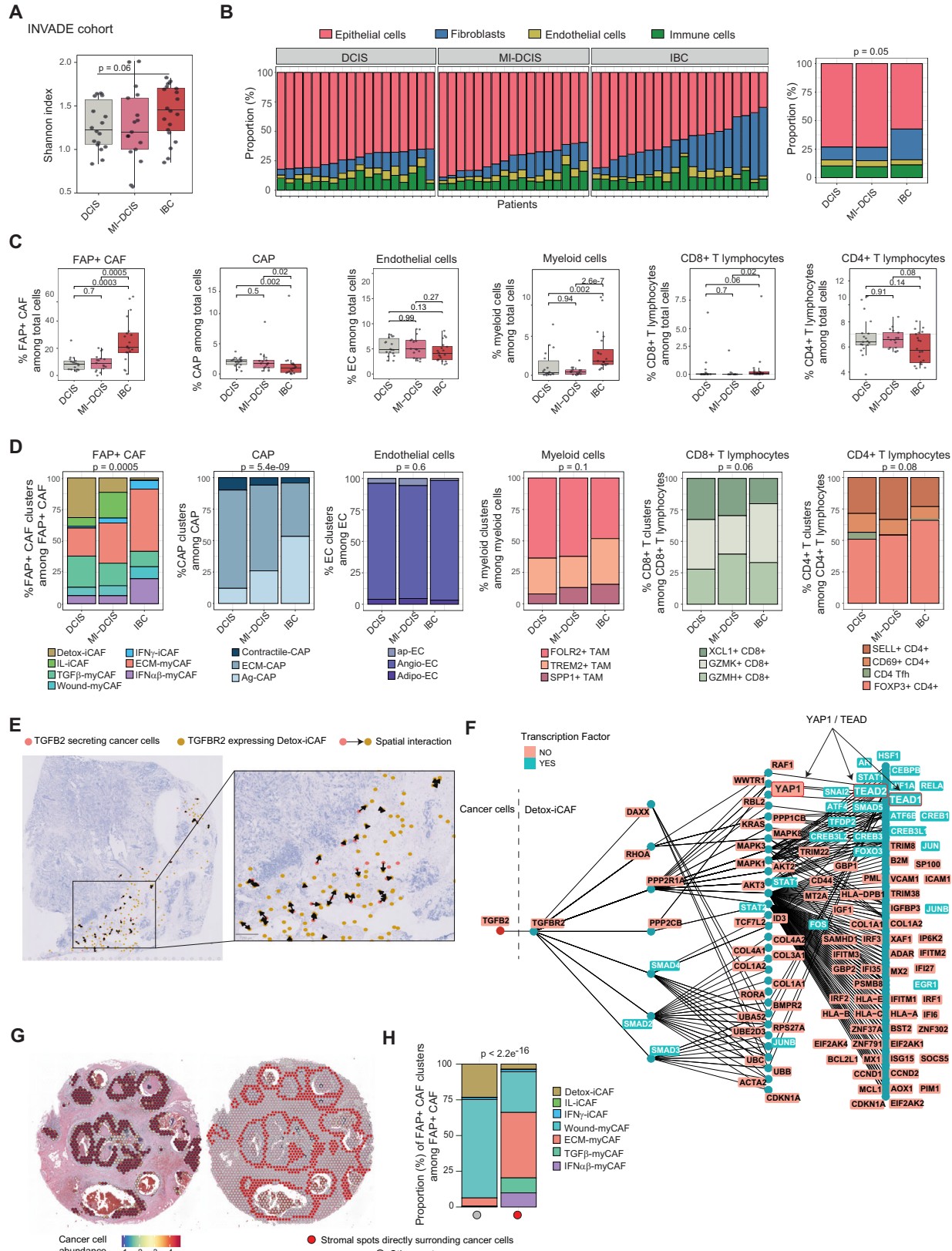

vicinity of cancer cells, while TGFβ-myCAF are associated with immune infiltration and found at the invasive margin, around intratumoral infiltrated lobules or blood vessels. Consistent with our observations, previous work has shown that *Podoplanin*-expressing myofibroblasts (corresponding to ECM-myCAF and IFNαβ-myCAF) are enriched at the interface with cancer cells where non-activated fibroblasts and EC were depleted[80]. Moreover, ECM-myCAF and TGFβ-myCAF colocalize with TREM2+ TAM and actively induce the TREM2+ TAM program to create an immunosuppressive area in the tumor bed. In relation to this spatial organization, ECM-myCAF are detected close to exhausted CD8+ T lymphocytes and immunosuppressive FOXP3+ CD4+ T lymphocytes. Finally, Wound-myCAF are mainly distributed in the intratumoral

**Fig. 6 | BC cellular composition is associated with tumor invasiveness.**
**A**–**D**, **H** Data from the INVADE cohort ($N$ = 55 BC patients), including Ductal Carcinoma in Situ (DCIS) lesions ($N$ = 18), micro-invasive DCIS (MI-DCIS) (i.e. DCIS lesions with invasive foci not exceeding 1 mm ($N$ = 17) and Invasive Breast Cancer (IBC) ($N$ = 20)). **A** Boxplot showing Shannon index from cell type fractions in DCIS, MI-DCIS and IBC samples. $p$ value from Mann–Whitney test. **B** Bar plots showing the relative composition of epithelial, stromal, endothelial and immune cells per patient (Left) and according to BC invasive status (Right). $p$ value from chi-squared test. **C** Boxplots of the relative proportions of FAP+ CAF, CAP, endothelial, myeloid and lymphoid cells in DCIS, MI-DCIS and IBC samples. $p$ values from Mann–Whitney test. **D** Bar plots showing the relative proportions of clusters among FAP+ CAF, CAP, endothelial, myeloid and lymphoid cells in DCIS, MI-DCIS and IBC samples. $p$ values from Fisher's exact test. **E** Communications from cancer cells to Detox-iCAF mediated by TGFβ2-TGFBR2 interaction in space on one section. Colored dots represent cells from a given cell type expressing either the receptor for Detox-iCAF (in yellow) or the ligand for cancer cells (in red). Arrows highlight cells close enough to communicate through the selected ligand-receptor, as inferred by SpaTalk. **F** Network of downstream pathways upregulated in Detox-iCAF following TGFβ2-TGFBR2 interaction in the section shown in (**E**) inferred using the atlas. **G** Left, Cancer cell abundance in each spot of a DCIS section inferred by deconvolution. Right, Manual selection by a pathologist of spots in direct contact with cancer cell-enriched spots (in red); other spots farther away from cancer cells are in gray. **H** Bar plot showing the relative proportions of FAP+ CAF clusters (assessed by deconvolution) in spots directly surrounding cancer cell-enriched spots (red dot) compared to other spots (gray dot) in the DCIS section. $p$ value from Fisher's exact test. In all boxplot the center line, box limits and whiskers indicate the median, upper and lower quartiles and 1.5 × interquartile range. Source data are provided as a Source Data file.

stroma, more distant from cancer cells than any other myCAF subsets. Collectively, the spatial distribution of all FAP+ CAF clusters is compatible with their plasticity and crosstalk with cancer or immune cells, and uncovers FAP+ CAF-immuno-permissive and immunosuppressive ECT (see Model, Fig. 8). However, the absence of normal fibroblasts in the single cell transcriptomic data from BC—potentially due to the limited number of cells that could be sampled—is a limitation in the spatial transcriptomic analysis that must be addressed in future work. This would contribute to a better understanding of the spatial relationship and the effects on the plasticity of Detox-iCAF and normal fibroblast crosstalk.

Recent observations show that BC progression requires both the invasive propensity of DCIS cancer cells and stroma permissiveness[53,77]. Our results on stroma are consistent with these findings but go a step further by demonstrating the role of specific FAP+ CAF clusters in the transition between DCIS and IBC. Indeed, we found that BC invasive states were significantly associated with an accumulation of specific FAP+ CAF clusters. Detox-iCAF are enriched in DCIS but their proportion decreases in MI-DCIS while IL-iCAF levels increase. In contrast, IBC accumulate ECM-myCAF and IFNαβ-myCAF. The low ECM-myCAF content in DCIS might be linked to the spatial architecture of DCIS, which restricts cancer cell localization and could maintain ECM-myCAF strictly at the periphery of the ducts. Interestingly, we uncovered that the levels of Detox-iCAF and TGFβ-myCAF at diagnosis are independent prognostic factors of a progression from DCIS to IBC. We previously showed that T cells can induce the transition from ECM-myCAF into TGFβ-myCAF[10], this could explain why T lymphocytes localized at the ductal basal membrane are, to some extent, predictive of a shorter recurrence in DCIS[91]. Developing treatments which target ECM-myCAF, TGFβ-myCAF and IFNαβ-myCAF and restore them to a Detox-iCAF (or possibly a normal-like) state might be a means to improve patient survival. The importance of alternative therapeutic strategies blocking ECM-myCAF differentiation, rather than depleting them, is highlighted by the fact that they can quickly reappear after ECM-myCAF depletion[42]. Our results highlight a promising avenue for the development of therapeutic strategies. Indeed, our data suggest that a combined therapy targeting both DPP4 and YAP1 in FAP+ CAF might be necessary to fully suppress the emergence of the immunosuppressive ECM-myCAF and TGFβ-myCAF clusters and therefore improve response to immunotherapies in BC patients. This therapeutic approach could also be useful in the context of DCIS tumors with a high-risk of invasive progression upon standard treatment, that we identified as having a depletion in Detox-iCAF and an enrichment in TGFβ-myCAF. Thus, blocking the transition from Detox-iCAF to ECM- and TGFβ-myCAF could particularly benefit these patients by preventing progression. In addition, assessment of the Detox-iCAF and TGFβ-myCAF content in DCIS at the time of diagnosis could help in identifying patients with low-risk DCIS, which is still quite unsatisfactory using standard histo-pathological criteria. Our findings could thus pave the way for more efficient and safer treatment de-escalation in DCIS. Finally, FAP has been identified as an attractive target for CAR-T therapy because of its role in shaping the immunosuppressive TME, a major obstacle in the treatment of solid tumors. By targeting FAP+ CAF, CAR-T cells could disrupt tumor's immunosuppressive niches, ultimately enhancing immunotherapy efficacy. Still, FAP+ CAF are highly heterogenous and part of a complex network within the TME. As we show here, they also contribute to the formation of immuno-protective niches, such as the one composed of FAP+ Detox-iCAF and FOLR2+ TAM. Thus, a singular focus on eliminating FAP+ CAF may inadvertently disrupt these protective mechanisms and hinder the intended immunotherapeutic effect. In conclusion, combining multi-omic data in a spatial context enables us to provide clues on FAP+ CAF identity, and how a given CAF cell state changes based on its neighboring cells. By deconvoluting spatial transcriptomics, we characterized different cell niches composed of specific FAP+ CAF clusters, associated with immuno-protective or immuno-suppressive cells, enriched either in endothelial or cancer cells. Thus, our study provides a comprehensive spatially resolved atlas of FAP+ CAF populations-related architecture in BC.

## Methods
### Cohorts of BC patients: inclusion and ethics
The study developed here is based on samples taken from surgical residues available after histopathologic analyses and not required for diagnosis. There is no interference with clinical practice. Analysis of tumor samples was performed in accordance with the relevant national law and with recognized ethical guidelines (Declaration of Helsinki) on the protection of people taking part in biomedical research. All patients with BC hospitalized at Institut Curie received a welcome booklet explaining that their samples may be used for research purposes. All patients included in our study were thus informed by their referring oncologist that biological samples collected through standard clinical practice could be used for research purposes and they gave their informed consent. In case of patient refusal, which could be either orally expressed or written, residual tumor samples were not included in our study. Human experimental procedures for analyses of tumor microenvironment by F. Mechta-Grigoriou's lab were approved by the Institutional Review Board and Ethics committee of the Institut Curie Hospital group (approval February 12, 2014) and CNIL (Commission Nationale de l'informatique et des Libertés approval no.: 1674356 delivered March 30, 2013). The Biological Resource Centre (BRC) is part of the Pathology Department in the Diagnostic and Theragnostic Medicine Department headed by Dr. A. Vincent-Salomon. BRC is authorized to store and manage human biological samples according to French legislation. The BRC has declared defined sample collections that are continuously incremented as and when patient consent forms are obtained (declaration number: DC-2008-57). The BRC follows all currently required national and international ethical rules, including the Declaration of Helsinki. The BRC has also been accredited with the AFNOR NFS-96-900 quality

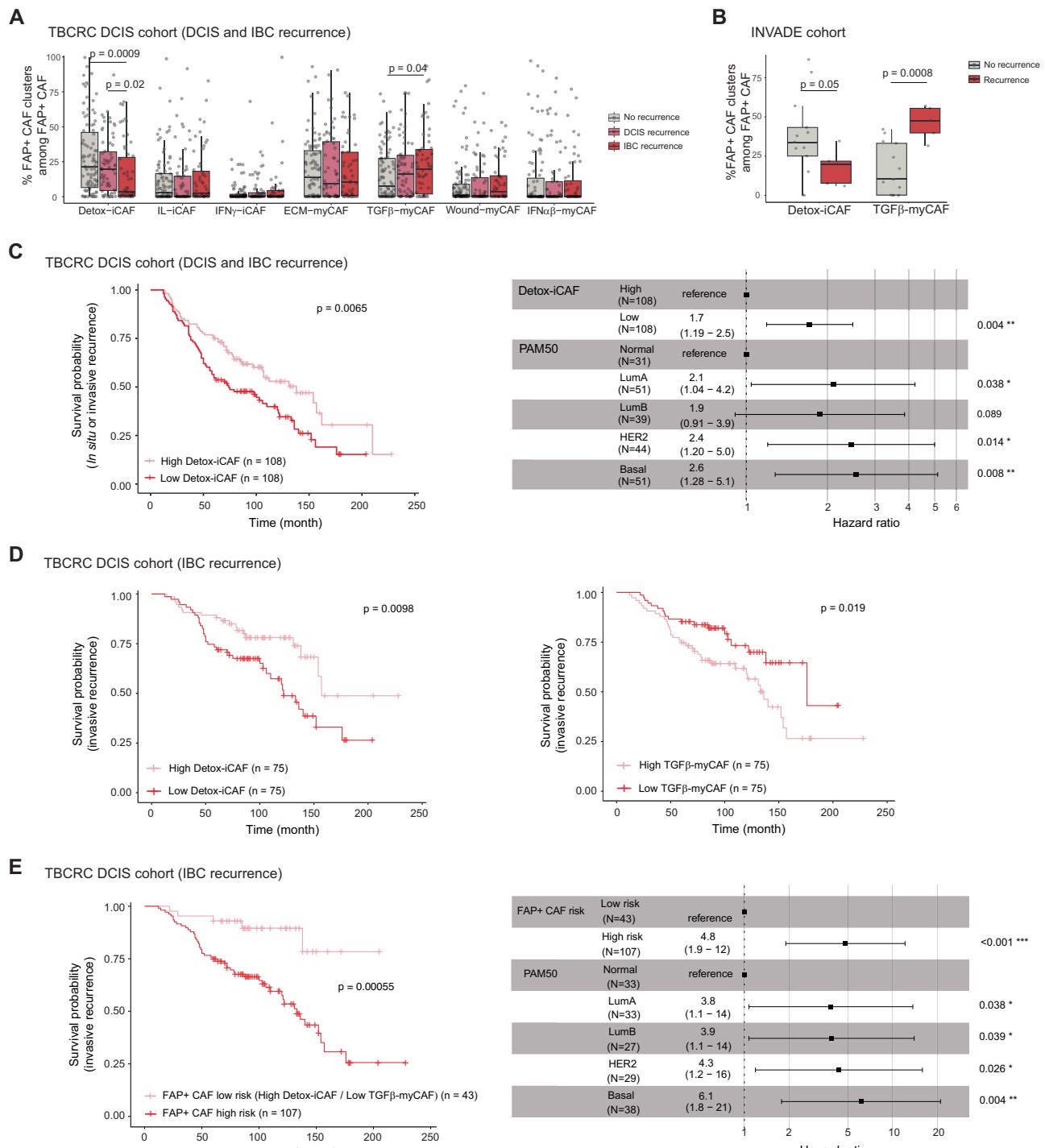

**Fig. 7 | FAP+ CAF composition predicts DCIS progression. A**, **C**–**E** Data from the TBCRC cohort of DCIS patients (*N* = 216 patients). **A** Boxplots of the relative proportions of FAP+ CAF clusters among FAP+ CAF in DCIS patients with DCIS recurrence (*N* = 66), with IBC recurrence (*N* = 55) or without recurrence (*N* = 95). *p* values from two-sided Mann–Whitney test. **B** Relative proportions of Detox-iCAF and TGFβ-myCAF between DCIS with (*N* = 5) or without recurrences (*N* = 13) in the INVADE cohort (*N* = 18). *p* values from two-sided Mann–Whitney test. **C** Left, Kaplan–Meier curve of time to recurrence (DCIS and IBC) stratified by the median of Detox-iCAF content in DCIS at diagnosis. *p* value from Log-rank test. Right, Forest plot for multivariate Cox proportional hazards model considering Detox-iCAF content (median stratification) and PAM50 classification. **D** Kaplan–Meier curves of

time to recurrence (IBC only) stratified by the median of Detox-iCAF (Left) or TGFβ-myCAF (Right) content among FAP+ CAF in DCIS at diagnosis. *p* values from Log-rank test. **E** Left, Kaplan–Meier curve of time to recurrence (IBC only) stratified in low risk patients (defined as high Detox-iCAF and low TGFβ-myCAF content) and high risk patients (other patients). *p* value from Log-rank test. Right, Forest plot for multivariate Cox proportional hazards model including low/high risk patients and PAM50 classification. In all boxplot the center line, box limits and whiskers indicate the median, upper and lower quartiles and 1.5 × interquartile range. In the forest plots, the center points shows the hazard ratio (HR) and lines represent 95% confidence interval (CI). Source data are provided as a Source Data file.

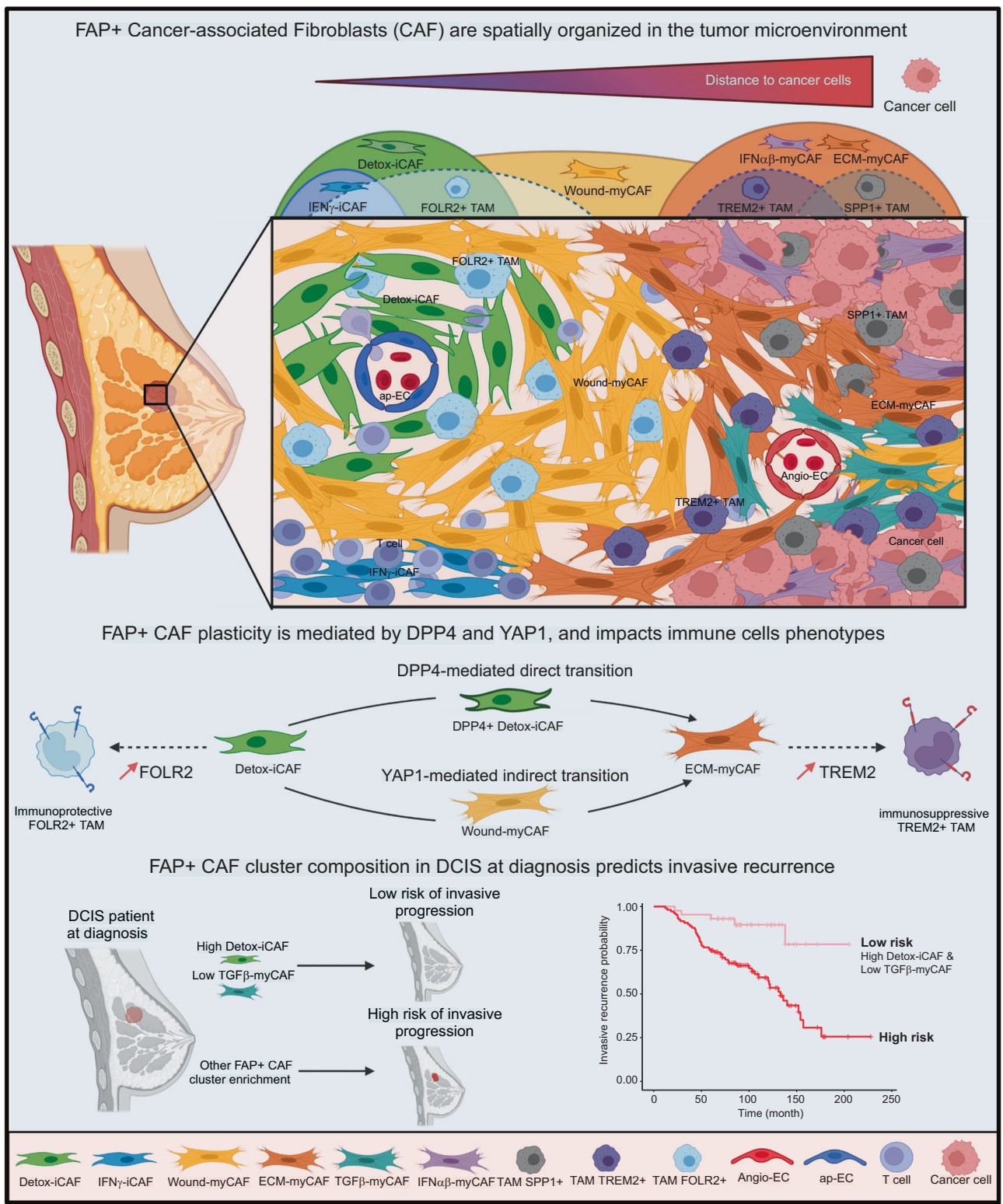

**Fig. 8 | Schematic model.** CAF heterogeneity and plasticity shape a structured organization of the tumor micro-environment in BC. In this paper, we describe spatial organization, plasticity and interactions of FAP+ CAF clusters with neighboring cells by combining analysis of single-cell data, spatial transcriptomics and functional assays. We identify spatially organized cellular EcoCellTypes, which are precisely localized within tumors and composed of specific FAP+ myCAF or iCAF clusters. Distances to cancer cells induce a gradient of FAP+ CAF cluster identities. Detox-iCAF are found around blood vessels composed of ap-EC and in close proximity to FOLR2+ TAM. Detox-iCAF serve as a reservoir and can give rise to ECM-myCAF in presence of cancer cells, either directly or indirectly through the Wound-myCAF cluster, by DPP4-

and YAP1/TEAD-dependent mechanisms. ECM-myCAF localize close to tumor cells, where they can reach a TGFβ-myCAF phenotype in presence of T lymphocytes. In addition, specific TAM are found in different FAP+ CAF cluster-enriched territories. While FOLR2+ TAM are close to Detox-iCAF, TREM2+ and SPP1+ TAM are enriched in ECM-myCAF, IFNαβ-myCAF and TGFβ-myCAF-enriched niches. Our data show that spatial organization in BC tumors is related to reciprocal interactions of FAP+ CAF clusters with cancer and immune cells in specific spatial domains. Importantly, we identify that the content in Detox-iCAF and TGFβ-myCAF at DCIS diagnosis is a predictive factor for the recurrence of DCIS into invasive breast cancer. The figure was created with Biorender.com.

label (renewed and currently valid until 2021). All samples are pseudo-anonymous when they arrive from the BRC in the lab. In addition, the BRC collections have been declared to the CNIL (approval no. 1487390 delivered February 28, 2011).

**BC prospective cohorts.** For this study, luminal (Lum) tumors were defined based on immunohistochemistry (IHC) staining for estrogen and progesterone receptors (ER and PR). The cutoff used to define hormone receptor positivity was 10% of stained cells. Luminal A and Luminal B tumors were distinguished based on proliferation index (Ki67 staining) with Lum A defined as Ki67 < 15% and Lum B with Ki67 > 15%. HER2 positive tumors were identified following ERBB2 immunostaining using American Society of Clinical Oncology guidelines. TN immunophenotype was defined as follows: ER− PR− ERBB2−. The prospective cohort 1 is composed of 84 fresh BC samples prior treatment and includes Lum A ($N = 45$), Lum B ($N = 33$), HER2 ($N = 3$) and TN ($N = 3$). Samples were analyzed by flow cytometry to characterize FAP+ CAF clusters and immune cell subpopulations. The prospective cohort 2 is composed of 16 fresh BC samples prior treatment and includes Lum A ($N = 9$), Lum B ($N = 6$) and HER2 ($N = 1$). Samples were collected and cultured in vitro to isolate primary FAP+ CAF cell lines. The clinical data of these two prospective cohorts are described in Supplementary Table 1.

**INVADE cohort.** The INVADE cohort is a retrospective series of 55 patients, who have been treated at Institut Curie between 1992 and 2014, and underwent surgery for a breast carcinoma prior to any treatment. This cohort includes 18 pure DCIS cases, 17 micro-invasive (MI-DCIS) cases (DCIS lesions with invasive foci of maximum 1 mm) and 20 primary IBC cases. Informed patient consents for the use of tissues for research purposes were collected, and ethical approval from the Institutional Review Board (Institut Curie breast cancer study group) was obtained for the use of all specimens. All samples were anonymized before analysis. Complete clinico-pathological data of the series are described in Supplementary Table 2. Histopathological review of the cases was done in accordance with the current standards by expert breast pathologists. Immunohistochemical evaluation of ER, PR and HER2 status was performed according to the ASCO/CAP recommendations[92], and histomolecular class was determined on the basis of IHC and proliferation index[93]. Clinical data on treatment were obtained from the Institut Curie electronic medical records.

**Differential analysis between fibroblasts from healthy breast tissues and Detox-iCAF from breast cancer**
Differentially expressed genes between fibroblasts isolated from healthy breast tissues[55,56] and Detox-iCAF from breast cancer[10] were obtained by using *FindMarkers* function from Seurat R package. To validate the result obtained and to account for sample size inflation in scRNA-seq data, cell-level counts were also aggregated at sample-level after pseudo-bulk reconstruction using muscat R package[94]. Differentially expressed genes at pseudo-bulk level were then obtained using DESeq2 R package.

**Trajectory inference on scRNA-seq data**
Single-cell trajectories on the human BC FAP+ CAF dataset[10] were computed using PAGA tree as implemented in R package dyno version 0.1.1 (https://github.com/dynverse/dyno) with the function *infer_trajectory* supplying as input the normalized dataset containing the 2000 most variable genes. For the downsampling step, 10 iterations were done using randomly 80% of the total FAP+ CAF dataset. We also used the Elpigraph-based method STREAM version 0.4.1[95]. For STREAM, gene filtering step was done using function *filter_genes* with the parameter min_num_cell set to 5, followed by a variable gene selection step with *select_variable_genes* and parameter loess_frac set to 0.01. The 3-dimension UMAP was recomputed using

*dimension_reduction*. The initial tree structure was seeded with n_cluster set to 10 and the elastic principal graph was obtained by setting epg_alpha to 0.015, epg_mu to 0.2 and epg_lambda to 0.02. Monocle3 trajectory graph (https://cole-trapnell-lab.github.io/monocle3/) was produced using the function learn_graph and plotted using plot_cells with default parameters. Root of the pseudotime was set in the PI16+ Detox-iCAF cluster.

**RNA velocity**
FAP+ CAF velocity analyses were done using Python packages velocyto (http://velocyto.org) and scVelo version 0.2.4 (https://scvelo.readthedocs.io/). As scVelo does not correct for batch effect, this method could not be accurately applied to merged samples. Analysis was therefore conducted on human BC FAP+ CAF scRNA-seq data[10] from the sample showing the highest Shannon index regarding FAP+ CAF cluster representation, as determined by the R function *diversity* from the vegan package. Loom files containing raw spliced and unspliced counts from each bam files were obtained by running the velocyto command-line pipeline with default parameters using the human reference genome GRCh38. scVelo was applied on each Loom files. Data were filtered and normalized using *filter_and_normalize* using the top 2000 most variable genes and first and second moments were computed with 30 principal components (PC) and 30 neighbors. The solution to the full dynamical model was obtained by running both *recover_dynamics* and *velocity* with mode set to dynamical.

**Gene signatures**
YAP1/TEAD-target gene signature is composed of the following genes identified in literature as being up-regulated by YAP1/TEAD: *ABHD2, BCAR4, BDNF, CHST3, CTGF, CYR61, DAB2, FMN2, FRY, GGH, GJA5, ITGB2, LHFP, LIFR, MFAP5, OLR1, PARVA, PDGFRL, PMP22, PRR16, PRSS23, PTGS2, PXDN, RASAL2, SCARA3, SMARCA1, SPARC, SPRED1, STXBP6.*

**Analysis of transcription factor activity**
The VIPER v1.32 and DoRothEA v1.10 R packages were used to perform transcription factor activity analysis[66]. Only Regulons with a high confidence level (A, B and C) were included in the analysis. Regulons enriched in each FAP+ CAF cluster were identified using the Seurat *FindAllMarkers* function. To identify transcription factors specifically implicated in the direct versus indirect transition, we recovered the top transcription factors, which displayed a higher mean score in the indirect transition compared to both the direct transition and the other cells. Similar approach was used for BC[9] and PDAC[42] mouse model datasets with a higher mean score in Wound-myCAF compared to Detox-iCAF and ECM-myCAF.

**Building a reference breast cancer atlas**
**Quality control and processing.** For construction of the BC atlas, we recovered 3 publicly available processed scRNA-seq datasets[35,55,56]. Starting from the available filtered count matrices and initial cell annotations from the authors, we re-processed each cell type individually using the R package Seurat[96]. For data from[55], treatment-naïve samples were retrieved, and low-quality cells were filtered based on the respective QC distribution for each cell type. Each cell type-specific dataset was log-normalized and scaled with default parameters, excepted if indicated otherwise. For cancer cells, fibroblasts, endothelial and mast cells, we kept cells with more than 1000 features detected. For T cells, we kept cells with less than 3000 features and having more than 500 counts detected. For B cells, we kept cells with more than 1000 features and less than 20,000 counts. For Pal et al. dataset[56], cells from healthy pre-menopausal samples with more than 750 features, less than 4000 and a percentage of mitochondrial transcript less than 20% were conserved. Normal fibroblasts were then extracted using Vimentin (VIM) expression. In the Wu et al. dataset[35],

T cells annotated by authors were kept for our analysis. Myeloid cells were processed with a clustering resolution of 0.3, and two clusters characterized by low features counts were excluded. Myeloid cells were scaled on Patient ID and nCount_RNA variables. At this step, all datasets (analyzing each cell type) were separately annotated (see below #"Cell type annotation"). All Seurat object obtained were then merged, log normalized and scaled. A principal component analysis (PCA) was computed on the 2500 most variable genes, and the UMAP was generated on the first 30 components of the PCA.

**Cell type annotation.** Dimensionality reduction and Louvain clustering (resolution 0.3–0.5) followed by either a marker-based annotation or a label transfer[57] were used to annotate the different cell type. Normal epithelial cells and myoepithelial cells were identified among EPCAM+ clusters shared across patients and based on TP63 and KIT expression, respectively, while cancer cell clusters were patient specific due to unique genomic rearrangements, as expected. CNV profiles were established to confirm non-tumoral phenotype of cell types, such as CAF, using InferCNV (see #"Spatial transcriptomics"; "InferCNV on scRNA-seq and spatial transcriptomic data") section. For CAF annotations, we used label transfer from the in-house FAP+ CAF and CAP datasets. Normal fibroblasts were clustered using the Louvain algorithm at a resolution of 0.2 and annotated based on PI16 and dermatopontin (DPT) expression. Similarly, endothelial cells were clustered at a resolution of 0.2 and annotated based on functional enrichment analysis using EnrichR[97]. For annotation of T cells, we integrated the T cells from[35,55,56,98] using FastMNN. Final T cell annotations were obtained by combining label transfer from[98], canonical markers and Louvain clustering at a resolution of 0.7. Only T cells from BC were kept in the atlas for further analysis. Myeloid cells were clustered at resolution of 0.4 and cluster 0 was processed and re-clustered at resolution 0.5 to increase resolution. Finally, the obtained clusters were annotated based on literature[89,99]. B cells, plasma cells and mast cells were annotated based on MS4A1, MZB1 and KIT expression, respectively. For the deconvolution, the cluster of cycling cells was removed to avoid cross-recognition of cycling normal cells with cancer cells. Based on all these annotations, we identified fine-grained fibroblast clusters, including 7 FAP+ CAF clusters: Detox-iCAF, IL-iCAF, IFNγ-iCAF, Wound-myCAF, ECM-myCAF, TGFβ-myCAF, IFNαβ-myCAF; 3 CAP clusters: Contractile-CAP, ECM-CAP, Ag-CAP (Antigen-processing CAP) and 2 normal fibroblast clusters: PI16+ universal fibroblasts and PI16− fibroblasts. We also identified different clusters of myeloid cells, including 3 clusters of APOE+ tumor-associated macrophages (TAM) (TREM2+ TAM; SPP1+ TAM; FOLR2+ TAM) and CXCL10+ Macrophages (Mac), 4 dendritic cell (DC) clusters: Monocyte-derived DC (Mo-DC), CLECL9A+ conventional DC type 1 (CLECL9A+ cDC1), CCR7+ LAMP3+ IL3RA− DC (DCreg), CLEC4C+ IL3RA+ plasmacytoid DC (pDC); one S100A12+ monocyte (S100A12+ Mo) cluster; and one cluster of mast cells (KIT+). Interestingly, S100A12+ Mo and CXCL10+ macrophages were the cell populations expressing the most the gene signature from myeloid-derived suppressor cells[100]. Clusters of T lymphocytes were defined as followed: 3 CD8+ clusters: precursor XCL1+ CD8+ T cells (XCL1+ CD8+), transitional GZMK+ CD8+ T cells (GZMK+ CD8+), differentiated GZMH+ CD8+ T cells (GZMH+ CD8+); 3 CD4+ clusters: memory CD69+ CD4+ T cells (CD69+ CD4+), naïve SELL+ CD4+ T cells (SELL+ CD4+), TNFRSF18+ CD4 T follicular helper (TNFRSF18+ CD4+ Tfh) and FOXP3+ CD4+ Treg. Two NK clusters were also identified: cytotoxic CD16+ GZMB+ NK (CD16+ NK) and immunoregulatory CD16− NKG2A+/KLRC1+ NK (NKG2A+ NKreg). We also identified two clusters of B lymphoid cells: B lymphocytes and plasma cells and three endothelial cell (EC) clusters: ACKR1+ antigen presenting EC (ap-EC), which might correspond to tumor-associated high-endothelial venules based on previous literature[101], CXCL12+ VEGFC+ Angiogenesis-EC (Angio-EC) and CD36+ adipogenesis-related EC (Adipo-EC), as also identified in a recent study[102]. Finally, we detected 3 epithelial clusters:

cancer cells, normal epithelial cells and myoepithelial cells, based on marker expression and InferCNV profiling. To validate that our BC atlas recapitulated all the cell types from the TME previously described, we recovered two independent scRNAseq dataset[13,35], which were not used for the atlas construction (except T cells from[35]). We then used label transfer from our BC atlas as reference to annotate each cells from the two datasets and generated a confusion matrix using the package pheatmap.

## Spatial transcriptomics

**Sample preparation.** Seven frozen BC samples were chosen based on tissue structure and RNA quality (RIN > 8). The "Visium Spatial Tissue Optimization Slide and Reagent Kit" (10X Genomics; #PN-1000193) was then used to optimize permeabilization conditions for BC tissues. Briefly, sections were fixed, stained and then permeabilized at different time points to capture mRNA, and the reverse transcription was performed to generate fluorescently labeled cDNA. The permeabilization time that resulted in the highest fluorescence signal with the lowest background diffusion was chosen. The best permeabilization time for BC tissue was 18 min. Cryostat sections of 10 μm of thickness were cut and placed on Visium Spatial Gene Expression slides (10X Genomics, PN-1000184). The slide was incubated for 1 min at 37 °C, then fixed with methanol for 30 min at −20 °C followed by Hematoxylin and Eosin (H&E) staining and images were taken under a high-resolution microscope. After imaging, the coverslip was detached by holding the slide in water and the slide was mounted in a plastic slide cassette. The spatial gene expression process, including tissue permeabilization, second strand synthesis and cDNA amplification, was performed according to the manufacturer's instructions (10X Genomics; #CG000239). cDNA quality was next assessed using Agilent High sensitivity DNA Kit (Agilent, #5067-4626). The spatial gene libraries were constructed using Visium Spatial Library Construction Kit (10X Genomics, PN-1000184). Processed public spatial transcriptomic data have been recovered from 10X Genomics website (https://support.10xgenomics.com/spatial-gene-expression/datasets) and from previous study[13].

**Spatial data processing.** Using SpaceRanger software v1.2.2 (10x Genomics), spatial raw base call (BCL) files were demultiplexed and mapped to the reference genome GRCh38. Alignment of the slide's barcoded spot patterns and selection of spots in the tissue were performed using Loupe Browser from 10x Genomics. The resulting count matrices were processed in Seurat v4.1.0 for log2 normalization, scaling, and dimension reduction.

**Spatial deconvolution.** Deconvolution of the spatial sections was done using the cell2location package version 0.1[67] implemented in Python3. The BC atlas described above was used to estimate the reference cell type signatures using RegressionModel with categorical_covariate_keys set to patient ID, other parameters were set to default. To avoid any bias during spatial deconvolution of BC tissues only cells from tumor samples and not cells from reduction mammoplasties were conserved for a total of 37 different cell types and states. Luminal breast cancer cells were kept for deconvolution. For the spatial mapping of the different cell types, we supplied each Visium section individually and set the hyperparameters N_cells_per_location to 15 after manual examination of the slides and detection_alpha to 200. We set the number of epoch to 30,000. For visualization and downstream analysis, we discretized the q05_cell_abundance_w_sf matrix from cell2location deconvolution by rounding the values up per cell type, or down for cancer cells, based on microscopely recognizable cell types and observed cell numbers, as assessed by a pathologist. Each inferred cell was plotted using a custom function derived from Seurat SpatialDimPlot, where each cell is represented by a point filled with a color representing the corresponding cell type and a jitter of 40 um within the spatial spot to avoid overplotting.

For deconvolution with SpatialDWLS[69], we used the function *runDWLSDeconv* with default parameters implemented in Giotto. For RCTD[68], deconvolution we first created a RCTD object and deconvoluted the section using the run.RCTD function with doublet mode set to multi. For quantification of cell clusters in pathological compartment, we first pulled all sections together and quantified the number of cells in each compartment independently from the section of origin and performed a fisher test between cellular clusters among the cell types. We then extracted the compartment of interest from the sections and computed the proportions for each section of cell clusters among a particular cell type per compartment. We generated a heatmap displaying the median proportion across the section of the percentage of each cellular cluster. We applied a Wilcoxon test one *versus* all to evaluate the enrichment in a pathological compartment of a particular cellular cluster.

## Niche and ECT identification

To identify groups of similar co-occurring cell types across BC sections analyzed by Visium, we merged the cell2location deconvolution output (q05_cell_abundance_w_sf) for each section, transformed the estimated cell-type proportions into isometric log ratios and created a batch balanced k-NN graph using the *bbknn* function implemented in Scanpy (https://scanpy.readthedocs.io) with default parameters. To identify spots composed of similar cell types, we applied Leiden clustering at a resolution of 0.6 on the resulting graph. We then computed the average cell type composition per cluster (called niches) and we generated a heatmap using the pheatmap R package. A hierarchical clustering on rows (cell types and states) and columns (niches) was finally applied with Euclidean distance and method set to Ward.D2. Heatmap's values were centered and scaled by rows.

**Pathologist annotations and quantifications.** Sections where the high-resolution scan was available were annotated by pathologists using the QuPath software[103] or Loupe Browser software (10X Genomics) and defined 5 main pathological features (Peritumor stroma, Intratumor stroma, Cancer cells, Lymphocyte aggregates and Normal ducts and lobules) based on morphology. The resulting annotations were exported as a GeoJson file, imported in R with the *geojson_read* function from the Geojsonio package, processed with the sf R package and transferred to each Visium spot for deconvolution's quantification in each area. Pathological features were transferred to each spot on a majority area basis, except for cancer cells where the annotation was transferred if more than 30% of the spot covered the tumor epithelial compartment annotation. When the scanned images from public sections were not available or of insufficient quality to be annotated using QuPath, the spots were annotated using the Loupe Browser software.

**InferCNV on scRNA-seq and spatial transcriptomic data.** To discriminate cancer cells from normal cells, copy number variations were inferred using InferCNV R package version 1.13.0. For the scRNA-seq data, normal fibroblasts were used as reference and epithelial cells, FAP+ CAF, CAP were used as query. The analysis was run using cluster_by_groups set to True, analysis_mode set to subclusters, otherwise default parameters were used. For spatial data, the spots annotated as Peritumor were used as reference. The analysis was run using *cluster_by_groups* set to False, *analysis_mode* set to subclusters, otherwise default parameters were used. CNV information was then imported to the Seurat object using the *add_to_seurat* function.

**Spatial distances analysis.** To compute the distribution of distances between each cell type in the different sections, we used the discretized deconvolution cell abundance matrix. For each cell type, we computed the Euclidian distance between each deconvoluted cell compared to all other cells belonging to another cell type by using the spot coordinates. We considered and kept the closest distance for each reference cell.

## Ligand-receptor analysis

**scRNA-seq analysis.** Ligand-Receptor analyses were performed using the R package CellChat version 1.4.0, with the provided CellChatDB database. Overexpressed genes and interactions were computed with identifyOverExpressedGenes and identifyOverExpressedInteractions respectively, with default parameters. Communication probabilities were computed using computeCommunProb with the default trimean method for calculating the average expression. Filtering of the cell-cell communications was done using filterCommunication, and the communication pathway probability was computed using computeCommunProbPathway with default parameters.

**Spatial transcriptomic analysis.** Spatial Ligand-Receptor analyses were performed using SpaTalk v1.0[76]. We used the function *dec_celltype* to transfer the cell2location deconvolution output on the SpaTalk object created using *createSpaTalk* with default parameters. We used the function *set_expected_cell()* with the result of cell2location cell abundance to indicate the number of cell per spot. We then used *find_lr_path* to infer cell-to-cell Ligand-Receptor communications and *dec_cci* to indicate the cell types of interest. Finally, we recovered activation of the downstream pathways using *get_lr_path* and plotted the result using *plot_lr_path* and *plot_lrpair* by indicating the cell implicated using the SpaTalk slot *cellpair*.

## Niche reference mapping

To map the niches on new BC sections, we first created and trained a scVI model with two layers using the deconvolution output of the 17 sections (which allowed us to identify the niches) and the 14 new BC sections collected from[73]. We then converted the model to an scANVI model using scvi.model.SCANVI.from_scvi_model and indicated the niches as the label of interest. We trained the model for 20 epochs and indicated 100 samples per label. We used the *predict* method of the model to transfer the niche labels on new sections, and finally exported the predicted label and plotted them on the sections.

## Analysis of ECT composition in the METABRIC cohort

To analyze ECT enrichments in BC molecular subtypes, we used the BayesPrism algorithm[74] to deconvolute transcriptomic data from the METABRIC cohort. Normalized expression matrix, clinical information and PAM50 subtype classifications were obtained from METABRIC (https://www.cbioportal.org/study/summary?id=brca_metabric). Luminal A ($N = 487$), luminal B ($N = 368$), HER2 ($N = 193$) and Basal-like TN ($N = 186$) BC patients were conserved for the analysis. Raw count matrix of 73,426 high-quality cells from our BC atlas was used as input for prior information. Labels were derived from the annotation of 10 ECT described above. ECT6 and ECT7 were pooled to group all epithelial cell types. Mitochondrial and ribosomal protein coding genes were removed as these genes are expressed at high magnitude and not informative in distinguishing cell types. MALAT1 and genes from chrX and chrY were also removed following indication from BayesPrism's authors. To reduce batch effects and speed up computation, we performed deconvolution only on protein coding genes. Default parameters to control Gibbs sampling and optimization were used. Final estimation of cell type fraction in each bulk RNA-seq sample was recovered using the updated theta matrix and used for downstream analysis. Stratification of tumors was done by applying hierarchical clustering on the matrix of ECT fraction obtained using correlation distance and Ward.D2 method from *pheatmap* R package. Differences in overall survival between the 4 subgroups of BC patients were assessed using Kaplan–Meier analysis and log-rank test statistics using the *survival* and *survminer* R packages.

## Analysis of scRNA-seq data from breast and pancreatic mouse models

We recovered two publicly available scRNA-seq datasets (accession numbers E-MTAB-12036 from EBI Biostudies and GSE149636 from the Gene Expression Omnibus) from BC[9] and PDAC[42] mouse models. To allow for cross-species comparisons, we converted mouse gene symbols to their human counterparts. Human and mouse genes were recovered from Ensembl database GRCh38.p13 and GRCm38.p6, respectively. Seurat v4.3.0 was used for all subsequent analyses. Counts were log-normalized with *NormalizeData* function, then the top 2000 most variable genes were selected using *FindVariableFeatures* function with vst method. Scaling of the data was applied using *ScaleData* function and PCA was performed using *RunPCA*. The cluster annotated as fibroblasts by the authors was isolated for the PDAC dataset[42], and cells annotated as S100A4+ fibroblasts were selected for the BC dataset[9]. Label transfer[57] using the BC atlas as reference was then applied and all cells not identified as fibroblasts using the annotations obtained from the label transfer step were removed. For the PDAC dataset and BC dataset, we finally recovered respectively 35,508 and 3363 FAP+ CAF, respectively. For the quantification of cells in wild-type (WT) and *Lrrc15*-diphteria toxin receptor knock-in mice, the following timepoints and conditions were conserved: WT mice: Skin_WTnaive_Stroma, D0_NoTx_Stroma, D7_DTRneg_Stroma, D14_DTRneg_Stroma, D21_DTRpos_Stroma *Lrrc15*-diphteria toxin receptor knock-in: Skin_WTnaive_Stroma, D0_NoTx_Stroma, D7_DTRpos_Stroma, D14_DTRpos_Stroma, D21_DTRpos_Stroma.

## Bulk RNA-seq from INVADE cohort

Frozen samples were processed for RNA extraction using kit (miRNeasy Mini Kit, Qiagen #217004) following the manufacturer's instructions. RNA integrity and quality were analyzed using Agilent 4200 TapeStation system. The library was prepared following the protocol of the Illumina® TruSeq Stranded mRNA kit according to the supplier's recommendations. Briefly, the key steps of this protocol were successively, starting from 1 μg of total RNA: purification of PolyA (containing mRNA molecules) using magnetic beads attached to poly-T oligonucleotides, fragmentation using divalent cations at high temperature to obtain fragments of ~300 bp, cDNA synthesis, and finally ligation of Illumina adapters and amplification of the cDNA library by PCR. Sequencing was then performed on the Illumina HiSeq2500 sequencer (75-bp paired end). Image analysis and base-calling were performed using Illumina Real Time Analysis (RTA 2.1.3) with default settings. TopHat2 (v2.0.10)[104] was used to align the raw RNAseq data on the human genome (hg19) and on a transcriptome from the refSeq annotations (April 2015 version) with the following parameters: bowtie2 (v2. 1. 0)[105] using the sensitive and fr-firsttrand parameters for the library type (strand specific protocol), allowing up to 2 mismatches in the seed of 25 bp and a gap of up to 10 bp in alignment, an intron size between 30 bp and 700 kbp, with a mean insert size between read pairs of 155 bp with a standard deviation of 80 bp. Raw counts were then calculated by reconstructed transcripts (26,093 genes), using Cufflinks toolkit (v2.2.1)[60], using default parameters and stranded mode. Raw count data are available from the Figshare data repository (DOI: 10.6084/m9.figshare.21591351).

## Deconvolution of bulk RNA-seq data

Fifty-five Bulk RNA-seq from the INVADE cohort and 216 bulk RNA-seq from the TBCRC 038 cohort were deconvolved using BayesPrism algorithm version v2.0[74]. Raw count matrix of 73,426 high-quality cells from our BC atlas was used as input for prior information. Labels were derived from the annotation of 39 cell types and states described above. Mitochondrial and ribosomal protein coding genes were removed as these genes are expressed at high magnitude and not informative in distinguishing cell types. MALAT1 and genes from chrX and chrY were also removed following indication from BayesPrism's authors. To reduce batch effects and speed up computation, we performed deconvolution only on protein coding genes. Default parameters to control Gibbs sampling and optimization were used. Final estimation of cell type fraction in each bulk RNA-seq sample was recovered using the updated theta matrix and used for downstream analysis.

## Flow cytometry analysis of BC samples

All antibodies used in our study have been listed in Supplementary Table 3. Fresh human BC samples were collected directly after macroscopic examination and selection of areas of interest by a pathologist. Tumor samples were stored in $CO_2$-independent medium and transferred to the research institute. All tumor samples were processed without any previous knowledge about CAF and immune cell infiltration. Samples were cut into small pieces (around 1 mm³) and digested in $CO_2$-independent medium (Gibco #18045-054) supplemented with 5% human serum (BioWest #54190-100), 2 mg/ml of collagenase I (Sigma #C0130), 2 mg/ml of hyaluronidase (Sigma #H3506) and 25 mg/ml of Dnase I (Roche #11284932001) during 45 min at 37 °C with permanent shaking (500 rpm). Cells were then filtrated through a 40 μm cell strainer (Fisher Scientific #223635447) and resuspended in PBS+ solution (PBS, Gibco #14190; EDTA 2 mM, Gibco #15575; Human Serum 1%, BioWest #S4190-100). After centrifugation, cells were counted using BeckmanCell Counter and resuspended to a concentration of $5 \times 10^5$ to $1 \times 10^6$. Cells were first incubated with Live/Dead dye (1:1000, BD Horizon™ Fixable Viability Stain 780 dye, BD Biosciences, #565388 for FAP+ CAF clusters and Fixable Violet Dead Cell Stain Kit, Thermo Fisher, #L34955 for TAM subsets) for 10 min at room temperature (RT) to exclude non-viable cells. After a rapid washing with PBS+, cell suspension was then stained for 20 min at RT with an antibody mix specific pour each cell type (as detailed below). Cells were then washed and acquired on the BD LSRFortessa™ analyzer (BD biosciences). For flow cytometry analysis, cells were first gated based on their size (FSC-A) and granularity (SSC-A). Cell types were then analyzed on the Live/Dead negative fraction and defined as epithelial (EPCAM+), hematopoietic (CD45+), endothelial (CD31+) and red blood cells (CD235a+). Specific surface markers are then added to the antibody mix to characterize FAP+ CAF clusters and TAM subsets. For all flow cytometry analysis, at least $5 \times 10^5$ events were recorded for each sample. Compensations were performed using single staining of anti-Mouse IgG and Negative control particle set (BD biosciences, #552843) with each antibody.

**Characterization of FAP+ CAF clusters.** The markers used for the gating strategy identifying FAP+ CAF (CAF-S1) clusters have been based on differentially expressed genes and described in detail in one of our recent publication[10]. These different markers were defined based on pairwise comparisons of expression profiles from FAP+ CAF clusters that helped us to identify specific genes for each cluster. In addition, we considered surface expressed proteins for which commercially antibodies were available allowing both flow cytometry analysis and cell sorting. These markers cannot be used separately as the identification of the clusters relies on the successive combination of different markers, as described below. The identification of FAP+ CAF clusters relies on the combination of these different markers tested successively. In brief, FAP+ CAF were first separated on the basis of the ANTXR1 protein level that distinguished myofibroblastic (myCAF, ANTXR1+) from inflammatory (iCAF, ANTXR1−). ANTXR1+ myCAF clusters were next distinguished according to SDC1, LAMP5, and CD9 protein levels. ANTXR1+ SDC1+ LAMP5− were defined as ECM-myCAF, ANTXR1+ LAMP5+ SDC1+/− as TGFβ-myCAF and ANTXR1+ SDC1− LAMP5− CD9+ as Wound-myCAF. ANTXR1− iCAF clusters were separated using GPC3, DLK1 and CD74 markers. ANTXR1− GPC3+ DLK1+/− were defined as Detox-iCAF; ANTXR1− GPC3− DLK1+ as

IL-iCAF and ANTXR1− GPC3− DLK1− CD74+ as IFNγ-iCAF. To allow this characterization, cells were stained with an antibody cocktail containing anti-EpCAM−BV605 (1:50; BioLegend, # 324224), anti-CD31-PECy7 (1:100, BioLegend, #303118), anti-CD45−BUV395 (1:20, BD Biosciences, #BD-563792), anti-CD235a-PerCP/Cy5.5 (1:50, Biolegend, #349109), anti-CD29-Alexa Fluor 700 (1:100, BioLegend, #303020), anti-FAP-APC (1:100, R&D Systems, #MAB3715), anti-ANTXR1-Alexa Fluor 405 (1:25, Novus Bio, #NB100-56585AF405), anti-SDC1-BUV737 (1:25, BD Biosciences, #612834), anti-LAMP5-PE (1:10, Miltenyi Biotec, #130-109-156), anti-GPC3-Alexa Fluor 549 (R&D systems, 1:20, #FAB2119T), anti-DLK1-Alexa Fluor 488 (R&D systems, 1:20, #FAB1144G), anti-CD9-BV711 (BD Biosciences, 1:100, #743050) and anti-CD74-BV786 (BD Biosciences, 1:100, #743736). All antibodies except FAP were purchased already conjugated with fluorescent dyes. Anti-FAP primary antibody was conjugated with fluorescent dye Zenon APC Mouse IgG1 labeling kit (1:100, Thermo Fisher Scientific, #Z25051). Isotype control antibodies for each CAF marker used were: iso-anti-CD29-Alexa Fluor 700 (Alexa Fluor 700 Mouse IgG1 κ Isotype Ctrl Antibody, 1:100, BioLegend, #400144), iso-anti-FAP-APC (Mouse IgG1 κ Isotype Control, R&D Systems, 1:200, #MAB002), iso-anti-ANTXR1-Alexa Fluor 405 (Mouse IgG1 Alexa Fluor 405-conjugated Antibody, 1:100, R&D systems, #IC002V), iso-anti-SDC1-BUV737 (BUV737 Mouse IgG1 κ Isotype Control; 1:100, BD Biosciences, #612758), iso-anti-LAMP5-PE (PE human IgG1 REA Control Antibody, 1:10, Miltenyi Biotec #130-104-613), iso-anti-GPC3-Alexa Fluor 549 (Mouse IgG2A Alexa Fluor594-conjugated Isotype Control, 1:20, R&D systems, #IC003T), iso-anti-DLK1-Alexa Fluor 488 (Mouse IgG2B Alexa Fluor488-conjugated Isotype Control, 1:20, R&D systems, IC0041G), iso-anti-CD9-BV711 (BV711 Mouse IgG1 κ Isotype Control, 1:100, BD Biosciences, #563044) and iso-anti-CD74-BV786 (BV786 Mouse IgG1 κ Isotype Control, 1:100, BD Biosciences, #563330). The cut-offs for the gating for each marker was defined based on the isotype controls represented in black in the representative FACS plots in Supplementary Fig. 1.

**Characterization of TAM subsets.** Among total CD45+ hematopoietic cells, CD3, CD19 and CD56 markers were used to exclude T lymphocytes (CD3+), B lymphocytes (CD19+) and NK cells (CD56+). TAM subsets were next characterized as CD14+ CD16+ cells and the percentage of TREM2+ and FOLR2+ macrophages was then evaluated. To do so, cell suspension was stained with an antibody mix containing anti-CD45-APCcy7 (1:50, BD Biosciences, #557833), anti-CD3-Alexa Fluor 700 (1:50, BD Biosciences, #557943), anti-CD14-Pecy7 (1:50, BD Biosciences, #557742), anti-CD19-PercPCpy5.5 (1:50, BD Biosciences, #561295), anti-CD16-BV650 (1:50, BD Biosciences, #563692), anti-CD56-BUV395 (1:50, BD Biosciences, #563554), anti-FOLR2-PE (1:50, Biolegend, #391704) and anti-TREM2-APC (primary antibody, 1:50, Novus Biologicals, #MAB17291). All antibodies were purchased already conjugated with fluorescent dyes, except TREM2. Isotype control antibodies for macrophages subsets were used: iso-anti-CD16-BV650 (BV650 Mouse IgG1 κ Isotype Control, 1:50, BD Biosciences, #563231), iso-anti-CD56-BUV395 (BUV395 Mouse IgG2b κ Isotype Control; 1:50, BD Biosciences, #563558), iso-anti-CD14-Pecy7 (PE-Cy7 Mouse IgG2a κ Isotype Control, 1:50, BD Biosciences, #557907), iso-anti-FOLR2-PEC (PE Mouse IgG1 κ Isotype Ctrl Antibody; 1:100, Biolegend, #400112) and iso-anti-TREM2-APC (Rat IgG2B Isotype Control, 1:50, Novus Biologicals, #MAB0061). For TREM2 detection, cells were stained for 20 min at RT with secondary antibody (Rat F(ab)2 IgG APC-conjugated Antibody 1:50, Novus Biologicals, #F0105B). Depending on the size of the tumor samples and the number of cells obtained after digestion, we were able to analyze by flow cytometry the content in both macrophage subsets and CAF-S1 clusters per sample in 25 samples.

**CAP scRNA-seq**
**Isolation of CAP from BC.** CAP were isolated from a total of three primary BC (surgical residues prior to any treatment) by using BDFACS

ARIA III sorter (BD Biosciences). BC were collected directly from the operating room after surgical specimen macroscopic examination and selection of areas of interest by a pathologist. Samples were cut into small pieces (around 1 mm³) and digested in $CO_2$-independent medium (Gibco #18045-054) supplemented with 150 µg/ml liberase (Roche #05401020001) and Dnase I (Roche #11284932001) for 40 min at 37 °C with shaking (180 rpm). After digestion, cells were processed and stained as described above (#"Flow Cytometry analysis of BC samples"). CAP fibroblasts were then gated on the Live/Dead negative fraction and defined as EPCAM− CD45− CD31− CD235a− FAP$^{Med}$ CD29$^{High}$.

**CAP scRNA-seq.** Upon isolation, CAP cells were directly collected into RNase-free tubes (Thermo Fisher Scientific, #AM12450) precoated with DMEM (GE Life Sciences, #SH30243.01) supplemented with 10% FBS (Biosera, #1003/500). Single-cell capture, lysis, and cDNA library construction were performed using Chromium system from 10X Genomics, with the following kits: Chromium Single Cell 3′ Library & Gel Bead Kit v2 kit (10X Genomics, #120237) and Chromium Single Cell A Chip Kits (10X Genomics, #1000009). Generation of gel beads in Emulsion (GEM), barcoding, post GEM-reverse transcription cleanup and cDNA amplification were performed according to the manufacturer's instructions. Cells were loaded accordingly on the Chromium Single cell A chips, and 12 cycles were performed for cDNA amplification. cDNA quality and quantity were checked on Agilent 2100 Bioanalyzer using Agilent High Sensitivity DNA Kit (Agilent, #5067-4626) and library construction followed according to 10X Genomics protocol. Libraries were next run on the Illumina HiSeq (for patients P1) and NovaSeq (for patients P2−3) with a depth of sequencing of 50,000 reads per cell. Processing of raw data, including demultiplexing of raw base call (BCL) files into FASTQ files, alignment, filtering, barcode, and Unique Molecular Identifiers (UMI) counting, were performed using 10X Cell Ranger pipeline version 2.1.1. Reads were aligned to Homo sapiens (human) genome assembly GRCh38 (hg38). Seurat v.3.0.0 was used for log normalization, scaling, dimensionality reduction and clustering using default parameters. Low-quality cells were first filtered out based on the distribution of the unique genes detected (nonzero count). Cells with less than 200 genes detected and more than 4000 genes detected (for patient 1), more than 4500 genes detected (for patients 2) or more than 3500 genes detected (for patient 3) were excluded. Cells with a fraction of mitochondrial genes higher than 5% were discarded. Integration of the samples was done using Seurat functions *FindIntegrationAnchors* and *IntegrateData* with 30 components. Graph-based clustering was applied using FindNeighbours (k = 20) and FindClusters functions (res = 0.4).

**Isolation and culture of primary FAP+ CAF clusters**
Fresh BC samples received after surgery were cut into fragments of ~1 mm³, put either in plastic petri dishes or in petri dishes coated with type I collagen at a final concentration of 9 µg/ml (Institut De Biotechnologie Jacques Boy, #207050357) and cultured in DMEM (Gibco, #41966-029) supplemented with 10% heat inactivated FBS (Biosera, #FB-1003-500) and 1% streptomycin and penicillin (Sigma, #p4333) for 2−3 weeks at 37 °C. Media was renewed every 3 days during an expansion phase of 2−3 weeks. When fibroblasts reached at least 50% of confluency, they were detached using TrypLE (Gibco, #12605-010), centrifuged at 300 × *g* for 5 min and plated in new plastic plates or collagen-coated plates using DMEM supplemented as above. To separate the different FAP+ CAF clusters, cells in both conditions were collected separately and sorted by BDFACS ARIA III using FAP+ CAF cluster-specific surface markers described in ref. 10. For cell sorting strategy, FAP+ CAF cultured on plastic plates were separated based on ANTXR1 and LAMP5 in 2 distinct clusters, ECM-myCAF (CD29+ FAP+ ANTXR1+ LAMP5−) and TGFβ-myCAF (CD29+

FAP+ ANTXR1+ LAMP5+). Cells were stained with an antibody mix containing anti-CD29-Alexa Fluor 700 (1:100, BioLegend, #303020), anti-FAP-APC (1:100, R&D Systems, #MAB3715), anti-ANTXR1-AF405 (1:25, Novus Bio, #NB100-56585AF405) and anti-LAMP5-APC (1:10, Miltenyi Biotec, #130-109-204). We applied similar strategy for cells cultured on collagen-coated plates. FAP+ CAF were sorted in 3 distinct iCAF clusters based on ANTXR1, GPC3 and CD74, defined as followed: Detox-iCAF (CD29+ FAP+ ANTXR1− GPC3+), IL-iCAF (CD29+ FAP+ ANTXR1− GPC3−) and IFNγ-iCAF (CD29+ FAP+ ANTXR1− GPC3− CD74+). To do so, cells were stained with an antibody mix containing anti-CD29-Alexa Fluor 700 (1:100, BioLegend, #303020), anti-FAP-APC (1:100, R&D Systems, #MAB3715), anti-ANTXR1-AF405 (1:25, Novus Bio, #NB100-56585AF405), anti-GPC3-AF700 (1:25, R&D systems, #FAB2119N) and anti-CD74-FITC (1:50, BD Biosciences, #555540). After sorting, cells were expanded in culture at 37 °C in DMEM media supplemented as above, in a humidified 1.5% $O_2$ and 5% $CO_2$ incubator, either on plastic dishes for ECM-myCAF and TGFβ-myCAF or on collagen-coated dishes for Detox-iCAF, IL-iCAF and IFNγ-CAF. To avoid any change in FAP+ CAF cluster identity, fibroblasts were maintained in the same culture condition after sorting. All experiments using FAP+ CAF primary cell lines were not performed beyond passage 10 to avoid fibroblast senescence. Using this protocol, ten different cell lines from ten different patients were isolated and used for functional assays. Verification of the identity of FAP+ CAF cluster cells was determined by flow cytometry using the same antibody mix as detailed above (#"Characterization of FAP+ CAF clusters").

## Characterization of CAF-S1 cluster identity upon culture by RNA sequencing

To validate by RNA sequencing the identity of sorted FAP+ CAF cells, RNAs were extracted using Qiagen miRNeasy Kit (Qiagen, #217004) according to the manufacturer's instructions. RNA extraction was performed at the same timepoint as flow cytometry analyses. Verification of RNA integrity and quality was performed using the Agilent RNA 6000 nano Kit (Agilent Technologies, #5067-1511). cDNA libraries were prepared using the TruSeq Stranded mRNA Kit (Illumina, #20020594) followed by sequencing on NovaSeq (Illumina). Overall quality of raw sequencing data was first checked using FastQC (v0.11.9). Reads were then aligned on a ribosomal RNA database using bowtie (2.4.2) and on the human reference genome (hg38) with STAR (2.7.6a). Additional controls on aligned data were performed to infer strandness (RSeQC 4.0.0), complexity (Preseq 3.1.1), gene-based saturation, read distribution or duplication level using Bioconductor R package DupRadar. The aligned data were then used to generate a final count matrix with all genes and all samples. Only genes with at least one read in at least 5% of all samples were kept for further analyses. Normalization and differential analysis between all FAP+ CAF clusters were conducted with DESeq2 R package. We used Detox-iCAF, IL-iCAF and IFNγ-iCAF gene signatures[10] to determine the enrichment score in Detox-iCAF, IL-iCAF and IFNg-iCAF clusters compared to myCAF clusters. Gene Set Enrichment Analysis (GSEA) software version 3.0 (Broad Institute) was used with the following parameters: Enrichment statistic = " classic", Metric for ranking genes = "Signal2Noise." A heatmap highlighting CAF-S1 cluster marker genes in the bulk RNAseq between sorted CAF-S1 clusters was generated using the R package pheatmap with clustering set to "ward.D2" and distance set to "correlation".

## Functional assays

**Isolation of immune cells.** Primary immune cells were isolated from peripheral blood of healthy donors (with informed consent) obtained from the "Etablissement Français du sang" through an approved convention with the Institut Curie, Paris, France. Briefly, peripheral blood mononuclear cells (PBMC) were isolated using Lymphoprep

(STEMCELL #07861) and immune cell populations were selectively isolated by magnetic cell separation using specific isolation Kits (Human Natural Killer Cell Isolation Kit, Miltenyi #130-092-657; Human CD14 Microbeads, Miltenyi #130-050-201; Human CD4+ CD25+ regulatory T cell Isolation Kit, Miltenyi #130-091-301). For co-culture experiments, $5 \times 10^4$ fibroblasts of each FAP+ CAF cluster were plated on 24-well plates in DMEM supplemented with 10% heat-inactivated FBS and 1% Penicillin streptomycin at 1.5% $O_2$ overnight for complete adherence. The medium was then removed and 500 μl of DMEM supplemented with 1% FBS containing $1 \times 10^5$ NK cells or $2.5 \times 10^5$ CD14+ monocytes or $5 \times 10^5$ CD4+ CD25+ T lymphocytes were added. Co-cultures were maintained for 24 h at 37 °C, 20% $O_2$ for NK cells and monocytes and for 16 h at 37 °C, 20% $O_2$ for T lymphocytes. For flow cytometry analysis, immune cells were harvested after incubation, washed, and stained for 10 min at RT with LIVE/DEAD dye (1:1000, Thermo Fisher, #L34955) to exclude dead cells. Cells were then stained for 30 min at RT with a mix containing specific antibodies for the characterization of immune cell populations. Cells were first gated based on their size (FSC-A) and granularity (SSC-A). TAM and NK populations were analyzed on the Live/Dead negative fraction and defined as myeloid cells (CD45+ CD3− CD19− CD14+), NK (CD45+ CD3− CD19− CD56+ CD16+/−) and T cells (CD45+ CD3+ CD4+). For TAM characterization, total amount of cells was recovered, and monocyte phenotype was assessed by flow cytometry for CD16, TREM2 and FOLR2 protein levels as detailed above ("Flow cytometry analysis of BC samples" #"Characterization of TAM"). CD14+ monocytes were differentiated from FAP+ CAF cells after analysis using CD45 staining. For NK characterization, cells were separated in two subsets according to CD56 and CD16 and defined as cytotoxic NK (CD16[high] CD56[Med]) and noncytotoxic NK (CD16− CD56[high]). The percentage of NKG2A among CD16+ cells as well as granzyme B and perforin levels were also evaluated. To do so, cells were recovered after co-culture and stained with an antibody mix containing anti-CD45-APCcy7 (1:50, BD Biosciences, #557833), anti-CD16-BV650 (1:50, BD Biosciences, #563692), anti-CD56-BUV395 (1:50, BD Biosciences, #563554) and anti-NKG2A-BV786 (1:50, BD Biosciences, #747917). All antibodies were purchased already conjugated with fluorescent dyes. Isotype control antibodies were used: iso-anti-CD16-BV650 (BV650 Mouse IgG1 κ Isotype Control, 1:50, BD Biosciences, #563231), iso-anti-CD56-BUV395 (BUV395 Mouse IgG2b κ Isotype Control, 1:50, BD Biosciences, #563558) and iso-anti-NKG2A-BV786 (BV786 Mouse IgG1 κ Isotype Control, 1:50, BD Biosciences, #563330). After cell surface staining, cells were fixed in paraformaldehyde 4% (PFA) (Electron Microscopy Sciences, #15710) for 15 min at RT. After a washing step with PBS+, cells were incubated with an antibody mix containing anti-Granzyme B PE (1:50, BD Biosciences, #561142) and anti-perforin-AF488 (1:50, BD Biosciences, #563764). Antibodies are suspended in PBS+ solution supplemented with 0.1% of Saponin (Sigma-Aldrich #S7900) and corresponding isotype control antibodies were used: iso-anti-Granzyme B PE (PE Mouse IgG1 κ Isotype Control, 1:50, BD Biosciences, #555749) and iso-anti-Perforin-AF488 (Alexa Fluor 488 Mouse IgG2b κ Isotype Control; 1:50, BD Biosciences, #558716).

For characterization of FOXP3+ regulatory T cells, CD4+ CD25+ T lymphocytes were harvested after co-culture with CAF and stained for 30 min at RT with an antibody mix containing anti-CD45-APCcy7 (1:50, BD Biosciences, #557833), anti-CD3-AF700 (1:50, BD Biosciences, #557943), anti-CD4-APC (1:10, Miltenyi Biotec, #130-113-210), anti-CD25-PE (1:20, Miltenyi Biotec, # 130-113-282), anti-PD-1-BUV737 (1:50, BD Biosciences, #612791) and anti-CTLA4-PEcy5 (1:50, BD Biosciences, #555854). Isotype control antibodies were used: iso-anti-CD25-PE (1:20, Miltenyi Biotec, 130-092-215), iso-anti-PD-1-BUV737 (BUV737 Mouse IgG1, κ Isotype Control; 1:50, BD Biosciences #564299) and iso-anti-CTLA4-PEcy5 (PE-Cy5 Mouse IgG2a, κ Isotype Control; 1:50, BD Biosciences, #555575). FOXP3 staining buffer set kit (eBioscience, #00-5523-00) was used to detect intra-nuclear FOXP3 protein. After fixation

and permeabilization for 1 h at RT, CD4+ CD25+ T lymphocytes were incubated with anti-FOXP3-Alexa Fluor 488 (1:40, Thermo Fisher Scientific, #53-4776-42) for 30 min at RT. The corresponding isotype control (Rat IgG2a kappa Isotype Control (eBR2a), Alexa Fluor 488, 1:200, eBiosciences, #53-4321-80) was also used.

### Transwell migration assay

For CD14+ migration assay, $5 \times 10^4$ FAP+ CAF clusters were plated in the lower chamber of a transwell plate (0.4 µm pore size, Corning HTS Transwell 24 wells #CLS3413) in 200 µl of DMEM supplemented with 10% heat-inactivated FBS and 1% Penicillin streptomycin at 1.5% $O_2$ overnight. After FAP+ CAF cell adherence, CD14+ monocytes ($2.5 \times 10^5$ cells in a volume of 50 µl DMEM supplemented with 1% FBS) were added in the upper chamber and incubated for 6 h at 37 °C 20% $O_2$. After incubation, CD14+ monocytes in the upper and lower chamber were recovered separately. In total, 0.5 µl of 10 µm carboxylated beads (Polyscience #18133) and DAPI (3 µM) were added to each sample before counting. Cell counting was performed by Flow Cytometry using precision beads for normalization and represented as percentage of migration, calculated as the ratio of the cell number in the lower chamber by the total number of monocytes.

### Silencing experiment using small-interference RNA (siRNAs)

For the short interfering RNA (siRNA) experiment, $2.5 \times 10^5$ FAP+ CAF primary fibroblasts of each cluster were plated in a 6 well plate and transfected with 10 nM of siRNA the same day. Transfected cells were incubated in DMEM supplemented with 10% heat inactivated FBS for 72 h at 37 °C and 1.5% $O_2$. siRNA control used was ON-TARGETplus Non-targeting siRNA (Target sequence UGG-UUU-ACA-UGU-UGU-GUG-A, Dharmacon #D-001810-02-05). YAP1 silencing was performed with two distinct siRNAs targeting YAP1 ((*YAP1(1)S* 5′-UGA-GAA-CAA-UGA-CGA-CCA-A-3′ and *YAP1(1)AS* 5′-UUG-GUC-GUC-AUU-GUU-CUC-A-3′ *YAP1(2) S* 5′-CCA-CCA-AGC-UAG-AUA-AAG-A-3′ and *YAP1(2)AS* 5′-CCA-CCA-AGC-UAG-AUA-AAG-A-3′). DPP4 silencing was also achieved using two different siRNA targeting DPP4 (5′-CAC-UCU-AAC-UGA-UUA-CUU-A-3′ and 5′-CAA-GUU-GAG-UAC-CUC-CUU-A-3′, Horizon Discovery, #LQ-004181-00-0005). TGFBRII silencing was performed using two different siRNA (5′-CAA-CAA-CGG-UGC-AGU-CAA-G-3′ and 5′-GAC-GAG-AAC-AUA-ACA-CUA-G-3′). Transfections were performed with DharmaFECT 2 Transfection Reagent (Horizon Discovery, #T-2005-01) according to manufacturer's instructions. Efficient YAP1 and DPP4 silencing were analyzed after 72 h by western blot.

### Protein extraction and western blot

**Protein extraction.** Cells were washed with cold PBS (Gibco #14190) and lyzed with 100 µl of Laemmli buffer (BioRad, #1610737) supplemented with DTT at a final concentration of 50 mM (Thermo Scientific, #11896744). Samples were next heated at 95 °C for 10 min and then sonicated for 10 min (cycles of 30 s ON/30 s OFF) and centrifuged during 10 min at 13,000 × *g* at 4 °C. The protein extract was short-term stored at −80 °C.

**Western blot.** Fifteen µl of proteins were loaded into a NuPAGE Novex 4-12% bis tris mini gels (Thermo Fisher, #NP0321BOX). The migration was performed for 2 h at 120 V in 1X NuPAGE® MOPS SDS Running Buffer (for Bis-Tris Gels only) (Invitrogen, #NP0001) in electrophoresis. The proteins were then transferred to a 0.45 µm nitrocellulose membrane (GE Healthcare #10600002) and incubated overnight at 4 °C with the appropriate primary antibodies: Human DPPIV/CD26 Antibody (1:000, R&D systems #AF1180); Human YAP (D8H1X) Rabbit monoclonal antibody (1:1000, Cell Signaling #14074), Actin (1:10.000; Sigma #A5441), SMAD2 (D43B4) XP Rabbit monoclonal antibody (1:1000, Cell signaling #5339), phosphor-SMAD2 (Ser465/467) Rabbit monoclonal antibody (1:1000, Cell signaling #3108) and TGFβ −Receptor II Rabbit monoclonal antibody (1:1000, Cell signaling

#41896). After several washes and 1 h of incubation using appropriate peroxidase-conjugated secondary antibodies (Jackson ImmunoResearch Laboratories #115-035-003), the proteins were visualized using enhanced chemiluminescence detection (Western Lightning Plus-ECL, PerkinElmer #NEL103E001EA). Analyses of immunoblots were performed using ImageJ software.

### Human breast cancer cell lines

MCF7 (ATCC #HTB-22) and MDA-MB-231 (ATCC #CRM-HTB-26) BC cell lines were cultured in DMEM (Gibco, #41966-029) supplemented with 10% FBS (Biosera, #FB-1003-500) and 1% streptomycin and penicillin (Sigma, #p4333). T47D (ATCC #HTB-133) were cultured in RPMI (Gibco, #11875093) supplemented with 10% FBS and 1% streptomycin and penicillin. MCF10A (ATCC #CRL-10317) were cultured in DMEM/F12 (Gibco, #11320033) media supplemented with 10% FBS and 1% streptomycin and penicillin. All cells were maintained in a humidified 20% $O_2$ and 5% $CO_2$ incubator. The cell identity was verified by using the Short Tandem Repeat (STR) DNA profiling (Promega # B9510) method.

### Co-culture of FAP+ CAF clusters with breast cancer cell lines

For co-culture assays, $7 \times 10^4$ fibroblasts from Detox-iCAF, IL-iCAF or IFNγ-iCAF clusters were plated in 6-well plates and incubated overnight at 37 °C, 1.5% $O_2$ to allow CAF adhesion on the plates. The next day, $3.5 \times 10^4$ MCF7 (Lum BC cells), T47D (Lum BC cells), MDA-MB-231 (TN BC cells) or MCF10 (non-tumoral epithelial cells isolated from breast tissue) were added on CAF culture. Cells were next collected at different timepoints (as indicated on corresponding Figure plots), washed in PBS and stained for 20 min at RT with the same antibody mix as detailed above (#"Characterization of CAF-S1 clusters"), containing additionally anti-EPCAM-BV650 (1:100, Biolegend, #324224) to differentiate CAF from epithelial cells by flow cytometry. Samples were next acquired on the LSRFortessa™ analyzer (BD biosciences) and data were analyzed using FlowJo 10.5.2. software.

### Stimulation of Detox-iCAF with TGFβ2 and MCF-7-derived conditioned medium

In total, $7 \times 10^4$ fibroblasts from Detox-iCAF cluster were plated in 6-well plates and incubated overnight at 37 °C 1.5% $O_2$ to allow cell adherence. The next day, the cells were stimulated with 10 ng/ml of TGFβ2 or conditioned media collected from MCF-7 breast cancer cells and centrifuged for 5 min at 300 × *g* to eliminate debris. Cells were next harvested at different timepoints (t0−72 h) for flow cytometry analysis as detailed above (#"Co-culture of FAP+ CAF clusters with BC cell line").

### Statistical analysis

All statistical analyses and graphical representation of data were performed in the R environment (https://cran.r-project.org, version 4.2.0) or using GraphPad Prism software (version 9.4.1). Statistical tests used are in agreement with data distribution: Normality was first checked using the Shapiro−Wilk test, and parametric or nonparametric two-sided tests were applied according to normality, as indicated in each figure legend. Symbols for significance: ns, non-significant; *<0.05, **<0.01; ***<0.001; ****<0.0001.

### Reporting summary

Further information on research design is available in the Nature Portfolio Reporting Summary linked to this article.

## Data availability

Raw scRNA-seq from CAP, spatial transcriptomic from BC sections and bulk RNA-seq data from the INVADE cohort and from cultured fibroblasts generated in this study are available on European Genome-Phenome Archive platform (https://ega-archive.org) under the controlled accession numbers: EGAS50000000220 and

EGAS50000000219. The controlled access is required as raw data contain identifying patient information. Data access can be granted via the EGA with completion of an institute data transfer agreement. Processed scRNA-seq data are available from the Figshare data repository link: https://doi.org/10.6084/m9.figshare.20348712. Processed spatial transcriptomics data generated in this study are available from the Figshare data repository link: https://doi.org/10.6084/m9.figshare.21591429. Public spatial transcriptomics data used in the study were obtained from the Zenodo data repository https://doi.org/10.5281/zenodo.4739739[13] and from 10x Genomics: Invasive Ductal Carcinoma: https://www.10xgenomics.com/resources/datasets/invasive-ductal-carcinoma-stained-with-fluorescent-cd-3-antibody-1-standard-1-2-0; https://www.10xgenomics.com/resources/datasets/human-breast-cancer-visium-fresh-frozen-whole-transcriptome-1-standard; https://www.10xgenomics.com/resources/datasets/human-breast-cancer-block-a-section-1-1-standard-1-1-0; Ductal Carcinoma In Situ: https://www.10xgenomics.com/resources/datasets/human-breast-cancer-ductal-carcinoma-in-situ-invasive-carcinoma-ffpe-1-standard-1-3-0; Lobular Carcinoma: https://www.10xgenomics.com/resources/datasets/human-breast-cancer-whole-transcriptome-analysis-1-standard-1-2-0. Count data from bulk RNA-seq from DCIS, MI-DCIS and IBC (INVADE cohort) are available from the Figshare data repository link: https://doi.org/10.6084/m9.figshare.21591351. Gene expression data and associated clinical data from TBCRC 038 were retrieved from ref. 77. Processed scRNA-seq datasets from PDAC and BC mice model were recovered from the ArrayExpress repository under the accession number E-MTAB-12036, and from the Gene Expression Omnibus (GEO) accession GSE149636, respectively. Processed scRNA-seq data from ref. 13 were downloaded through the Broad Institute Single Cell portal at https://singlecell.broadinstitute.org/single_cell/study/SCP1039. The scRNA-Seq data of T cells from triple negative breast cancer patients have been recovered from ref. 35. Processed scRNA-seq from ref. 56 were recovered from GEO series GSE161529. The remaining data are available within the article, Supplementary Information or Source data file. Source data are provided with this paper.

## Code availability

Codes used for this study are available on Figshare under the https://doi.org/10.6084/m9.figshare.25092977 and on Zenodo under the https://doi.org/10.5281/zenodo.10809335.

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

## Acknowledgements

We are grateful to Laetitia Fuhrmann, Charlotte Martinat and Andreia Goncalves for collecting clinical data of the patients (SIRIC, INCa-DGOS-4654); Laetitia Lesage and André Nicolas for their help at the experimental pathology platform and Coralie Guerin and Lea Guyonnet at the cytometry core. The ICGex NGS platform was supported by the ANR-10-EQPX-03 (Equipex) and ANR-10-INBS-09-08 (France Genomic Consortium) grants. H.C. is supported by the Foundation for Medical Research (FRM, grant number 13683), and R.M. by the Foundation de France (00119142/WB–2021-36276). G.G. and F.M.-G. are permanent scientists at Inserm. Y.K. is supported by the Institut National du Cancer and INCa (INCa-DGOS-9963; INCa-11692). The experimental work was supported by grants from the Ligue Nationale Contre le Cancer (LNCC, *Labelisation*), Inserm (PC201317), INCa (CaLYS INCa-11692, CAFHeros INCa-16101), ITMO Cancer of Aviesan (2021-2030 cancer control strategy framework, Pre-Caution), SIGN'it 2019 from the Foundation ARC, the European TRANSCAN-3 ERA-NET CHRYSALIS (ARCPARTN-TRANS2022080005422) and Magnolia (INCa-16786), and the ANR as part of the PIA France 2030 with the funding of the CASSIOPEIA RHU (ANR-21-RHUS-0002). F.M.-G. acknowledges the "French Pink Ribbon Association" and the "Simone and Cino del Duca Foundation" for attribution of their respective Grand Prix, as well as the FRM for the Rozen Price and the LNCC for the Duquesne Price. F.M.-G. is very grateful to all her funders for providing support throughout the years.

## Author contributions

F.M.-G. conceived all the project and designed the concept of experiments. H.C. and Y.K. performed all bioinformatic and statistical analyses with the participation of A.M. R.M. performed experiments and acquired data, together with G.G. and C.R. A.V.-S. built cohorts of patients, provided human samples and expertise in pathology analyses, together with L.D. who annotated BC sections. R.L., G.G. and R.M. performed spatial transcriptomics in the experimental pathology platform headed by D.M., in AVS's department. F.P. performed the single cell data from CAP from BC patients. E.R. provided expertise on myeloid cells. S.B. and M.B. performed new generation sequencing. A.P. is project manager and

help in getting funding. F.M.-G. supervised the entire project and wrote the paper with H.C., Y.K. and R.M., with suggestions from all authors.

## Competing interests

F.M.-G. received research support from Innate-Pharma, Roche, Institut Roche and Bristol-Myers-Squibb (BMS). E.R. received grants from BMS, AstraZeneca, Janssen-Cilag and Fonds Amgen France; and travel support from BMS, Hoffmann La Roche, AstraZeneca, Merck Sharp & Dohme. Other authors declare no potential conflict of interest.
