## [Peer Review File · Nature Communications]

Deciphering the spatial landscape and plasticity of immunosuppressive fibroblasts in breast cancerEditorial Note: Parts of this Peer Review File have been redacted as indicated to remove third-party material where no permission to publish could be obtained.

REVIEWER COMMENTS

Reviewer #1 (Remarks to the Author): with expertise in CAFs, omics

This study by Crozier et al investigated the extent and topology of fibroblast plasticity in breast cancer. Overall, the study represents a state-of-the-art approach to addressing an area of research that is highly topical and therefore likely to achieve significant impact. The results presented provide valuable insight into the molecular mechanisms and differentiation trajectories regulating fibroblast heterogeneity in breast cancer. The spatial co-localisation of different CAF subpopulations with other cell types is also investigated highlighting possible roles for specific CAF phenotypes in shaping the immune landscape. The importance of CAF subtypes in regulating the invasive capacity of DCIS/breast cancer lesions is then investigated and CAF subpopulation abundance is shown to have potential utility as a biomarker for the progression of DCIS lesions to invasive breast cancer.

However, I do have some concerns about certain aspects of the manuscript, which should be addressed prior to publication. Specific comments are provided below but the main areas to address include further validation of the mechanisms underlying CAF activation, examining the response to co-culture with additional cancer cell lines/organoids and known myCAF stimuli (e.g. TGF-beta); the use of statistical tests (particularly fisher's exact tests) in certain analyses should be reconsidered to ensure that the conclusions drawn are accurately supported; the description of EcoCellTypes (ECTs) requires further elaboration or consideration, as currently written it is difficult to know how to interpret the statistical validity and biological relevance of these groupings; and finally the abundance and distribution of "healthy tissue" fibroblast subpopulations should be incorporated into the data presented, these cells are largely ignored after the initial figures but would be expected to be an important consideration in the analysis of DCIS progression and spatial distribution.

Specific comments:

Introduction:

In the current manuscript the introduction provides detailed description of previous work from the Mechta-Grigoriou lab. However, the description of other key pieces of literature is relatively cursory. Given the many significant contributions made by this research team it is expected that their prior work is described but this section should be written in a manner that provides a more balanced review of available literature.

The text (in the introduction and throughout the manuscript) would benefit from further proof-reading as some phrasing is unclear or grammatically incorrect.

Results:

FAP+ CAF plasticity revealed by in silico analysis and functional assays:

- A key premise of the initial results presented is that Detox iCAF are distinct from previously described PI16+ Universal fibroblasts. The authors demonstrate this by showing differential expression between these two subpopulations (identifying 3374 significantly differentially expressed genes). It should be confirmed that this large number of DE genes are still statistically significant at a sample (as opposed to single-cell) level, to account for sample size inflation in single-cell data (as described previously e.g. PMID: 33257685).
- In vitro sorted populations are shown to maintain their immunophenotype by flow cytometry. What time point was this analysis performed? This information should be provided in the figure legend, the authors should also elaborate on how long these phenotypes stable for after sorting. Were experiments performed beyond the 72h time-points presented in Panels L-O?
- It should also be confirmed that the in vitro sorted FAP+ CAF subpopulation cultures truly reflect the transcriptomic subpopulations originally identified after being maintained in culture. At a minimum this should be shown by assessing a panel of subpopulation marker genes (independent of those used for cell sorting) by qPCR or ideally using transcriptomic analysis.
- Can the authors comment on why collagen coated plates increase iCAF? Is this due to preferential expansion of specific populations, or different migratory capacity?
- The description of the co-culture method used in Figure 1 L-O and later experiments requires further clarity. How were early time points collected? Presumably the cells wouldn't have adhered within the initial 15 to 30 mins so were fibroblasts plated first

allowed to adhere and then MCF7 or MCF10 plated “on top”? Or are the early time point results from cells collected in suspension/semi-adherence?

- The finding that Detox-iCAF are much more susceptible to myCAF conversion than IL-iCAF and IFN γ -iCAF is really interesting. However, to demonstrate the biological relevance of this finding it should be confirmed that similar results are observed when alternative myCAF stimuli are used. For example, through co-culture with alternative Breast cancer cell lines or organoids and following treatment with recombinant TGF-beta.
- In Line 234 – A “significant” accumulation of Detox-iCAF and Wound-myCAF is described. However, the data only shows a shift in the proportion. This language should be corrected to accurately describe the data shown. Notably, to confirm a significant increase in the proportion of these subpopulations this should be demonstrated across biological replicates (not as a significant p-value from a Fisher’s exact test).

Cancer cells convert Detox-iCAF into ECM-myCAF through a DPP4- and YAP1-dependent mechanism

- In the first paragraph (lines 243-263), the description in this text is quite hard to understand. From the figure legend it is clearer how these pathways were identified for further analysis, but the process used should be described more clearly in the main text.
- In figure 2 H, J and L, applying a Fisher’s exact test is not particularly informative in relation to the conclusions drawn regarding these experiments. The statistical significance of this test demonstrates a difference in the overall proportion of different subpopulations but fails to test the specific hypotheses commented on in the results section. For example,
 - Line 267-271 –Significant differences in Wound-myCAF proportions following siYAP1 or siDPP4 are not assessed with this approach.
 - Line 278 - Is the increase in ECM-myCAF cells upon YAP1 silencing significant?
 - Instead, individual time course analyses for each subpopulation should be performed to enable quantitative and clear description of the changes observed, specifically showing their statistical significance.
 - Can the authors elaborate on rationale to explain the compensatory mechanism of activation described between the indirect and direct activation trajectories?
 - Are the cells somehow more sensitive to activation via the opposing mechanism following silencing of DPP4 or YAP1?

- As described above it is important to demonstrate that this mechanism is consistently involved in response to other myCAF stimuli (co-culture with alternative Breast cancer cell lines or organoids and following treatment with recombinant TGF-beta).
- This will also provide valuable validation and further mechanistic elucidation of findings described later in the manuscript (data presented in Figure 5E-F), suggesting that TGF-beta signalling and juxtacrine signalling is responsible for direct conversion of detox-iCAF to myCAF as opposed to indirect activation via Wound-myCAF.
- In the figure legend there is no description of what time-point the representative western blots (panels G, I and K) represent. What degree of knock down is observed at t0 and at t72h?

Spatial organization of FAP+ CAF clusters in breast cancer

- In the deconvolution of spatial transcriptomic spots there is no description provided for where the “healthy fibroblast” populations are found and how this relates to the Detox-iCAF population. It would be helpful to understand the spatial distribution of these different phenotypes.
- As described above, the use of a fisher exact test in Figure 3D and Figure 3E is not very informative for understanding the subpopulations that are significantly enriched within specific tissue compartments.
- The description of these results should also be modified to more accurately describe the data presented. For example, line 337 states that FAP+ CAF precisely co-localise with other cell types. However, the data shown only confirms that they are found in the same tissue compartment not “precisely co-localise”.

Unsupervised spatial analysis reveals shared regional cellular compositions across patients.

- The analysis performed to generate figure 4B and identify EcoCellTypes, requires extensive clarification from both a technical and biological perspective:
 - I could not find this section of analysis described in the methods section and the description provided in the results omits important details of how the clustering was performed (e.g. distance metrics used, clustering method used). It is also important to demonstrate that identification of these ECTs is reproducible in a validation set.
 - From a biological perspective it is not clear what the ECTs are supposed to represent.

Figure 4A shows identification of niches that represent spots with similar constituent cells. So, what is the rationale for clustering cell-types/states distribution across these niches further? Would it not be more informative to either remove the niches and examine correlations between each cell-type across all spots to measure spatial “co-localisation patterns”. Or assess what cells are associated with particular niches, for example by examining Pearson’s residuals from a Chi squared test.

- In relation to the ECTs, there seems to be discrepancies in the “validation” by flow cytometry (presented in panels D-G). The correlations tested here are inconsistent with the Spatial Transcriptomic data or unclear because of a lack of clarity regarding what the ECTs truly represent. For example, the flow cytometry analysis shows ECM-myCAF positively correlate with TREM2+ TAM and negatively with FOLR2+ TAM. However, the correlation between ECM-MyCAF has not been shown in the Spatial Transcriptomic data. Indeed, figure 4B suggests that all three of these populations are enriched in similar niches (e.g. niche 6). Based on the ECTs it should be expected that the Wound-myCAF and TREM2+ TAM populations positively correlate (as they are both assigned to ECT10) but this data is not shown.

- Can the authors clarify why a subset (n=25) of the BC cohort (n=87) was used in figure 4 F&G and describe how was this subset selected?

- What is the rationale for analysing NK cell activation in panels L-O? Was T-cell activation also assessed? If so, what were the results? For example, previous studies have suggested a role in causing Treg phenotypes however the ST data suggests that this may not be the case given the relative scarcity of Tregs in Niche 6. Do the data from this study no longer support a relationship between myCAF and Tregs?

Specific FAP+ CAF clusters are associated with breast cancer invasiveness

- TGF-B/TGFBR2 axis identified as possible mediator of tumour cell induced ECM-myCAF activation but not as a mechanism of Wound-myCAF activation. As described above, could this be tested in vitro? Additionally, does this suggest that the wound-myCAF phenotype would be induced by paracrine signalling from cancer cells only and could the role of juxtacrine vs paracrine signalling in this process be further elucidated in vitro (e.g. conditioned media)?

Detox-iCAF and TGF β -myCAF proportions predict progression from DCIS to IBC

- As per previous comments, were any differences observed in the normal fibroblast populations? Given that these samples are DCIS presumably healthy fibroblast populations are a major constituent of the stromal compartment in these tissues?

Reviewer #2 (Remarks to the Author): with expertise in single-cell omics, spatial transcriptomics

Comments to the Author

This study utilized trajectory inference, spatial transcriptomics, and functional assays to create an integrative high-resolution map of breast cancer (BC). It provides fresh insights into the spatial organization of the BC tumor microenvironment, with a specific focus on FAP+ CAF diversity, plasticity and interactions with surrounding cells. This is an interesting study, but some of the data interpretation requires further clarification.

1. In Fig. 2D, FOSL1 is the main up-regulated gene in the indirect transition, and FOSL1 is frequently overexpressed in multiple types of human cancers including IBC, with the functions related to cancer invasion and metastasis. Why does the author focus on TEAD1 instead of FOSL1? Does FOSL1 influence the plasticity of FAP+ CAF clusters?

2. Cell2location, SpatialDWLS, and RCTD have been identified as the top-performing methods for the cell type deconvolution of spots in spatial transcriptomics. However, it should be noted that the results obtained from these three methods may exhibit some degree of variation. In this study, the authors utilized Cell2location for their analysis. It raises the question of whether the main findings and conclusions reached by the authors can be replicated using the other two methods.

3. Ten EcoCellTypes (ECT) were identified. Do these ECT exhibit any differences between TN and LUM?

4. Myeloid-derived suppressor cells are the major immunosuppressive cells found in BC, but they are not detected in the comprehensive BC cellular landscape (Fig. 3B and Supplementary Fig. 3C). Please clarify the reason.

5. The relationship between the transition of FAP+ CAF clusters, their spatial location, and function requires further discussion and elucidation.

Minor comments:

1. CAF fibroblasts should be written as CAF because the existing abbreviation "CAF" already stands for cancer-associated fibroblasts. Some abbreviations are not defined, such as PDAC.

2. Please use consistent colors for the different subtypes of CAF in Fig. 1D, as in the other panels.

3. Supplementary Fig. 1I lacks legend description.

4. What is the corresponding method to the result "upon TGFBR2 stimulation, Detox-iCAF increased YAP1 activity (Fig. 5F)"?

Reviewer #3 (Remarks to the Author): with expertise in CAFs, omics

Summary

Crozier et al build on prior data from their group concerning FAP expressing CAFs. The highlight of this work is the effort to untangle spatial relationships of CAF subtypes in solid tumors, which remains a challenging topic to approach and of great importance as the world moves towards potential clinical targeting of CAFs. While the topic is of great interest at this time, enthusiasm is dampened by the limited integration of the CAF subtypes characterized here with similar subtypes delineated across the recent literature. The conclusions to be drawn from the clinical correlations are overstated. Finally, the in vitro work, which makes up the bulk of the functional validation for the story (key to understanding the value of this work towards translational science), is based on a relatively few number of markers. Particularly the FACS plot gating seems inconsistent and is difficult to follow.

Major Comments

1. This paper focuses on an FAP expressing CAF subtype termed “Detox-iCAF”. While gene groups characterized as “detoxification” pathways may well be highlighted by the gene expression patterns shown by this subgroup, the focus of this work is far from detoxification. I would recommend selecting a name more closely aligned with the specific fibroblast/CAF behaviors of this subset highlighted in the work.
2. The labeling in figure 1 A-C is very confusing and makes the comparisons difficult to follow – within these three panels, there are labels for “universal fibroblasts”, “fibroblasts”, “fibroblasts (healthy tissue)”, “normal fibroblasts” as seemingly separate entities. The identity of and rationale for selection of the “normal fibroblasts” used in panel 1C in particular deserves specific attention, presumably this is simply a comparison of activated and non-activated fibroblasts more generally, rather than specific to so called “Detox-iCAFs”? One would expect myCAFs (characterized by α SMA expression) to show a similar appearance based on the markers highlighted? As such, this panel adds little to the story.
3. I would wonder if there are genetic animal models by which some of the proposed in silico lineage relationships might be demonstrated in vivo beyond the addition of published scRNA-seq datasets (1P, Q)? Do the authors believe these represent committed lineages? If so, could this be validated? If not, a greater discussion of ongoing plasticity would add significant nuance to the paper.
4. It should be made clear to the reader on in vitro panels precisely which protein markers are being used to delineate each of the subtypes. Furthermore, given that the identity of the CAF subpopulations are largely based on the upregulated expression of a single marker (or in some cases 2 markers), it would make the manuscript significantly more readable if each sub-type were simply named after their protein marker since these protein markers are relied on for validation. Eg. Detox-iCAF = GPC3+ CAFs. This seems supported by S3-C. Furthermore, a discussion beyond the previously published work about how these particular markers were selected for this paper would be helpful. Presumably there are many potential protein candidates based on the gene expression data?
5. Building on the prior comment, it would benefit the reader significantly to have some specific gene panels displayed of functionally relevant genes related to fibroblast activities to validate the selection of the number of clusters used here. Panel 1G is of the few panels that displays individual genes for specific clusters.

6. What does LRRC15 expression look like in human or mouse breast cancer data? The introduction of this marker, which in other closely related work is used to define a specific subpopulation, should come with a placement of its relevance in the largely body of the presented work (or a discussion of its exclusion).
7. Supplemental S1 – the FACS plots should be made large enough to be able to see the axes. The gating strategy is difficult to follow, particularly in S1D – I have some concern about the ANTXR antibody. Based on markers used, IL-iCAF does not appear to have independent identity. Similarly, the identity of TGFB-myCAF as separate from ECM-myCAF is not convincing. The distinction between ECM-myCAF and IFNab-myCAF is further unclear based on S3E plot. Perhaps the additional of more standard markers of myCAFs would be helpful to clarify. Similar value cut-offs for gating should be used between cell types: In many circumstances, it seems that 10^4 is used as the cut off, but in other circumstances, it seems that 10^3 is used.
8. Any antigen presenting features among the CAFs here?
9. The CAF markers used by Friedman et al., Nat Cancer 2020 should be discussed.
10. How does DPP4 expression play into the originally presented CAF subpopulations? Can this be used as a more general marker? Similarly how does Yap expression play out based on the data presented in figure 1?
11. The proposed role of YAP1 is interesting; however, it should be considered in the context that YAP1 likely plays a strong role in myCAFs.
12. The language used in relation to Figure 6 comes through a little too strongly given the level of extrapolation to draw the correlations provided. While the results are of interest, the language could be toned down – eg. sentence on lines 520-521 as well as 521-524.
13. Discussion of the current clinical work exploring FAP as a CAR-T target for solid tumors is warranted in this manuscript

Minor Comments

1. The first sentence of the abstract is too strong – This behavior of FAP+ CAFs is proposed, but there are certainly variations between cancer types and some assay dependence.
2. The selection of the name “EcoCellTypes” is unclear – would recommend something more specific to the cell types of interest

3. Line 187 should likely read “in” rather than “on”
4. Would consider an additional control for 1L – does the ECM increase with time even without cancer cells?

Reviewer #4 (Remarks to the Author): with expertise in breast cancer, omics

In the manuscript entitled « Comprehensive spatial landscape and plasticity of immunosuppressive fibroblasts in breast cancer », Croizer et al combined scRNAseq and spatial transcriptomics to characterize the phenotype and function of FAP+ CAFs subsets. The authors identified 10 spatially distinct co-localization patterns of cluster of immune cells and CAFs and showed that immunosuppressive ECM-myCAF shift monocytes towards TREM2 macrophages in vitro, and cluster with SPP1+ TAM in situ, while immunoprotective Detox-iCAF shift monocytes towards FOLR2+ macrophages in vitro, and cluster with FOLR2+ TAMs in situ.

Overall, the study is very insightful and will bring new and important knowledge on TME niches within breast cancer, and beyond. The experimental plan is well-designed, the data are robust and main findings were validated in multiple datasets, with mechanistical experiments in vitro. In my opinion, the main findings of the current study is the identification of two clusters of paired subsets of fibroblast/macrophage, one immunosuppressive and one immunoprotective, especially that the authors showed mechanistically how distinct FAP+ CAF subset induce different macrophage phenotype. Further, recent papers extensively described these immunosuppressive TREM2 or SPP1 macrophage (Park & Merad, Nat Immunology 2023 ; Ruben & Pittet, Science 2023) Thus, I would suggest to edit the manuscript by focusing on these two pairs

Major comments:

1. It is unclear why the authors investigate the effect of FAP+ subset on outcome of DCIS since the main work was performed on invasive breast carcinoma. If the authors did not observe a prognostic effect of FAP+ subset on breast cancer outcome, I would suggest to investigate the prognostic value of ECMmyCAF/TREM2 TAM ratio and Detox-iCAF/FOLR2+ TAM in breast cancer publically available RNAseq datasets (all-comers), breast cancer subset (TNBC vs luminal and within luminal lobular vs ductal), or biopsies of TNBC pCR vs residual disease (I would expect that residual disease would be enriched in ECMmyCAF/TREM2)

Minor comments

1. I found the abstract not easy to read. I would suggest that the authors remove the DPP4 and YAP1 mechanisms from the abstract, and rather focus on the two main niches: immunosuppressive ECM-myCAF/TREM2 macrophages and the immunoprotective Detox-iCAF/FOLR2 macrophages
2. Similarly, I would suggest to edit the title and mention macrophages/fibroblast pairs
3. It would be nice to add histology images, H&E and IHC to illustrate these distinct niches from the morphological point of view

Point by point response to Reviewers' comments

We first would like to thank the four Reviewers for their positive evaluation of our work, and for considering our paper suitable for publication in *Nature Communications*, pending modifications. We have now included in the new version of the text the modifications in apparent for addressing all their concerns. All these modifications improved our manuscript, and we thank all of you for your suggestions.

For better visualization of the discussion below, initial comments of the Reviewers are indicated in italic and our answers in normal style. We also highlighted below the text now added in the new version of the manuscript in blue.

Reviewer #1 (Remarks to the Author): with expertise in CAFs, omics

This study by Crozier et al investigated the extent and topology of fibroblast plasticity in breast cancer. Overall, the study represents a state-of-the-art approach to addressing an area of research that is highly topical and therefore likely to achieve significant impact. The results presented provide valuable insight into the molecular mechanisms and differentiation trajectories regulating fibroblast heterogeneity in breast cancer. The spatial co-localisation of different CAF subpopulations with other cell types is also investigated highlighting possible roles for specific CAF phenotypes in shaping the immune landscape. The importance of CAF subtypes in regulating the invasive capacity of DCIS/breast cancer lesions is then investigated and CAF subpopulation abundance is shown to have potential utility as a biomarker for the progression of DCIS lesions to invasive breast cancer.

However, I do have some concerns about certain aspects of the manuscript, which should be addressed prior to publication. Specific comments are provided below but the main areas to address include further validation of the mechanisms underlying CAF activation, examining the response to co-culture with additional cancer cell lines/organoids and known myCAF stimuli (e.g. TGF-beta); the use of statistical tests (particularly fisher's exact tests) in certain analyses should be reconsidered to ensure that the conclusions drawn are accurately supported; the description of EcoCellTypes (ECTs) requires further elaboration or consideration, as currently written it is difficult to know how to interpret the statistical validity and biological relevance of these groupings; and finally the abundance and distribution of "healthy tissue" fibroblast subpopulations should be incorporated into the data presented, these cells are largely ignored after the initial figures but would be expected to be an important consideration in the analysis of DCIS progression and spatial distribution.

We thank the Reviewer for the positive assessment on our manuscript, and for considering the interest of the use of state-of-the-art approaches to decipher FAP+ CAF cluster phenotypes, localization and impact on invasion in breast cancer. We are really grateful for this positive appreciation of our work.

As detailed below, we have now addressed the points raised by the Reviewer. Indeed, we have included additive functional assays in order to validate the mechanisms underlying CAF activation, in particular by testing additional cancer cells and TGF β stimuli. We also used more accurate statistical tests in certain analyses, as recommended. Furthermore, we describe ECT in a more detailed manner and provided additional data to underline their biological relevance in breast cancer. We thank the Reviewer for the careful reading of our manuscript, which help us in improving it.

Specific comments:

Introduction:

In the current manuscript the introduction provides detailed description of previous work from the Mechta-Grigoriou lab. However, the description of other key pieces of literature is relatively cursory. Given the many significant contributions made by this research team it is expected

that their prior work is described but this section should be written in a manner that provides a more balanced review of available literature.

The text (in the introduction and throughout the manuscript) would benefit from further proof-reading as some phrasing is unclear or grammatically incorrect.

As requested, we have added references for CAF populations in the **Introduction (p3-5)** and **Discussion (p19-23)**. We have also done our best to correct typos and grammatical errors thanks to English-speaking authors. We thank the Reviewer for the careful reading of our manuscript.

Results:

FAP+ CAF plasticity revealed by in silico analysis and functional assays:

• A key premise of the initial results presented is that Detox iCAF are distinct from previously described PI16+ Universal fibroblasts. The authors demonstrate this by showing differential expression between these two subpopulations (identifying 3374 significantly differentially expressed genes). It should be confirmed that this large number of DE genes are still statistically significant at a sample (as opposed to single-cell) level, to account for sample size inflation in single-cell data (as described previously e.g. PMID: 33257685).

As suggested by the Reviewer, in order to account for sample size inflation in single-cell data, we have conducted a differential gene expression analysis between normal fibroblasts and Detox-iCAF at pseudo-bulk level. To do so, we aggregated cell-level counts into sample counts using Muscat's tool described in (Crowell *et al.*; *Nature Communication*, 2020). Differential analysis was performed at this pseudo-bulk level using DESeq2. Interestingly, these analyses revealed a significant overlap between the genes identified as differentially expressed by single-cell RNAseq analysis and at pseudo-bulk level. More specifically, among the 2038 genes identified as up-regulated in Detox-iCAF at scRNA-Seq level, 82.8% (1688 genes, including FAP) remained significantly up-regulated at pseudo-bulk level. This new analysis confirmed the validity of the differential genes detected at single-cell level and improved the quality of the analysis performed in the first version of our manuscript. We thank the Reviewer for this suggestion. These novel data are now described in the new version of the manuscript in **Sup Fig. 1C, p6**, its corresponding legend and in the **Methods p34**.

Text added in the Results' section, p6

To account for sample size inflation in scRNA-seq data, we confirmed this result at sample level after pseudo-bulk reconstruction (**Supplementary Fig. 1C**, see also Methods, **#Differential analysis between fibroblasts from healthy tissue and Detox-iCAF**).

Text added in the Methods, p34

Differential analysis between fibroblasts from healthy breast tissues and Detox-iCAF from breast cancer

Differentially expressed genes between fibroblasts isolated from healthy breast tissues^{55, 56} and Detox-iCAF from breast cancer¹⁰ were obtained by using *FindMarkers* function from Seurat R package. To validate the result obtained and to account for sample size inflation in scRNA-seq data, cell-level counts were also aggregated at sample-level after pseudo-bulk reconstruction using muscat R package⁹². Differentially expressed genes at pseudo-bulk level were then obtained using DESeq2 R package.

Text added in the legend of the Sup Fig. 1C, p59

(C) FAP expression in Detox-iCAF and in normal fibroblast after pseudo-bulk reconstruction. P-value from Wald test.

• In vitro sorted populations are shown to maintain their immunophenotype by flow cytometry. What time point was this analysis performed? This information should be provided in the figure

legend, the authors should also elaborate on how long these phenotypes stable for after sorting. Were experiments performed beyond the 72h time-points presented in Panels L-O?

We thank the reviewer for the careful reading of our manuscript, and we apologize for the lack of information in the initial version of the text. The immunophenotypes of the sorted CAF populations were assessed 1 week after sorting from patients. This timing corresponds to the time needed for the fibroblast populations to adapt and proliferate in culture. These CAF populations were maintained for 2-3 weeks using the same culture conditions as those used to isolate the cells. All experiments described in our paper have been performed until passage 6 after sorting, to avoid any risk of cellular senescence. As recommended, we have now added these information in the **Legend** of the **Supplementary Fig. 1G, p59-60**.

In addition, for the co-culture experiments between cancer cells and CAF (**Fig. 2L-O**), as requested by the Reviewer, we have now analyzed the CAF phenotype at a longer time point, e.g. 8 days following the co-culture with cancer cells. We confirmed that the ECM-myCAF cluster continues to accumulate over time, at the expense of the Wound-myCAF and Detox-iCAF clusters, as observed at earlier timepoints. These results are now included in **Fig. 1L** and described in the corresponding legend, **p53**.

Text added in the Legend of Supplementary Fig. 1G, p59-60

(G) Gating strategy used to characterize sorted primary FAP⁺ CAF from plastic dishes for ECM-myCAF and TGF β -myCAF and from collagen-coated dishes for Detox-iCAF, IL-iCAF and IFN γ -iCAF. This analysis was performed 1 week after cell sorting to allow cell adaptation and proliferation in the aforementioned conditions. All experiments were performed until passage 6 after sorting to avoid cell senescence. FAP⁺ CAF cells are gated on FAP⁺ CD29⁺ fibroblasts. Among FAP⁺ CAF cells, ANTXR1 staining is used to distinguish inflammatory iCAF (ANTXR1⁻) from myofibroblastic myCAF (ANTXR1⁺). Among ANTXR1⁺ myCAF, SDC1⁺ LAMP5⁻ were identified as ECM-myCAF, SDC1⁺LAMP5⁺ as TGF β -myCAF and SDC1⁻ LAMP5⁻ CD9⁺ as Wound-myCAF. Among ANTXR1⁻, DLK1, GPC3 and CD74 protein levels were used to identify Detox-iCAF (DLK1⁺ GPC3⁺), IL-iCAF (DLK1⁺ GPC3⁻) and IFN γ -iCAF (DLK1⁻ GPC3⁻ CD74⁺).

Text of the Legend of Fig. 1L, p53

(L) Bar plots showing the fraction of each FAP⁺ CAF cluster among FAP⁺ CAF after co-culture of the Detox-iCAF cluster with the luminal MCF7 BC cell line. Percentages are identified by flow cytometry analyses using FAP⁺ CAF cluster-specific markers. Timepoints below each bar plot indicate the duration of the co-culture. Data are mean \pm SEM (n = 3).

*• It should also be confirmed that the *in vitro* sorted FAP⁺ CAF subpopulation cultures truly reflect the transcriptomic subpopulations originally identified after being maintained in culture. At a minimum this should be shown by assessing a panel of subpopulation marker genes (independent of those used for cell sorting) by qPCR or ideally using transcriptomic analysis.*

As requested, we have now performed bulk RNA sequencing from each CAF population after cell sorting from BC patient samples and following *in vitro* culture at the same time points as those used for flow cytometry analyses. Interestingly, we confirmed that expression of genes of each specific CAF-S1 cluster signature (Detox-, IL-, IFN γ -iCAF identified from patients as shown in Kieffer *et al*, *Cancer Discovery*, 2020) are enriched in each isolated CAF-S1 cluster. We thus validated that each CAF-S1 cluster isolated *in vitro* exhibits the expected gene signature. These data confirm that the *in vitro* sorted CAF-S1 clusters reflect the transcriptomic subpopulations originally identified in patients. They have been added in the **New Supplementary Fig. 1I** and described in the **text p7**, with the corresponding Methods (**#Characterization of CAF-S1 cluster identity upon culture by RNA sequencing**), **p48-49**.

Text added in the Results' section, p7

We also performed bulk RNA sequencing from each CAF population and validated that the transcriptomic profiles of the *in vitro* sorted CAF-S1 clusters are similar to those of the subpopulations originally identified in patients¹⁰ (**Supplementary Fig. 1I**).

Text added in the Legend of Supplementary Fig. 1I, p60

(I) Left, Gene Set Enrichment Analysis (GSEA) plots showing the enrichment score (ES) for Detox-iCAF, IL-iCAF and IFN γ -iCAF gene signatures¹⁰ in sorted Detox-iCAF, sorted IL-iCAF and sorted IFN γ -iCAF clusters, respectively, compared to sorted ECM-myCAF cluster amplified on plastic dishes. **Right**, Heatmap showing differentially expressed genes between sorted CAF-S1 clusters cultured *in vitro*.

Text added in the Methods' section, p48-49

Characterization of CAF-S1 cluster identity upon culture by RNA sequencing

To validate by RNA sequencing the identity of sorted FAP+ CAF cells, RNAs were extracted using Qiagen miRNeasy Kit (Qiagen, #217004) according to the manufacturer's instructions. RNA extraction was performed at the same timepoint as flow cytometry analyses. Verification of RNA integrity and quality was performed using the Agilent RNA 6000 nano Kit (Agilent Technologies, #5067-1511). cDNA libraries were prepared using the TruSeq Stranded mRNA Kit (Illumina, #20020594) followed by sequencing on NovaSeq (Illumina). Overall quality of raw sequencing data was first checked using FastQC (v0.11.9). Reads were then aligned on a ribosomal RNA database using bowtie (2.4.2) and on the human reference genome (hg38) with STAR (2.7.6a). Additional controls on aligned data were performed to infer strandness (RSeQC 4.0.0), complexity (Preseq 3.1.1), gene-based saturation, read distribution or duplication level using Bioconductor R package DupRadar. The aligned data were then used to generate a final count matrix with all genes and all samples. Only genes with at least one read in at least 5% of all samples were kept for further analyses. Normalization and differential analysis between all FAP+ CAF clusters were conducted with DESeq2 R package. We used Detox-iCAF, IL-iCAF and IFN γ -iCAF gene signatures¹⁰ to determine the enrichment score in Detox-iCAF, IL-iCAF and IFN γ -iCAF clusters compared to myCAF clusters. Gene Set Enrichment Analysis (GSEA) software version 3.0 (Broad Institute) was used with the following parameters: Enrichment statistic = "classic", Metric for ranking genes = "Signal2Noise." A heatmap highlighting CAF-S1 cluster marker genes in the bulk RNAseq between sorted CAF-S1 clusters was generated using the R package pheatmap with clustering set to 'ward.D2' and distance set to 'correlation'.

• *Can the authors comment on why collagen coated plates increase iCAF? Is this due to preferential expansion of specific populations, or different migratory capacity?*

It has been shown that 3-dimensional culture of mouse PDAC models can promote conversion between iCAF and myCAF (Biffi, *Cancer Discovery*, 2019). In line with this, we observed that collagen-coated plates conditions increase the proportion of iCAF, suggesting that these conditions promote their attachment to the plates and are compatible with their proliferation. As we found that YAP1 is a key player in the indirect transition from Detox-iCAF to Wound-myCAF and subsequently to ECM-myCAF, we can hypothesize that the slight reduction of the stiffness in collagen-coated dishes might favor iCAF maintenance in culture. As requested, we have now added a comment discussing this point in the new version of the Text, **p20**.

Text added in the Discussion, p20

In line with previous studies showing that 3-dimensional culture of mouse PDAC models can promote conversion between iCAF and myCAF⁵, we observed that collagen-coated plates increase the proportion of iCAF. As we found that YAP1 is a key player in the indirect transition from Detox-iCAF to Wound-myCAF and subsequently to ECM-myCAF, we can hypothesize that the slight reduction of the stiffness in collagen-coated dishes might favor iCAF maintenance in culture.

• *The description of the co-culture method used in Figure 1 L-O and later experiments requires further clarity. How were early time points collected? Presumably the cells wouldn't have adhered within the initial 15 to 30 mins so were fibroblasts plated first allowed to adhere and then MCF7 or MCF10 plated "on top"? Or are the early time point results from cells collected in suspension/semi-adherence?*

We are sorry for the lack of clarity of the protocol used in the first version of our manuscript. As requested, we have now included these precisions in the description of the co-culture experiment in the Method section, **p52**.

Text added in the Methods' section, p52

For co-culture assays, 7×10^4 fibroblasts from Detox-iCAF, IL-iCAF or IFN γ -iCAF clusters were plated in 6- well plates and incubated overnight at 37°C, 1.5% O₂ to allow CAF adhesion on the plates. The next day, 3.5×10^4 MCF7 (Lum BC cells), T47D (Lum BC cells), MDA-MB-231 (TN BC cells) or MCF10 (non-tumoral epithelial cells isolated from breast tissue) were added on CAF culture. Cells were next collected at different timepoints (as indicated on corresponding Figure plots), washed in PBS and stained for 20 min at RT with the same antibody mix as detailed above (#Characterization of CAF-S1 clusters), containing additionally anti-EPCAM-BV650 (1:100, Biolegend, #324224) to differentiate CAF from epithelial cells by flow cytometry. Samples were next acquired on the LSRFortessa™ analyzer (BD biosciences) and data were analyzed using FlowJo 10.5.2. software.

• *The finding that Detox-iCAF are much more susceptible to myCAF conversion than IL-iCAF and IFN γ -iCAF is really interesting. However, to demonstrate the biological relevance of this finding it should be confirmed that similar results are observed when alternative myCAF stimuli are used. For example, through co-culture with alternative Breast cancer cell lines or organoids and following treatment with recombinant TGF-beta.*

We thank the Reviewer for these suggestions, which confirm our initial statements and improve the quality of our manuscript. As suggested, we have now tested the impact of two alternative breast cancer (BC) cell lines (e.g. the luminal BC cell line T47D and the triple negative BC cell line MDA-MB231), and one non-tumoral epithelial cell line isolated from breast tissues (MCF10A) on Detox-iCAF plasticity. By this way, we confirmed that Detox-iCAF can give rise to ECM-myCAF in presence of these BC cell lines, but not with normal epithelial cells. These results are added now in the **New Fig. 1M-O** and described **p8**.

In addition, as suggested by the Reviewer, we have also analyzed the impact of TGF β 2/TGFBR2 axis activation on the transition from Detox-iCAF to ECM-myCAF. To do so, we stimulated Detox-iCAF fibroblasts with TGF β 2 ligand and analyzed their phenotype at different timepoints after stimulation (following similar kinetic as this one tested after BC cell co-culture). We first validated activation of the TGFBR2 pathway by analyzing SMAD2 phosphorylation and next observed that stimulation of Detox-iCAF with TGF β 2 is sufficient to promote the transition from Detox-iCAF to ECM-myCAF. Moreover, consistent with these observations, we observed that silencing of *TGFBR2* prevented the transition from Detox-iCAF toward ECM-myCAF in presence of cancer cells, confirming that TGFBR2 is necessary for this transition. Taken together, these results highlight the role of the TGF β 2/TGFBR2-dependent pathway in the emergence of ECM-myCAF from Detox-iCAF. These results are added now in the **New Fig. 2Q, R** and described **p10**.

Text added in the Results' section, p8 (for T47D and MDA-MB231)

Moreover, co-culture of Detox-iCAF with two alternative breast cancer cell lines (T47D and MDA-MB-231) confirmed transitions from Detox-iCAF toward ECM-myCAF in presence of cancer cells (**Fig. 1M, N** and **Supplementary Fig. 2B, C**). Importantly, maintenance of Detox-iCAF alone (**Supplementary Fig. 2D**) or co-culture with non-tumoral breast epithelial cells

(MCF10A) (**Fig. 10** and **Supplementary Fig. 2E**) did not induce Wound-myCAF or ECM-myCAF, showing that the presence of cancer cells is required to induce these clusters.

Text added in the Results' section, p10 (for TGF β data)

Based on recent data showing that the TGF β 2/TGFBR2 axis drives LRRC15+ ECM-myCAF differentiation in PDAC mouse models⁴², we wondered if this pathway could be involved in CAF-S1 cluster plasticity. We first observed that TGF β 2 stimulation, validated by SMAD family member 2 (SMAD2) phosphorylation, gradually increased the ECM-myCAF content (**Supplementary Fig. 2K-M**), suggesting that TGF β 2 is sufficient to promote the transition from Detox-iCAF to ECM-myCAF. Reciprocally, silencing of TGFBR2 in Detox-iCAF prevented the transition from Detox-iCAF to ECM-myCAF in the presence of cancer cells (**Fig. 2Q, R**), confirming that TGFBR2 is necessary for this transition. Collectively, these data highlight the role of the TGF β 2/TGFBR2-dependent pathway in the emergence of ECM-myCAF from Detox-iCAF.

• *In Line 234 – A “significant” accumulation of Detox-iCAF and Wound-myCAF is described. However, the data only shows a shift in the proportion. This language should be corrected to accurately describe the data shown. Notably, to confirm a significant increase in the proportion of these subpopulations this should be demonstrated across biological replicates (not as a significant p-value from a Fisher’s exact test).*

We do agree with the reviewer that our initial wording was a bit strong. As we could not have access to the sample metadata in public mouse datasets we collected, we modified the main Text **p8**, as suggested by the Reviewer.

Text added in the Results' section, p8

In agreement with the trajectories, selective ablation of ECM-myCAF led to a shift in the proportions of CAF-S1 clusters in favor of Detox-iCAF and Wound-myCAF clusters (**Fig. 1S, Right**).

Cancer cells convert Detox-iCAF into ECM-myCAF through a DPP4- and YAP1-dependent mechanism

• *In the first paragraph (lines 243-263), the description in this text is quite hard to understand. From the figure legend it is clearer how these pathways were identified for further analysis, but the process used should be described more clearly in the main text.*

We thank the reviewer for this feedback on the first paragraph. We have now rephrased this section and added how we identified these pathways in the main text, **p9**.

Text added in the Results' section, p9

.....Using this approach, we identified *DPP4* (Dipeptidyl Peptidase 4) as the main up-regulated gene in FAP+ CAF in the direct transition between Detox-iCAF and ECM-myCAF (**Fig. 2A, B**). In the two scRNA-seq datasets from TNBC and PDAC mouse models, *DPP4* expression was also specifically upregulated in Detox-iCAF but progressively declined in the Wound-myCAF and ECM-myCAF clusters (**Fig. 2C**). Previous studies in other pathologies have demonstrated that inhibition of DPP4 can lead to reduced pathological fibrosis, which suggests that DPP4 may play a crucial role in myofibroblast formation^{64, 65}, and highlight the pertinence of DPP4 in the direct transition from Detox-iCAF to ECM-myCAF. Regarding the indirect transition, we identified 7 YAP1-TEAD (Yes-Associated Protein - Transcriptional Enhanced Associate Domain)-target genes among the top-50 up-regulated genes in the indirect pathway via the Wound-myCAF (**Supplementary Fig. 2J**). Moreover, by computing transcription factor activity scores using Dorothea⁶⁶, we confirmed that TEAD activity was specifically increased in FAP+ CAF undergoing the indirect transition in human BC (**Fig. 2D, E**), as well as in the Wound-myCAF in TNBC and PDAC mouse models (**Fig. 2F**).

- In figure 2 H, J and L, applying a fisher's exact test is not particularly informative in relation to the conclusions drawn regarding these experiments. The statistical significance of this test demonstrates a difference in the overall proportion of different subpopulations but fails to test the specific hypotheses commented on in the results section. For example,
- Line 267-271 –Significant differences in Wound-myCAF proportions following siYAP1 or siDPP4 are not assessed with this approach

We thank the Reviewer for the careful reading of our manuscript. For better clarity of the evolution of each individual CAF-S1 cluster in time, we have now computed individual time course and performed statistical tests comparing siCtrl with siDPP4 or siYAP1 for each population at each time point, as requested. These new data are now included into the **New Fig. 2H, K, N and R**, and described all along **p9-10**.

- Line 278 - Is the increase in ECM-myCAF cells upon YAP1 silencing significant?

Thanks to the statistical analyses described above (**New Fig. 2H, K, N and R**), we observed a transient significant increase in the ECM-myCAF cluster upon YAP1 silencing from 6h to 24h. This can be explained by the fact that inhibiting YAP1-dependent indirect transition (from Detox-iCAF to Wound-myCAF) does not prevent the DPP4-dependent direct transition from Detox-iCAF toward ECM-myCAF. Consistent with this, YAP1 inhibition strongly reduces Wound-myCAF content.

- Instead, individual time course analyses for each subpopulation should be performed to enable quantitative and clear description of the changes observed, specifically showing their statistical significance.

As requested, we have now performed individual time course statistical analysis, which are now shown in the **New Fig. 2H, K, N and Q**, and described all along **p9-10**.

- Can the authors elaborate on rationale to explain the compensatory mechanism of activation described between the indirect and direct activation trajectories?
- Are the cells somehow more sensitive to activation via the opposing mechanism following silencing of DPP4 or YAP1?

We thank the reviewer for this comment. We have now tried to better describe the DPP4-dependent transient trajectory from Detox-iCAF to ECM-myCAF and the YAP1-dependent indirect transition from Detox-iCAF to Wound-myCAF then to ECM-myCAF. As requested, we validated the existence of a compensatory mechanism between these two trajectories, e.g. the YAP1-dependent indirect transition when DPP4 is silenced, and the DPP4-dependent direct transition when YAP1 is silenced. In line with this, we first observed that DPP4-silencing (inhibition of the direct transition) favors Wound-myCAF accumulation (indirect transition) at short time points, and that YAP1 silencing (inhibition of the indirect transition) favors ECM-myCAF accumulation (through direct trajectory). In agreement with this observation, we found that YAP1 protein level remains unchanged upon DPP4 inactivation and that DPP4 protein level is not affected by YAP1 inactivation throughout the kinetics of co-culture with cancer cells. Finally, the concomitant silencing of both DPP4 and YAP1 in Detox-iCAF completely abrogated the appearance of Wound-myCAF and ECM-myCAF following co-culture with cancer cells, thereby demonstrating the key role of both DPP4 and YAP1 in ECM-myCAF accumulation. Taken together, these data confirm the existence of two distinct trajectories giving rise to ECM-myCAF from Detox-iCAF, the DPP4-dependent direct transition and the YAP1-dependent indirect transition through Wound-myCAF. These new data are now shown in the **New Fig. 2** and described **p10**.

Text added in the Result section, p10

Therefore, we tested the hypothesis of a compensatory mechanism between the two trajectories. Specifically, we investigated whether *DPP4* silencing caused an increase in YAP1-dependent indirect transition, and conversely, whether *YAP1* silencing leads to increased *DPP4*-dependent direct transition. Interestingly, inactivation of both *DPP4* and *YAP1* blocked both the direct transition from Detox-iCAF to ECM-myCAF and the indirect path through the Wound-myCAF cluster (**Fig. 2M-O**). Moreover, consistent with this compensatory mechanism, YAP1 protein level remained unchanged upon *DPP4* inactivation and *DPP4* protein level was not affected by *YAP1* inactivation throughout the kinetics of co-culture with cancer cells (**Fig. 2P**), suggesting that ECM-myCAF can be generated by two different trajectories and that one can replace the other when one is inactivated.

- *As described above it is important to demonstrate that this mechanism is consistently involved in response to other myCAF stimuli (co-culture with alternative Breast cancer cell lines or organoids and following treatment with recombinant TGF-beta).*
- *This will also provide valuable validation and further mechanistic elucidation of findings described later in the manuscript (data presented in Figure 5E-F), suggesting that TGF-beta signalling and juxtacrine signalling is responsible for direct conversion of detox-iCAF to myCAF as opposed to indirect activation via Wound-myCAF.*

As mentioned above, we agree with the Reviewer and we have now tested the impact of two alternative breast cancer (BC) cell lines (the luminal BC cell line T47D and the triple negative BC cell line MDA-MB231), and one non-tumoral epithelial cell line isolated from breast tissues (MCF10A) on Detox-iCAF plasticity. By this way, we confirmed that Detox-iCAF can give rise to ECM-myCAF in presence of these BC cell lines, but not with normal epithelial cells. These results are added now in the **New Fig. 1M-O** and described **p8**.

In addition, as suggested by the Reviewer, we have also analyzed the impact of TGF β 2/TGFBR2 axis activation on the transition from Detox-iCAF to ECM-myCAF. To do so, we stimulated Detox-iCAF fibroblasts with TGF β 2 ligand and analyzed their phenotype at different timepoints after stimulation (following similar kinetic as this one tested after BC cell co-culture). We first validated activation of the TGFBR2 pathway by analyzing SMAD2 phosphorylation and next observed that stimulation of Detox-iCAF with TGF β 2 is sufficient to promote the transition from Detox-iCAF to ECM-myCAF. Moreover, consistent with these observations, we observed that silencing of *TGFBR2* prevented the transition from Detox-iCAF toward ECM-myCAF in presence of cancer cells, confirming that TGFBR2 is necessary for this transition. Taken together, these results highlight the role of the TGF β 2/TGFBR2-dependent pathway in the emergence of ECM-myCAF from Detox-iCAF. These results are added now in the **New Fig. 2Q, R** and described **p10**.

Text added in the Results' section, p8 (for T47D and MDA-MB231)

Moreover, co-culture of Detox-iCAF with two alternative breast cancer cell lines (T47D and MDA-MB-231) confirmed transitions from Detox-iCAF toward ECM-myCAF in presence of cancer cells (**Fig. 1M, N** and **Supplementary Fig. 2B, C**).

Text added in the Results' section, p10 (for TGF β data)

Based on recent data showing that the TGF β 2/TGFBR2 axis drives LRR15+ ECM-myCAF differentiation in PDAC mouse models⁴², we wondered if this pathway could be involved in CAF-S1 cluster plasticity. We first observed that TGF β 2 stimulation, validated by SMAD family member 2 (SMAD2) phosphorylation, gradually increased the ECM-myCAF content (**Supplementary Fig. 2K-M**), suggesting that TGF β 2 is sufficient to promote the transition from Detox-iCAF to ECM-myCAF. Reciprocally, silencing of TGFBR2 in Detox-iCAF prevented the transition from Detox-iCAF to ECM-myCAF in the presence of cancer cells (**Fig. 2Q, R**), confirming that TGFBR2 is necessary for this transition. Collectively, these data highlight the

role of the TGF β 2/TGFBR2-dependent pathway in the emergence of ECM-myCAF from Detox-iCAF.

• *In the figure legend there is no description of what time-point the representative western blots (panels G, I and K) represent. What degree of knock down is observed at t0 and at t72h?*

We apologize for the lack of information in the initial version of our paper and we thank the reviewer for the careful reading of our manuscript. The representative western blots (shown **Fig. 2G, I, K**) correspond to the beginning of the co-culture, just before adding the cancer cells (t0). This information is now provided in the **New legend of the Fig. 2G, J, M, p54**.

As suggested by the Reviewer, we have also now verified the efficiency of the silencing at different timepoints of the co-culture with cancer cells (T0, 24h and 72h). By this way, we demonstrated that DPP4 and YAP1 were efficiently silenced throughout the kinetics of the co-culture with cancer cells. These new results are now included in the **New Fig. 2P, p55**.

Text added in the Legend of Fig. 2G, J and M, p54

(G) Representative western blot showing DPP4 protein level in primary Detox-iCAF transfected either with siCtrl or with 2 specific siRNA targeting DPP4 (siDPP4#1 and siDPP4#2). The timepoint corresponds to the beginning of the co-culture (t0)..... **(J)** Representative western blot showing YAP1 protein level in primary Detox-iCAF transfected either with siCtrl or with 2 specific siRNA targeting YAP1 (siYAP1#1 and siYAP1#2). The timepoint corresponds to the beginning of the co-culture (t0)..... **(M)** Same as in **(G)** showing silencing of both DPP4 and YAP1 in primary Detox-iCAF at the beginning of the co-culture.

Text added in the Legend of Fig. 2P, p55

(P) Representative western blots showing DPP4 and YAP1 protein levels in primary Detox-iCAF transfected either with siCtrl or with 2 specific siRNA targeting YAP1 (siYAP1#1 and siYAP1#2) or with 2 specific siRNA targeting DPP4 (siDPP4#1 and siDPP4#2). Three timepoints (t0, 24h and 72h) upon the co-culture of Detox-iCAF with MCF7 cancer cells are shown. Actin is used as an internal control for protein loading.

Spatial organization of FAP+ CAF clusters in breast cancer

• *In the deconvolution of spatial transcriptomic spots there is no description provided for where the “healthy fibroblast” populations are found and how this relates to the Detox-iCAF population. It would be helpful to understand the spatial distribution of these different phenotypes.*

We do agree with the Reviewer that localizing normal fibroblasts in BC spatial transcriptomic data would have been of great interest. However, we did not identify healthy fibroblasts in any of the scRNAseq BC dataset we analyzed upon BC sample dissociation. This could be explained by several factors, including sparsity of non-activated fibroblasts in tumor tissues, limited resolution and size of capture area in spatial transcriptomic data. Still, as cell2location deconvolution was shown to be robust in the case of missing cell types from the reference⁶⁷, we choose to focus on cells specifically isolated from BC. We thus did not include reduction mammaplasties in our reference to avoid any bias during spatial deconvolution of BC tissues. As requested, we have now described this point in a more precise manner in the **Methods’** section **p38** and discussed it more in the **Discussion, p21**.

Text added in the Methods’ section, p38

The BC atlas described above was used to estimate the reference cell type signatures using *RegressionModel* with *categorical_covariate_keys* set to patient ID, other parameters were set to default. To avoid any bias during spatial deconvolution of BC tissues only cells from tumor samples and not cells from reduction mammaplasties were conserved for a total of 37 different cell types and states.

Text added in the Discussion, p21

Collectively, the spatial distribution of all FAP+ CAF clusters is compatible with their plasticity and crosstalk with cancer or immune cells, and uncovers FAP+ CAF-immuno-permissive and immunosuppressive ECT (See Model, **Fig. 8**). However, the absence of normal fibroblasts in the single cell transcriptomic data from BC -potentially due to the limited number of cells that could be sampled- is a limitation in the spatial transcriptomic analysis that must be addressed in future work. This would contribute to a better understanding of the spatial relationship and the effects on the plasticity of Detox-iCAF and normal fibroblast crosstalk.

- *As described above, the use of a fisher exact test in Figure 3D and Figure 3E is not very informative for understanding the subpopulations that are significantly enriched within specific tissue compartments.*

We agree with the Reviewer that the Fisher Exact test only highlights a shift in cell type composition between the different pathological compartments. To address the specific enrichment of the different cell types within these compartments, we have now performed heatmaps showing the median proportion of each cell cluster for each pathological compartment across the different samples. To more accurately address the specific enrichment of a cell type within a compartment, we applied a Wilcoxon test comparing one cell type versus the others within each compartment and we showed the p-values in the heatmaps. We thank the Reviewer as we have now gained more insight in the localization of the different cell populations. We have now integrated these new data in the **New Fig. 3D** and modified the Text accordingly, all along the **Results' section p11-12**, in the **Legend of the Figure 3, p56** and in the **Methods' section, p35**.

Text added in the Legend of the Figure 3, p56

(D) Heatmap of the median proportion of each cell state among the corresponding cell type within each pathological compartment (Tumor cell-enriched areas; Intratumor stroma; Peritumor stroma; Lymphocyte aggregates and Normal ducts and lobules), as shown in **(A)**. * corresponds to a significant ($P < 0.05$) enrichment (red and orange) or depletion (yellow) (% indicated on scale bars) of a cell state compared to the others within a particular pathological annotation. P-values from Wilcoxon (one *versus* all) rank sum test.

Text added in the Methods' section, p35

For quantification of cell clusters in pathological compartment, we first pulled all sections together and quantified the number of cells in each compartment independently from the section of origin and performed a fisher test between cellular clusters among the cell types. We then extracted the compartment of interest from the sections and computed the proportions for each section of cell clusters among a particular cell type per compartment. We generated a heatmap displaying the median proportion across the section of the percentage of each cellular cluster. We applied a Wilcoxon test one *versus* all to evaluate the enrichment in a pathological compartment of a particular cellular cluster.

- *The description of these results should also be modified to more accurately describe the data presented. For example, line 337 states that FAP+ CAF precisely co-localise with other cell types. However, the data shown only confirms that they are found in the same tissue compartment not "precisely co-localise".*

As requested, we have now more accurately described this part of the results. As detailed above and as requested, we have now removed the sentence "Deconvolution of BC section also showed that FAP+ CAF cluster were precisely colocalized with several other cell types" as well as the notion of colocalization. In the **New Fig. 3**, we focus and mention cell type enrichment within pathological compartment, as recommended by the Reviewer.

Unsupervised spatial analysis reveals shared regional cellular compositions across patients.
• *The analysis performed to generate figure 4B and identify EcoCellTypes, requires extensive clarification from both a technical and biological perspective:*

-I could not find this section of analysis described in the methods section and the description provided in the results omits important details of how the clustering was performed (e.g. distance metrics used, clustering method used). It is also important to demonstrate that identification of these ECTs is reproducible in a validation set.

We apologize for the lack of clarity in the description of these results in the initial version of our manuscript.

The section of analysis mentioned by The Reviewer was initially described in the “#Spatial Transcriptomics **Spatial Deconvolution” subsection in the Methods. As requested, we moved it in a separate section referred to as “#Niche and ECT identification” for clarifying the new version of the text, **p38-39**. In addition, we also now provide a validation of the ECT identified. As described in our manuscript, the ECT characterize the cellular composition of the niches shown in **Fig. 4**. To validate the identification of these ECT in a new dataset, we retrieved 14 publicly available Visium sections from breast cancer and applied the scANVI workflow to demonstrate that we can re-identify these structures in these new sections. This analysis showed that the niches, characterized by specific composition in cell types and states, are consistently identified in these new sections. We added this new result in the **New Supplementary Fig. 8**. Moreover, their spatial organization closely mirrors the original niches and does not extend beyond the reference boundaries. We thank the Reviewer for this suggestion that extends the reach of our study by allowing others to map new sections on our reference breast cancer spatial dataset.

Text added in the Methods section, p38-39

Niche and ECT identification: To identify groups of similar co-occurring cell types across BC sections analyzed by Visium, we merged the cell2location deconvolution output (q05_cell_abundance_w_sf) for each section, transformed the estimated cell-type proportions into isometric log ratios and created a batch balanced k-NN graph using the *bbknn* function implemented in Scanpy (<https://scanpy.readthedocs.io>) with default parameters. To identify spots composed of similar cell types, we applied Leiden clustering at a resolution of 0.6 on the resulting graph. We then computed the average cell type composition per cluster (called niches) and we generated a heatmap using the pheatmap R package. A hierarchical clustering on rows (cell types and states) and columns (niches) was finally applied with Euclidean distance and method set to Ward.D2. Heatmap’s values were centered and scaled by rows.

Text added in the Results section, p13

We wondered if our BC spatial dataset could be used as a reference to project the cellular niches on new BC sections. To do so, we retrieved 14 publicly available BC sections⁷³ analyzed by spatial transcriptomics but lacking niche annotations. After deconvolution, we built a latent embedding using scANVI with all sections as input and used the label transfer capacities of scANVI to evaluate the detection of the niches in the new sections (See Methods, *#Niche reference mapping*). This approach revealed that we were able to re-identify the different niches in new BC sections and that the spatial organization of the mapped niches closely mirrored the original niches (**Supplementary Fig. 8D, E**). Thus, this analysis showed that our breast cancer spatial dataset can serve as reference in future studies to map BC cellular niches on new data.

Text added in the Methods section, p40

Niche reference mapping: To map the niches on new BC sections, we first created and trained a scVI model with two layers using the deconvolution output of the 17 sections (which allowed us to identify the niches) and the 14 new BC sections collected from⁷³. We then converted the model to an scANVI model using `scvi.model.SCANVI.from_scvi_model` and indicated the

niches as the label of interest. We trained the model for 20 epochs and indicated 100 samples per label. We used the *predict* method of the model to transfer the niche labels on new sections, and finally exported the predicted label and plotted them on the sections.

Text added in the Legend of the Supplementary Fig. 8D, E, p63

(D) Predicted BC cellular niches on an external cohort of BC sections ⁷³ analyzed by spatial transcriptomics. **(E)** Assignment probability of the niches for the spots from the external cohort.

-From a biological perspective it is not clear what the ECTs are supposed to represent. Figure 4A shows identification of niches that represent spots with similar constituent cells. So, what is the rationale for clustering cell-types/states distribution across these niches further? Would it not be more informative to either remove the niches and examine correlations between each cell-type across all spots to measure spatial “co-localisation patterns”. Or assess what cells are associated with particular niches, for example by examining Pearson’s residuals from a Chi squared test.

We are sorry for not having provided enough explanations on ECT in the first version of our manuscript. We have now provided additional information on ECT construction and meaning. In brief, each ECT represents specific cell types and cell states co-existing within the same tumor area (e.g. each area being a niche) across BC patients. In other words, we use the concept of niches as a mean to identify local microenvironments that exhibit similar composition in cell types and cell states (referred to as ECT). By clustering cell types within niches (**New Fig. 4C**, old Fig. 4B), we could see that specific cell types and states preferentially coexist within particular local microenvironments, thereby leading to the notion of ECT. While we acknowledge the suggestion to examine correlations between each cell type across all spots, analyzing spatial co-localization of cell types (ECT) within identified niches provides a more nuanced understanding of cellular relationships in the tissue. Indeed, this allows us to explore not only which cell types are present together but also whether their distribution follows a pattern within specific niches. Moreover, by this way, we could identify colocalization patterns between different ECTs that specifically occur in a given niche but not in others, which a correlation analysis across all spots would miss. Still, as requested, we have examined Pearson’s residuals from a Chi squared test. This is provided below (**Figure 1 for the Reviewer**), where the Reviewer can appreciate that the two methods give similar results for the different cell types. The comparison between the two methods demonstrates the consistency in the outcomes for various cell types, providing confidence in the reliability of our findings. As requested, we have now detailed the ECT definition and this particular analysis in the new version of the manuscript, **p13**.

Figure 1 for the Reviewer: Pearson's residuals from a Chi squared test (Right) show similar patterns of co-occurrence between cell types and states, compared to results shown in the **New Fig. 4C** (Left).

Text added in the Results' Section, p13

We next aimed to identify unique patterns of cell types that preferentially coexisted within specific niches. To do so, we applied hierarchical clustering on the mean abundance of cell types per niche. This allowed us to identify 10 co-localization patterns within the niches that were shared across sections (**Fig. 4C**). We refer to these co-localization patterns as EcoCellTypes (ECT) which stands for ecosystem of cell types (#See Methods, #Niches and ECT identification) (**Fig. 4C**). ECT represented cell types and states co-existing across BC patients within the same tumor area (e.g. each area being a niche). Interestingly, some ECTs were found together in specific niches but not others, highlighting the interest of this approach to discover niche-specific colocalization pattern. Thus, each ECT contains a specific composition of cell types or states localized in close proximity, and we identified specific ECT enriched in different FAP+ CAF clusters.

• *In relation to the ECTs, there seems to be discrepancies in the "validation" by flow cytometry (presented in panels D-G). The correlations tested here are inconsistent with the Spatial Transcriptomic data or unclear because of a lack of clarity regarding what the ECTs truly represent. For example, the flow cytometry analysis shows ECM-myCAF positively correlate with TREM2+ TAM and negatively with FOLR2+ TAM. However, the correlation between ECM-MyCAF has not been shown in the Spatial Transcriptomic data. Indeed, figure 4B suggests that all three of these populations are enriched in similar niches (e.g. niche 6). Based on the ECTs it should be expected that the Wound-myCAF and TREM2+ TAM populations positively correlate (as they are both assigned to ECT10) but this data is not shown.*

We would like to insist on the fact that there is no discrepancy in the data shown in ECT and in flow cytometry, and we are sorry for the lack of clarity in the first version of the text. As mentioned by the Reviewer, the content in Wound-myCAF and TREM2+ TAM populations is indeed correlated in BC samples analyzed by flow cytometry, thereby validating the ECT10. As requested, to better visualize interactions between cell types found in each ECT, we have now provided correlation analyses from flow cytometry data, showing the relationships in the content of FAP+ CAF clusters among them as well as with immune cells. These data are now included in the **New Fig. 5B, C**, and described **p15**.

Text added in the Results' Section, p15

As anticipated from ECT composition, using flow cytometry from BC samples (Prospective cohort, **Supplementary Table S1**), we observed that Detox-iCAF showed a negative correlation with ECM-myCAF, TGF β -myCAF, and Wound-myCAF and a positive correlation with IL-iCAF and IFN γ -iCAF in BC patients (**Fig. 5B**). Similarly, consistent with ECT9 and ECT10, we confirmed a positive correlation between ECM-myCAF, TGF β -myCAF and Wound-myCAF. We also validated cell co-occurrence in the immuno-suppressive niche by demonstrating that these myCAF clusters were positively correlated with TREM2+ TAM and negatively correlated with FOLR2+ TAM (**Fig. 5C**). Similarly, Detox-iCAF content showed a positive correlation with FOLR2+ TAM, consistent with the immuno-protective ECT4, and a negative correlation with TREM2+ TAM proportion (ECT10) (**Fig. 5C**).

• *Can the authors clarify why a subset (n=25) of the BC cohort (n=87) was used in figure 4 F&G and describe how was this subset selected?*

We are sorry for the lack of clarity. We have not selected these samples in purpose. Depending on the size of the tumor samples and the number of cells obtained after digestion of fresh BC samples, we were able to analyze the content in both macrophage subsets and CAF-S1 clusters per sample in 25 samples. This precision is now given in the **Methods** section **p46**.

Text added in the Methods' Section, p46

Depending on the size of the tumor samples and the number of cells obtained after digestion, we were able to analyze by flow cytometry the content in both macrophage subsets and CAF-S1 clusters per sample in 25 samples.

• *What is the rationale for analysing NK cell activation in panels L-O? Was T-cell activation also assessed? If so, what were the results? For example, previous studies have suggested a role in causing Treg phenotypes however the ST data suggests that this may not be the case given the relative scarcity of Tregs in Niche 6. Do the data from this study no longer support a relationship between myCAF and Tregs?*

We thank the reviewer for raising this point. Indeed, as we observed that NK cells are part of the immunosuppressive ECT1, we initially focused our attention on NK cells. But, as requested by the Reviewer, we have now assessed the impact of FAP+ CAF clusters on T lymphocytes and TAM by performing functional assays. By this way, we now show that ECM-myCAF and TGF β -myCAF increase the percentage of FOXP3+ T lymphocytes among the CD4+ CD25+ T cells, in particular the percentages of PD1+ and CTLA4+ T cells among regulatory FOXP3+ CD4+ T lymphocytes. We have also evaluated the impact of these FAP+ CAF clusters on the proportions of TREM2+ and FOLR2+ TAM. These new data have been included in the **New Fig. 5E-J** and described in the new Text **p16**.

Text added in the Results' Section, p16

Upon co-culture of FAP+ CAF clusters with CD14+ monocytes, we observed an increase in the proportion of CD16+ cells among CD14+ monocytes (**Fig. 5E**). In addition, ECM-myCAF and TGF β -myCAF strongly increased the proportion of TREM2+ macrophages among total CD14+ CD16+ myeloid cells (**Fig. 5F**), while Detox-iCAF, IL-iCAF, and IFN γ -iCAF increased the content of FOLR2+ macrophages (**Fig. 5G**). These findings suggest that ECM-myCAF and TGF β -myCAF play an active role in modulating the identity of myeloid subtypes to form the immuno-suppressive niche, while Detox-iCAF attract monocytes and induce a FOLR2+ phenotype to form the immuno-protective niche. We also tested the impact of CAF-S1 clusters on the differentiation of CD4+ CD25+ FOXP3+ T lymphocytes *in vitro*. We observed that ECM-myCAF and TGF β -myCAF increased the percentages of FOXP3+ T cells among the CD4+ CD25+ population, while iCAF clusters had no impact (**Fig. 5H**). Moreover, ECM-myCAF and TGF β -myCAF significantly increased the percentages of both PD-1+ and CTLA4+ T cells among FOXP3+ T lymphocytes (**Fig. 5I, J**).

Specific FAP+ CAF clusters are associated with breast cancer invasiveness

• *TGF-B/TGFBR2 axis identified as possible mediator of tumour cell induced ECM-myCAF activation but not as a mechanism of Wound-myCAF activation. As described above, could this be tested in vitro? Additionally, does this suggest that the wound-myCAF phenotype would be induced by paracrine signalling from cancer cells only and could the role of juxtacrine vs paracrine signalling in this process be further elucidated in vitro (e.g. conditioned media)?*

We thank the reviewer for raising this point. As mentioned above, we have now analyzed the impact of the TGF β /TGFBR2 axis on the transition from Detox-iCAF to ECM-myCAF. These data are now included in the New **Supplementary Fig. 2K-M** and **New Fig. 2Q, R** and described **p10**. As requested, we have also analyzed the impact of conditioned medium (CM) derived from MCF7 cancer cells on Detox-iCAF and observed that this CM promotes transition from Detox-iCAF to Wound-myCAF. As this pattern is not the same as this one observed upon TGF β 2 stimulation, we can hypothesize that CM contains other secreted signaling molecules than TGF β 2 (such as CYR61?) to promote this specific transition. These data are now shown in the **New (Supplementary Fig. 2N, O)**, and discussed **p10**.

Text added in the Results' section, p10 (for TGF β data)

Based on recent data showing that the TGF β 2/TGFBR2 axis drives LRRC15+ ECM-myCAF differentiation in PDAC mouse models⁴², we wondered if this pathway could be involved in CAF-S1 cluster plasticity. We first observed that TGF β 2 stimulation, validated by SMAD family member 2 (SMAD2) phosphorylation, gradually increased the ECM-myCAF content (**Supplementary Fig. 2K-M**), suggesting that TGF β 2 is sufficient to promote the transition from Detox-iCAF to ECM-myCAF. Reciprocally, silencing of TGFBR2 in Detox-iCAF prevented the transition from Detox-iCAF to ECM-myCAF in the presence of cancer cells (**Fig. 2Q, R**), confirming that TGFBR2 is necessary for this transition. Collectively, these data highlight the role of the TGF β 2/TGFBR2-dependent pathway in the emergence of ECM-myCAF from Detox-iCAF.

Text added in the Results' Section, p10 (for conditioned medium)

Finally, we evaluated the impact of conditioned medium (CM) derived from MCF7 cancer cells on Detox-iCAF and observed that CM promoted the transition from Detox-iCAF to Wound-myCAF (**Supplementary Fig. 2N, O**). The pattern of induced clusters differed between TGF β 2 and CM stimulation, suggesting that CM contains other secreted signaling molecules in addition to TGF β 2 which promote this specific transition and that the TGF β 2-mediated effect requires direct contact between cancer cells and Detox-iCAF.

Detox-iCAF and TGF β -myCAF proportions predict progression from DCIS to IBC
• *As per previous comments, were any differences observed in the normal fibroblast populations? Given that these samples are DCIS presumably healthy fibroblast populations are a major constituent of the stromal compartment in these tissues?*

The differences in universal fibroblasts in DCIS, MI-DCIS and IBC samples are shown in the **New Supplementary Fig. 9C**. As mentioned by the Reviewer, we observed that universal fibroblasts (isolated from human mammaplasties) accumulate more in DCIS compared to IBC, as expected. These results are shown in **Supplementary Fig. 9C**. and described **p16-17**.

Text in the Results' section, p16-17

We next determined the proportion of each cell type and cell state identified in our BC atlas by deconvolution and observed that cell type composition was distinct according to BC invasiveness (**Fig. 6C, D** and **Supplementary Fig. 9B, C**). In particular, we observed a significant accumulation of universal fibroblasts (identified in healthy tissues) in DCIS (**Supplementary Fig. 9C**), while the significant increase in fibroblasts in IBC was mainly due to FAP+ CAF (**Fig. 6C**).

Reviewer #2 (Remarks to the Author): with expertise in single-cell omics, spatial transcriptomics

Comments to the Author

This study utilized trajectory inference, spatial transcriptomics, and functional assays to create an integrative high-resolution map of breast cancer (BC). It provides fresh insights into the spatial organization of the BC tumor microenvironment, with a specific focus on FAP+ CAF diversity, plasticity and interactions with surrounding cells. This is an interesting study, but some of the data interpretation requires further clarification.

We are really grateful to the Reviewer for the positive assessment of our work, and for considering our study interesting and providing new insights into the field. We have now answered to all reviewer's concerns, which improve the quality of our manuscript.

1. In Fig. 2D, FOSL1 is the main up-regulated gene in the indirect transition, and FOSL1 is frequently overexpressed in multiple types of human cancers including IBC, with the functions related to cancer invasion and metastasis. Why does the author focus on TEAD1 instead of FOSL1? Does FOSL1 influence the plasticity of FAP+ CAF clusters?

We thank the Reviewer for raising this point. We have been interested by the YAP1/TEAD1 pathway, first because we observed that YAP1/TEAD1-dependent gene signature is up-regulated in the scRNAseq data of the indirect transition (Detox-iCAF to Wound-myCAF and next to ECM-myCAF). We also focused on the YAP1-pathway because we observed that FAP+ myCAF clusters can be stimulated in plastic dishes. Nevertheless, we sought to address the question of the Reviewer regarding FOSL1. To do so, we inactivated *FOSL1* in Detox-iCAF and tested the impact of this inactivation in the transition from Detox-iCAF to Wound-myCAF and to ECM-myCAF. As shown in the **Figure 2 for Reviewer** below, inactivation of *FOSL1* has no impact on the transition from Detox-iCAF to Wound-myCAF and to ECM-myCAF in presence of cancer cells.

Figure 2 for Reviewer: Inactivation of *FOSL1* in Detox-iCAF has no impact on the transition from Detox-iCAF to Wound-myCAF and to ECM-myCAF in presence of cancer cells. **(A)** Representative western blot showing *FOSL1* protein level in primary Detox-iCAF transfected either with siCTRL or with two specific siRNA targeting *FOSL1*. The time points analyzed are 24 hours, 48 hours and 72 hours after co-culture with cancer cells. Actin is used as an internal control for protein loading. **(B)** Bar blots showing the fraction of FAP+ CAF clusters after co-culture of MCF7 cancer cells with Detox-iCAF transfected either with siCtrl or with *FOSL1*-targeting siRNA (siFOSL1#1 and siFOSL1#2). Data are mean \pm SEM ($n = 3$). Respective identity of each cluster is based on specific markers assessed by flow cytometry: ECM-myCAF (ANTXR1+ SDC1+ LAMP5-, pink); TGF β -myCAF (ANTXR1+ SDC1+ LAMP5+, dark green); IL-iCAF (ANTXR1- DLK1+ GPC3-, light green); Detox-iCAF (ANTXR1- DLK1+/- GPC3+, yellow) and IFN γ -iCAF (ANTXR1- DLK1- GPC3- CD74+, blue)

2. Cell2location, SpatialDWLS, and RCTD have been identified as the top-performing methods for the cell type deconvolution of spots in spatial transcriptomics. However, it should be noted that the results obtained from these three methods may exhibit some degree of variation. In this study, the authors utilized Cell2location for their analysis. It raises the question of whether the main findings and conclusions reached by the authors can be replicated using the other two methods.

As requested by the Reviewer, we have now applied two other deconvolution methods, RCTD and SpatialDWLS, and conducted a comparative analysis with the spatial transcriptomic data that we generated and deconvoluted using cell2location. To perform this in-depth comparison, we studied the section showed in the **Fig 3C**, which is composed of tumor and peritumor compartments, invasive margin, lymphocyte aggregates, as well as normal ducts and lobules, and thus representative of all pathological compartments. Consistent with the Reviewer's observation, some variability and performance differences could be expected due to variations in the algorithms' methodologies. Still, our results indicate a high degree of correlation across most cell types with these two other methods, with RCTD showing particularly strong agreement. As requested, these results are now added in **Supplementary Fig. 5D, E** and described in the **Results'** section, **p11**, its corresponding Legend **p62**, as well as in **Methods, p38**.

Text added in the Results section, p11

We also tested RCTD and SpatialDWLS, two other top-performing deconvolution methods^{68, 69, 70}, both of which produced results entirely consistent with cell2location (**Supplementary Fig. 6D, E**).

Text added in the Legend of the Supplementary Fig. 6, p62

(D) Comparison of deconvolution results of the section displayed in **Fig. 3C** (representative of the main pathological compartments) for the different cell types and states by using two other deconvolution methods (RCTD and SpatialDWLS). Cell2location values represent the number of deconvoluted cells per spot. RCTD and SpatialDWLS results are expressed as cell type proportions per spot. **(E)** Data show correlations between Cell2location deconvolution and RCTD or SpatialDWLS methods. The symbol * indicates significant ($P < 0.05$) correlations from Pearson correlation test. Colors show correlation coefficients.

Text added in the Method p38

For deconvolution with SpatialDWLS⁶⁹, we used the function *runDWLSDeconv* with default parameters implemented in Giotto. For RCTD⁶⁸, deconvolution we first created a RCTD object and deconvoluted the section using the *run.RCTD* function with doublet mode set to multi.

3. Ten EcoCellTypes (ECT) were identified. Do these ECT exhibit any differences between TN and LUM?

We thank the reviewer for raising this point. As requested, we have now investigated the proportions of the different ECT in the distinct BC subtypes by performing deconvolution of transcriptomic data from the METABRIC cohort (https://www.cbioportal.org/study/summary?id=brca_metabric) using the BayesPrism algorithm. We performed this analysis on 1234 patients including 487 Lum A, 368 Lum B, 193 HER2 and 186 Basal-like TN BC and highlighted significant differences in ECT composition among BC subtypes. Specifically, in terms of immune components, we found that Lum A BC exhibit a lower fraction of immuno-suppressive cells (ECT1) compared to LumB, HER2, and Basal-like TN BC. Conversely, Lum A BC display a higher proportion of naive CD4 T cells, as well as precursor and cytotoxic CD8+ T lymphocytes (which constitute the ECT2) than the other BC subtypes. We also observed significant differences in the stromal compartment. Lum A BC show a higher content in ECT5 (composed of IL-iCAF, Adipo-EC, contractile-CAP) and in ECT9 (ECM-myCAF, TGF β -myCAF, Ag-CAP) compared to Lum B, HER2 and Basal-like TN BC. In contrast, ECT10 (Wound-myCAF, ECM-CAP and TREM2+ TAM) is enriched in Lum B and HER2 and particularly abundant in Basal-like TN BC, while ECT4 (Detox-iCAF, FOLR2+ TAM) accumulate in Lum A BC. This analysis of the METABRIC cohort led us to look at the impact of tumor ECT composition on patient survival. Remarkably, ECT composition of BC allowed us to stratify patients in 4 subgroups (C1 to C4) with different overall survival. Patients from the C2 subgroup, which is enriched in ECT1 (immuno-suppressive), ECT3 (plasma cells) and ECT10 (TREM2+ TAM, Wound-myCAF), are mainly composed of Basal-like TN (31.5%), Lum B (31.7%) and HER2 (23%) BC subtypes and show the worst survival, as expected. In contrast, patients from the C3 subgroup -enriched in ECT2 (immune precursors, effectors), ECT4 (Detox, FOLR2+ TAM) and ECT5 (IL-iCAF, adipo-EC)- are mainly of the Lum A subtype (69.6%) and show the best overall survival. The C1 and C4 subgroups are mainly composed of Luminal patients (Lum A and Lum B). While their 5-years survival is good and comparable to this one of the C3 subgroup, the prognosis of the C1 and C4 subgroups fall after 5 years. Compared to the C3 subgroup, this difference of survival can be explained -at least in part- by enrichment in Lum B BC and in ECT6-7 (Epithelial cells, IFN $\alpha\beta$ -myCAF, SPP1+ TAM) for the C4 subgroup and in ECT9 (Angio-EC, ECM-myCAF, TGF β -myCAF) for the C1 subgroup. These new data are now included in the **New Fig. 4D-F** and described **p14**, and its corresponding legend **p56**, as well as in the **Methods** section, **p40-41**.

Text added in the Results section, p14

To investigate if the composition of these ECT differed between BC molecular subtypes, we performed deconvolution of transcriptomic data from the METABRIC cohort (487 Lum A, 368 Lum B, 193 HER2 and 186 Basal-like TN BC) using BayesPrism⁷⁴. In terms of immune components, we found that Lum A BC exhibited a lower fraction of immuno-suppressive cells (ECT1) compared to Lum B, HER2, and Basal-like TN BC (**Fig. 4D**). Conversely, Lum A BC displayed a higher proportion of naive CD4 T cells, as well as precursor and cytotoxic CD8+ T lymphocytes (which constituted the ECT2) compared to the other BC subtypes. We also observed significant differences in the stromal compartment: Lum A BC contained more ECT5 (composed of IL-iCAF, Adipo-EC, contractile-CAP) and ECT9 (ECM-myCAF, TGF β -myCAF, Ag-CAP) compared to Lum B, HER2 and Basal-like TN BC. In contrast, ECT10 (Wound-myCAF, ECM-CAP and TREM2+ TAM) was enriched in Lum B and HER2 and particularly abundant in Basal-like TN BC, while ECT4 (Detox-iCAF, FOLR2+ TAM) accumulated in Lum A (**Fig. 4D**). Remarkably, ECT composition in BC allowed us to stratify patients into 4 subgroups (C1 to C4) with different overall survival. (**Fig. 4E, F**). Patients in the C2 subgroup, with an enrichment in ECT1 (immuno-suppressive), ECT3 (plasma cells) and ECT10 (TREM2+ TAM, Wound-myCAF), were mainly composed of Basal-like TN (31.5%), Lum B (31.7%) and HER2 (23%) BC subtypes and showed the worst survival, as expected. In contrast, patients in the C3 subgroup -enriched in ECT2 (immune precursors, effectors), ECT4 (Detox, FOLR2+ TAM) and ECT5 (IL-iCAF, adipo-EC)- were mainly composed of the Lum A subtype (69.6%) and showed the best overall survival. The C1 and C4 subgroups were mainly composed of Luminal patients (Lum A and Lum B). While their 5-year survival was good and comparable to that of the C3 subgroup, the prognosis of the C1 and C4 subgroups fell after 5 years. Compared to the C3 subgroup, these survival differences could be explained in part by the C4 subgroup's enrichment in Lum B BC and ECT6-7 (Epithelial cells, IFN $\alpha\beta$ -myCAF, SPP1+ TAM) and the C1 subgroup's abundance of ECT9 (Angio-EC, ECM-myCAF, TGF β -myCAF) (**Fig. 4E, F**). In conclusion, ECT were associated with the overall survival of BC patients, which is linked to ECT-enrichment in BC molecular subtypes.

Text added in the Legend of the Fig. 4D-F, p56

(D-F) Data from the METABRIC cohort (N = 1234 BC patients). **(D)** Proportions of each ECT in Lum A (N = 487), Lum B (N = 368), HER2 (N = 193) and Basal-like TN (N = 186) BC subtypes. P-values from Mann-Whitney test. **(E)** Heatmap and clustering of all BC samples (columns) from the METABRIC cohort showing 4 subgroups of patients (C1 = 263, C2 = 483, C3 = 227, C4 = 261) with different ECT enrichments. **(F) Left**, Kaplan–Meier curves showing overall survival of the 4 BC patient subgroups (C1-C4) stratified in the heatmap. P-value from Log-rank test. **Right**, Distribution of the BC molecular subtypes within the 4 subgroups (C1-C4) of BC patients.

Text added in the Methods, p40-41

Analysis of ECT composition in the METABRIC cohort: To analyze ECT enrichments in BC molecular subtypes, we used the BayesPrism algorithm⁷⁴ to deconvolute transcriptomic data from the METABRIC cohort. Normalized expression matrix, clinical information and PAM50 subtype classifications were obtained from METABRIC (https://www.cbioportal.org/study/summary?id=brca_metabric). Luminal A (N = 487), luminal B (N = 368), HER2 (N = 193) and Basal-like TN (N = 186) BC patients were conserved for the analysis. Raw count matrix of 73 426 high-quality cells from our BC atlas was used as input for prior information. Labels were derived from the annotation of 10 ECT described above. ECT6 and ECT7 were pooled to group all epithelial cell types. Mitochondrial and ribosomal protein coding genes were removed as these genes are expressed at high magnitude and not informative in distinguishing cell types. MALAT1 and genes from chrX and chrY were also removed following indication from BayesPrism's authors. To reduce batch effects and speed up computation, we performed deconvolution only on protein coding genes. Default parameters to control Gibbs sampling and optimization were used. Final estimation of cell type fraction in each bulk RNA-seq sample was recovered using the updated theta matrix and used for downstream analysis. Stratification of tumors was done by applying hierarchical clustering

on the matrix of ECT fraction obtained using correlation distance and Ward.D2 method from *heatmap* R package. Differences in overall survival between the 4 subgroups of BC patients were assessed using Kaplan–Meier analysis and log-rank test statistics using the *survival* and *survminer* R packages.

4. Myeloid-derived suppressor cells are the major immunosuppressive cells found in BC, but they are not detected in the comprehensive BC cellular landscape (Fig. 3B and Supplementary Fig. 3C). Please clarify the reason.

We thank the Reviewer for this comment on Myeloid-derived suppressor cells (MDSC). To address this point, we considered the MDSC gene signature established in (*Alshetaiwi, Science Immuno, 2021*), which was originally used to identify MDSC in scRNAseq data. We applied this signature in the same manner by computing a module score with Seurat, which uncovered that S100A12+ Monocytes and CXCL10+ macrophages were the myeloid clusters expressing the most the MDSC program in our BC atlas (see below **Figure 3 for the Reviewer**). CXCL10+ macrophages are in the same ECT as cancer cells and IFN $\alpha\beta$ -myCAF (ECT7), while S100A12+ monocytes are associated with CLEC9A+ cDC1 (ECT8). We thank the Reviewer for this insight in our work, where looking at the crosstalk between tumor cells, IFN $\alpha\beta$ -myCAF and MDSC could be an interesting scope for a future work. We have now indicated this point in the new version of the Text, **p33**.

Text added in the Results section, p33

Interestingly, S100A12+ Mo and CXCL10+ macrophages were the cell populations expressing the most the gene signature from myeloid-derived suppressor cells⁹⁸.

Figure 3 for Reviewer: Applying MDSC gene signature (*Alshetaiwi, Science Immuno, 2021*) by computing a module score with Seurat, showed that S100A12+ Monocytes and CXCL10+ macrophages were the myeloid clusters expressing the most the MDSC program in our BC atlas.

5. The relationship between the transition of FAP+ CAF clusters, their spatial location, and function requires further discussion and elucidation.

We apologize to the Reviewer for the lack of discussion of this important aspect of the presented work. We believe that these additional insights into the spatial distribution and functional transitions of FAP+ CAF clusters significantly enhance the comprehensiveness of our study. These findings are now discussed in the manuscript to provide a more comprehensive understanding of the complex interactions within the tumor microenvironment.

Text modified in the Discussion section, p20-21

Our functional analysis revealed that Detox-iCAF, primarily located in the interlobular stroma, can give rise to ECM-myCAF through two transitions, one direct and one indirect via the Wound-myCAF cluster. Notably, ECM-myCAF are systematically close to cancer cells, while Wound-myCAF are located further away. This suggests that juxtacrine interactions between

Detox-iCAF and cancer cells may be required for the direct transition, while paracrine signaling could induce the larger area of Wound-myCAF. This is supported by our experiments showing that cancer cells-conditioned media induce Wound-myCAF from Detox-iCAF but not ECM-myCAF, while co-culture with cancer cells induces both. Observations in DCIS support this hypothesis, as we identified ECM-myCAF directly adjacent to the tumor nests and Wound-myCAF farther away. Specific FAP+ CAF clusters and immune cells are spatially organized in tumors, offering insights into FAP+ CAF-mediated immunoregulation. We uncovered the crosstalk between different FAP+ CAF clusters and immune cells subtypes, which modulates immune cell identity and states, highlighting immuno-permissive and immunosuppressive ECT in BC. We previously demonstrated that TGF β -myCAF can be induced from ECM-myCAF through interactions with T lymphocytes¹⁰. Interestingly, we found that TGF β -myCAF are enriched in immune infiltrated sites in BC, including the invasive margin, around intra-tumoral blood vessels, and around intra-tumoral lobules. We also identify ECT enriched in Detox-iCAF, FOLR2+ TAM and ap-EC. FOLR2+ TAM are known to be close to peritumoral blood vessels in BC⁸⁹. Consistent with these observations, Detox-iCAF attract monocytes and induce a FOLR2+ TAM program. Detox-iCAF are localized close to ap-EC-enriched blood vessels, which express markers of tumor-associated high endothelial venule (TA-HEV), suggesting that Detox-iCAF may play a role in the crosstalk with ap-EC and TA-HEV maturation. In contrast to Detox-iCAF, IL-iCAF are primarily associated with normal breast epithelial structures and found in the intralobular stroma enriched in Adipo-EC, while IFN γ -iCAF are associated with lymphocyte aggregates and enriched in the tumor bed. We observed that IL-iCAF are not able to differentiate into ECM-myCAF, suggesting that myofibroblasts observed in BC might not be primarily derived from the intralobular stroma. As the intralobular stroma is not present in mice⁹⁰, this may explain why IL-iCAF are not detected in scRNA-seq data from mouse models. Moreover, ECM-myCAF, IFN $\alpha\beta$ -myCAF and TGF β -myCAF accumulate within the invasive compartment, with ECM-myCAF and IFN $\alpha\beta$ -myCAF in the close vicinity of cancer cells, while TGF β -myCAF are associated with immune infiltration and found at the invasive margin, around intratumoral infiltrated lobules or blood vessels. Consistent with our observations, previous work has shown that *Podoplanin*-expressing myofibroblasts (corresponding to ECM-myCAF and IFN $\alpha\beta$ -myCAF) are enriched at the interface with cancer cells where non-activated fibroblasts and EC were depleted⁸⁰. Moreover, ECM-myCAF and TGF β -myCAF colocalize with TREM2+ TAM and actively induce the TREM2+ TAM program to create an immunosuppressive area in the tumor bed. In relation to this spatial organization, ECM-myCAF are detected close to exhausted CD8+ T lymphocytes and immunosuppressive FOXP3+ CD4+ T lymphocytes. Finally, Wound-myCAF are mainly distributed in the intratumoral stroma, more distant from cancer cells than any other myCAF subsets. Collectively, the spatial distribution of all FAP+ CAF clusters is compatible with their plasticity and crosstalk with cancer or immune cells, and uncovers FAP+ CAF-immuno-permissive and immunosuppressive ECT (See Model, **Fig. 8**).

Minor comments:

1. *CAF fibroblasts should be written as CAF because the existing abbreviation "CAF" already stands for cancer-associated fibroblasts. Some abbreviations are not defined, such as PDAC.*

We thank the Reviewer for this comment. As requested, we have now changed "CAF fibroblasts" for "CAF" and done our best to explain the abbreviations, such as PDAC, when first used in the text. We thank the Reviewer for the careful reading of the Text, which improves our manuscript

2. *Please use consistent colors for the different subtypes of CAF in Fig. 1D, as in the other panels.*

We thank the Reviewer for this comment. As requested, we have now regenerated the panel in the **New Fig. 1D**, using the same colors and point size as in the other panels of the Figure.

3. Supplementary Fig. 1I lacks legend description.

We thank the Reviewer for the careful reading of our manuscript, and we apologize for the lack of this legend in the initial version of our paper. As requested, we have now included a legend describing the old Supplementary Fig. 1I., which corresponds now to the **New Supplementary Fig. 2J**, in the new version of the Text.

Text added in the Legend of the Supplementary Fig. 2J, p60

(J) UMAP of the FAP+ CAF dataset from BC highlighting cells implicated in the direct transition from Detox-iCAF to ECM-myCAF (red) and in the indirect transition from Detox-iCAF to Wound-myCAF (blue).

4. What is the corresponding method to the result "upon TGFBR2 stimulation, Detox-iCAF increased YAP1 activity (Fig. 5F)"?

We apologize for this point raised by the Reviewer. We agree that the Method used for this analysis was initially hard to find in the first version of our manuscript, and the methodology not detailed enough in the Result section. To answer to this comment, we have now cited in a better way in the Results section, **p17**, the corresponding Method, which is described in the **"#Spatial transcriptomic analysis" subsection** of the **"#Ligand-Receptor analysis" section, p40**. Moreover, to make the method clearer for the reader, we added some information about the Method used in the Result section, **p17**.

Text added in the Result's section, p17

In turn, upon TGFBR2 stimulation, Detox-iCAF increased YAP1 activity, as identified by the ligand-receptor transcription-factor knowledge-graph-based approach implemented in SpaTalk (**Fig. 6F**) (see also Methods, **#Ligand-Receptor analysis**), consistent with the role of YAP1/TEAD in FAP+ CAF plasticity we described above.

Text in the Methods' section, p40

Spatial transcriptomic analysis: Spatial Ligand-Receptor analyses were performed using SpaTalk v1.0⁷⁶. We used the function `dec_celltype` to transfer the cell2location deconvolution output on the SpaTalk object created using `createSpaTalk` with default parameters. We used the function `set_expected_cell()` with the result of cell2location cell abundance to indicate the number of cell per spot. We then used `find_lr_path` to infer cell-to-cell Ligand-Receptor communications and `dec_cci` to indicate the cell types of interest. Finally, we recovered activation of the downstream pathways using `get_lr_path` and plotted the result using `plot_lr_path` and `plot_lrpair` by indicating the cell implicated using the SpaTalk slot `cellpair`.

Reviewer #3 (Remarks to the Author): with expertise in CAFs, omics

Summary

Crozier et al build on prior data from their group concerning FAP expressing CAFs. The highlight of this work is the effort to untangle spatial relationships of CAF subtypes in solid tumors, which remains a challenging topic to approach and of great importance as the world moves towards potential clinical targeting of CAFs. While the topic is of great interest at this time, enthusiasm is dampened by the limited integration of the CAF subtypes characterized here with similar subtypes delineated across the recent literature. The conclusions to be drawn from the clinical correlations are overstated. Finally, the in vitro work, which makes up the bulk of the functional validation for the story (key to understanding the value of this work towards translational science), is based on a relatively few number of markers. Particularly the FACS plot gating seems inconsistent and is difficult to follow.

We are grateful to the Reviewer for the positive assessment of our study. We have now addressed all Reviewer's concerns, which increase the quality of our work. Thank you very much for the careful reading of our manuscript.

Major Comments

1. This paper focuses on an FAP expressing CAF subtype termed “Detox-iCAF”. While gene groups characterized as “detoxification” pathways may well be highlighted by the gene expression patterns shown by this subgroup, the focus of this work is far from detoxification. I would recommend selecting a name more closely aligned with the specific fibroblast/CAF behaviors of this subset highlighted in the work.

We thank the Reviewer for raising this point, and we are sorry for the lack of clarity in the initial version of our manuscript. The nomenclature of all FAP+ CAF / CAF-S1 clusters has been given based on differentially expressed genes and described in a previous publication (Kieffer, *et al.*, *Cancer Discovery*, 2020). As shown in our previous paper, the gene signature characterizing the Detox-iCAF cluster includes genes involved in detoxification, such as GPX3, MGST1, SOD2, SOD3, ANXA1, PDK4 among others. We do agree that the DPP4-dependent and YAP-1-dependent pathway involved in the direct transition from Detox-iCAF to ECM-myCAF and in the indirect transition through the Wound-myCAF, respectively, are not associated with detoxification. Indeed, upon cancer cell exposure, Detox-iCAF lose their detoxification identity and become myofibroblastic. The Reviewer’s comment is very interesting as highlighting this switch. As requested, we have now included this mention in the new version of the Text, **p9**

Text added in the Result’s section, p9

Thus, consistent with *in silico* trajectories among FAP+ CAF clusters, these findings show that cancer cells promote a switch from a detoxification-associated inflammatory pathway (Detox-iCAF) to a myofibroblastic signature (ECM-myCAF) in FAP+ CAF both *in vitro* and *in vivo*.

2. The labeling in figure 1 A-C is very confusing and makes the comparisons difficult to follow – within these three panels, there are labels for “universal fibroblasts”, “fibroblasts”, “fibroblasts (healthy tissue)”, “normal fibroblasts” as seemingly separate entities. The identity of and rationale for selection of the “normal fibroblasts” used in panel 1C in particular deserves specific attention, presumably this is simply a comparison of activated and non-activated fibroblasts more generally, rather than specific to so called “Detox-iCAFs”? One would expect myCAFs (characterized by α SMA expression) to show a similar appearance based on the markers highlighted? As such, this panel adds little to the story.

We do agree with the Reviewer that the initial labeling of fibroblasts from healthy tissues in **Fig. 1A-C** was difficult to follow, and we apologize for that. Based on the Reviewer’s comment, we sought to better describe that the FAP+ CAF (CAF-S1) are specific to breast cancer tissues, while both universal fibroblasts and normal fibroblasts were isolated from reduction mammoplasties. In order to clarify, we changed the labeling in the **New Fig. 1A, B** and strictly highlight the tissue of origin (breast cancer or reduction mammoplasty) for less confusion. For the **New Fig. 1C**, we now detail that we compare the genes up-regulated in fibroblasts isolated from reduction mammoplasties (Left part of **Fig. 1C**) to the genes up-regulated in the Detox-iCAF from breast cancer (Right part). We also modified the legend of **Fig. 1A-C** accordingly, **p53**. Our goal in this analysis is to show that while Detox-iCAF are transcriptionally close to the universal fibroblasts isolated from healthy breast tissues (as seen on the UMAP **Fig. 1A, B**), they are different and show a substantial transcription reprogramming characterized by up-regulation of *FAP* and *ACTA2*. We therefore modified the new version of the Text **p6** to better explain the rationale of this analysis, as requested.

Text added in the Results’ section, p6

As Detox-iCAF showed transcriptional similarities with universal fibroblasts, we sought to confirm that the Detox-iCAF cluster was transcriptionally different from normal fibroblasts. Differential analysis between fibroblasts from healthy mammary tissue and Detox-iCAF confirmed that Detox-iCAF showed a large number of up-regulated genes compared to normal

fibroblasts, such as FAP (**Fig. 1C**), consistent with the CAF-S1 isolation method based on FAP marker ^{10, 17, 18, 19}.

Text added in the legend of the Fig. 1, p53

(C) Differential analysis between Detox-iCAF and normal fibroblasts isolated from healthy breast tissues, confirming up-regulation of *FAP* and *ACTA2* genes in Detox-iCAF.

Text added in the Methods, p34

Differential analysis between fibroblasts from healthy breast tissues and Detox-iCAF from breast cancer: Differentially expressed genes between fibroblasts isolated from healthy breast tissues ^{55, 56} and Detox-iCAF from breast cancer ¹⁰ were obtained by using *FindMarkers* function from Seurat R package. To validate the result obtained and to account for sample size inflation in scRNA-seq data, cell-level counts were also aggregated at sample-level after pseudo-bulk reconstruction using *muscat* R package ⁹². Differentially expressed genes at pseudo-bulk level were then obtained using *DESeq2* R package.

3. *I would wonder if there are genetic animal models by which some of the proposed in silico lineage relationships might be demonstrated in vivo beyond the addition of published scRNA-seq datasets (1P, Q)? Do the authors believe these represent committed lineages? If so, could this be validated? If not, a greater discussion of ongoing plasticity would add significant nuance to the paper.*

As requested, we have now discussed in a more detailed manner data obtained from mouse models and *in vitro* culture, in the **Discussion p19-20**.

Text added in the Discussion, p19-20

DPP4 has also been recently implicated in the transition from normal fibroblasts to iCAF in mice ⁸⁴. Moreover, genetic evidence in PDAC mouse models showed that iCAF can be converted into LRRC15+ myCAF (LRRC15 being a specific marker of ECM-myCAF ¹⁰) by up-regulating the TGFBR2-dependent signaling pathway ^{5, 42}. Consistent with these findings, YAP- and TGFβ-signaling pathways can act simultaneously to promote a cellular transition of DPP4+ adipocyte progenitors toward DPP4- SMA+ myofibroblasts ^{85, 86}. Moreover, we have provided here a spatial resolution of these interactions by using SpaTalk and showing the TGFβ-mediated crosstalk between Detox-iCAF and cancer cells at the invasive margin leading to YAP1/TEAD activation. In line with previous studies showing that 3-dimensional culture of mouse PDAC models can promote conversion between iCAF and myCAF ⁵, we observed that collagen-coated plates increase the proportion of iCAF. As we found that YAP1 is a key player in the indirect transition from Detox-iCAF to Wound-myCAF and subsequently to ECM-myCAF, we can hypothesize that the slight reduction of the stiffness in collagen-coated dishes might favor iCAF maintenance in culture. Thus, these studies, as well as our results, reveal that FAP+ CAF populations can convert into one another depending on the spatial and biological context.

4. *It should be made clear to the reader on in vitro panels precisely which protein markers are being used to delineate each of the subtypes. Furthermore, given that the identity of the CAF subpopulations are largely based on the upregulated expression of a single marker (or in some cases 2 markers), it would make the manuscript significantly more readable if each sub-type were simply named after their protein marker since these protein markers are relied on for validation. Eg. Detox-iCAF = GPC3+ CAFs. This seems supported by S3-C. Furthermore, a discussion beyond the previously published work about how these particular markers were selected for this paper would be helpful. Presumably there are many potential protein candidates based on the gene expression data?*

5. *Building on the prior comment, it would benefit the reader significantly to have some specific gene panels displayed of functionally relevant genes related to fibroblast activities to validate the selection of the number of clusters used here. Panel 1G is of the few panels that displays individual genes for specific clusters.*

The names of the FAP+ CAF (CAF-S1) clusters and the markers used for the gating strategy identifying them have been based on differentially expressed genes and described in detail in one of our recent publication (Kieffer, *Cancer Discovery*, 2020). In brief, these different markers were defined based on pairwise comparisons of gene expression profiles from FAP+ CAF clusters that helped us to identify specific genes of each cluster. Moreover, we considered surface expressed proteins for which commercially antibodies were available allowing both flow cytometry analysis and cell sorting, as shown in (Kieffer, *Cancer Discovery*, 2020).

Still, as recommended by the Reviewer, we have now better explained the markers and the gating strategy used to identify the different CAF-S1 clusters all along the new version of the manuscript, in particular in the Methods, **p44-45** (#*Characterization of FAP+ CAF clusters*). In brief, the gating strategy relies on nine different markers, based on the differentially expressed genes between the different FAP+ CAF clusters identified in (Kieffer, *Cancer Discovery*, 2020). FAP+ CAF were first separated on the basis of the ANTXR1 protein level that distinguished myofibroblastic (myCAF, ANTXR1⁺) from inflammatory (iCAF, ANTXR1⁻). ANTXR1⁺ myCAF clusters were next distinguished according to SDC1, LAMP5, and CD9 protein levels. ANTXR1⁺ SDC1⁺ LAMP5⁻ were defined as ECM-myCAF, ANTXR1⁺ LAMP5⁺ SDC1^{+/-} as TGFβ-myCAF, and ANTXR1⁺ SDC1⁻ LAMP5⁻ CD9⁺ as Wound-myCAF. ANTXR1⁻ iCAF clusters were separated using GPC3, DLK1 and CD74 markers. ANTXR1⁻ GPC3⁺ DLK1^{+/-} were defined as Detox-iCAF; ANTXR1⁻ GPC3⁻ DLK1⁺ as IL-iCAF and ANTXR1⁻ GPC3⁻ DLK1⁻ CD74⁺ as IFNγ-iCAF. The cut-offs for the gating and for each marker was defined based on the isotype controls represented in black on the FACS plots. As recommended by the Reviewer, we have now indicated in more details in the FACS plots the distinct markers characterizing these CAF-S1 clusters and we have described them carefully in the new version of the text, **p44-45**.

Text added in the Methods, p44-45

Characterization of FAP+ CAF clusters: The markers used for the gating strategy identifying FAP+ CAF (CAF-S1) clusters have been based on differentially expressed genes and described in detail in one of our recent publication ¹⁰. These different markers were defined based on pairwise comparisons of expression profiles from FAP+ CAF clusters that helped us to identify specific genes for each cluster. In addition, we considered surface expressed proteins for which commercially antibodies were available allowing both flow cytometry analysis and cell sorting. These markers cannot be used separately as the identification of the clusters relies on the successive combination of different markers, as described below. The identification of FAP+ CAF clusters relies on the combination of these different markers tested successively. In brief, FAP+ CAF were first separated on the basis of the ANTXR1 protein level that distinguished myofibroblastic (myCAF, ANTXR1⁺) from inflammatory (iCAF, ANTXR1⁻). ANTXR1⁺ myCAF clusters were next distinguished according to SDC1, LAMP5, and CD9 protein levels. ANTXR1⁺ SDC1⁺ LAMP5⁻ were defined as ECM-myCAF, ANTXR1⁺ LAMP5⁺ SDC1^{+/-} as TGFβ-myCAF and ANTXR1⁺ SDC1⁻ LAMP5⁻ CD9⁺ as Wound-myCAF. ANTXR1⁻ iCAF clusters were separated using GPC3, DLK1 and CD74 markers. ANTXR1⁻ GPC3⁺ DLK1^{+/-} were defined as Detox-iCAF; ANTXR1⁻ GPC3⁻ DLK1⁺ as IL-iCAF and ANTXR1⁻ GPC3⁻ DLK1⁻ CD74⁺ as IFNγ-iCAF.

6. *What does LRRC15 expression look like in human or mouse breast cancer data? The introduction of this marker, which in other closely related work is used to define a specific subpopulation, should come with a placement of its relevance in the largery body of the presented work (or a discussion of its exclusion).*

LRRC15, first identified in (Dominguez, *Cancer Discovery*, 2020), is a marker that we also discovered as a specific marker of the ECM-myCAF cluster in human breast cancer in our recent study (Kieffer, *Cancer Discovery*, 2020). We have also confirmed its high expression in the ECM-myCAF cluster in the BC atlas built in our current study (see **Figure 4 for Reviewer**). Interestingly, Lrrc15 has also been identified in pancreatic cancer mouse models as a specific

marker of CAF and we took advantage of the recently published *Lrrc15*-diphtheria toxin receptor knock-in PDAC mice (Buechler, *Nature*, 2021) to test the impact of ECM-myCAF depletion *in vivo*. As requested, and to better clarify this point, we have now included the *LRRC15* expression in human breast cancer in the **Supplementary Fig. 2I** and described **p8**.

Figure 4 for Reviewer: UMAP showing *LRRC15* expression in the ECM-myCAF cluster in scRNAseq data from Kieffer, et al., *Cancer Discovery*, 2020 (Left) and in the BC atlas built in the current study (and shown in Fig. 3B).

Text added in the Results' section, p8

As *LRRC15* was also identified as a specific marker of ECM-myCAF in human BC¹⁰ (**Supplementary Fig. 2I**), we took advantage of the recently published *Lrrc15*-diphtheria toxin receptor knock-in PDAC mouse model⁵⁴ to investigate how FAP+ CAF cluster composition evolved after ECM-myCAF depletion *in vivo*.

Text added in the corresponding legend of Supplementary Fig. 2I, p60

(I) UMAP showing *LRRC15* expression in the ECM-myCAF cluster in FAP+ CAF clusters from BC¹⁰ (Left) and in the BC scRNAseq cellular atlas built in the current study and shown in Fig. 3C (Right).

7. Supplemental S1 – the FACS plots should be made large enough to be able to see the axes. The gating strategy is difficult to follow, particularly in S1D – I have some concern about the ANTXR antibody. Based on markers used, IL-iCAF does not appear to have independent identity. Similarly, the identity of TGFB-myCAF as separate from ECM-myCAF is not convincing. The distinction between ECM-myCAF and IFNab-myCAF is further unclear based on S3E plot. Perhaps the additional of more standard markers of myCAFs would be helpful to clarify. Similar value cut-offs for gating should be used between cell types: In many circumstances, it seems that 10^4 is used as the cut off, but in other circumstances, it seems that 10^3 is used.

As requested, we have now increased the size of the FACS plots to better see the axes in the **New Supplementary Fig. 1**. As mentioned above, the gating strategies to isolate and identify the FAP+ CAF / CAF-S1 clusters use these markers successively and progressively, which has been now described in more details in the New version of the Text. Moreover, the cut-offs for the gating for each marker was defined based on the isotype controls represented in black in the FACS plots. As recommended by the Reviewer, we have now more precisely described these progressive gating strategies using these markers successively in the new version of the **Methods** section, **p44** (#Characterization of FAP+ CAF clusters).

Text added in the Methods, p44

Characterization of FAP+ CAF clusters: The markers used for the gating strategy identifying FAP+ CAF (CAF-S1) clusters have been based on differentially expressed genes and described in detail in one of our recent publication¹⁰. These different markers were defined based on pairwise comparisons of expression profiles from FAP+ CAF clusters that helped us to identify specific genes for each cluster. In addition, we considered surface expressed proteins for which commercially antibodies were available allowing both flow cytometry analysis and cell sorting. These markers cannot be used separately as the identification of the

clusters relies on the successive combination of different markers, as described below. The identification of FAP+ CAF clusters relies on the combination of these different markers tested successively. In brief, FAP+ CAF were first separated on the basis of the ANTXR1 protein level that distinguished myofibroblastic (myCAF, ANTXR1+) from inflammatory (iCAF, ANTXR1-). ANTXR1+ myCAF clusters were next distinguished according to SDC1, LAMP5, and CD9 protein levels. ANTXR1+ SDC1+ LAMP5- were defined as ECM-myCAF, ANTXR1+ LAMP5+ SDC1+/- as TGF β -myCAF and ANTXR1+ SDC1- LAMP5- CD9+ as Wound-myCAF. ANTXR1- iCAF clusters were separated using GPC3, DLK1 and CD74 markers. ANTXR1- GPC3+ DLK1+/- were defined as Detox-iCAF; ANTXR1- GPC3- DLK1+ as IL-iCAF and ANTXR1- GPC3- DLK1- CD74+ as IFN γ -iCAF.....The cut-offs for the gating for each marker was defined based on the isotype controls represented in black in the representative FACS plots in **Supplementary Fig. 1**.

8. Any antigen presenting features among the CAFs here?

We thank the Reviewer for raising this point. Indeed, we described previously that the IFN γ -iCAF cluster exhibits high expression of CD74 (Please, see below **Figure 5 for Reviewer**), encoding MHC class II invariant chain, a marker of the antigen-presenting CAF (apCAF) subset identified in pancreatic cancer (*Elyada, Cancer Discovery, 2019; Huang, Cancer Cell, 2022*), suggesting that this IFN γ -iCAF cluster might be the “apCAF” counterpart in breast cancer. As requested, we have now given this precision in the new version of the text, **p4**.

Figure 5 for Reviewer: Expression of *CD74* in FAP+ CAF clusters from breast cancer

Text added in the Introduction p4

Based on differentially expressed genes as previously described in detail ¹⁰, the three iCAF clusters are characterized by detoxification pathway (Detox-iCAF), interleukin-signaling pathway (IL-iCAF) and IFN γ -mediated response (IFN γ -iCAF). IFN γ -iCAF express the CD74 antigen and are potentially reminiscent of the antigen-presenting CAF (ap-CAF) identified in PDAC ^{6, 45}.

9. The CAF markers used by Friedman et al., Nat Cancer 2020 should be discussed.

As observed in **Fig. 1R**, we have been very interested by the data from mouse mammary tumors published in (*Friedman, Nat Cancer, 2020*), as we could identify by label transfer our main CAF-S1 clusters identified in human BC in this mouse model. In line with this findings, several markers (indicated in Fig. 1d in Friedman’s paper), as well as CAF cluster identities, revealed in the study from Dr. Scherz-Shouval’s laboratory are in complete agreement with ours. Indeed, the Cxcl12+ Mt2a+ immune regulatory E correspond to our inflammatory CAF (iCAF), in particular our IL-iCAF; the Il6+ Inflammatory A to our Detox-iCAF and the Ag-presentation cluster to our CD74+ IFN γ -iCAF. Regarding myofibroblastic CAF (myCAF), the Col5a1+ Col11a1+ ECM cluster correspond to our ECM-myCAF and the Col4a1+ cluster to our TGF β -myCAF. Interestingly, Ctgf, marker of the Wound-healing cluster in this study, is a well-identified YAP1-TEAD target gene, in agreement with the role of YAP1 we identified in the

transition from Detox-iCAF to Wound-myCAF in our current manuscript. Expression of these markers identified in (Friedman, Nat Cancer, 2020) is shown below, in the **Figure 6 for Reviewer**. Consistent with these observations, we have been able to make the correspondence between the FAP+ CAF clusters identified from human breast cancer with those identified by (Friedman, Nat Cancer, 2020) from mouse mammary tumors by using the unsupervised method of label transfer as shown **Fig. 1R** and **Supplementary Fig. 2H**, and described **p8**.

Editorial Note: Figure redacted

Figure 6 for Reviewer: UMAP showing expression of markers identified in (Friedman, Nat Cancer, 2020) (B) in scRNAseq dataset of CAF-S1 clusters from human breast cancer (shown in A).

Text added in the Results, p8

To confirm these observations *in vivo*, we leveraged previously published scRNA-seq datasets, which examined changes in TME composition following tumor implantation in mice^{9, 42} (**Fig. 1R, S**). In these mouse models, BC⁹ and PDAC⁴² cancer cells were transplanted, followed by sampling and scRNA-seq at different timepoints after grafting. We isolated fibroblasts from the two datasets and annotated the different clusters using label transfer, thereby identifying Detox-iCAF, Wound-myCAF and ECM-myCAF as the main FAP+ CAF clusters in these BC and PDAC mouse models (**Supplementary Fig. 2H**).

Text added in the Supplementary Fig. 2H, p60

(H) Violin plots showing prediction scores obtained from label transfer using the BC atlas applied on the BC mouse dataset (**Left**) and the PDAC mouse dataset (**Right**).

10. How does DPP4 expression play into the originally presented CAF subpopulations? Can this be used as a more general marker? Similarly how does Yap expression play out based on the data presented in figure 1?

As requested, we show expression of DPP4 and YAP1-target genes in FAP+ CAF clusters in the **new Fig. 2B, E**. In this BC dataset (enriched in FAP+ CAF), the cells representing the direct transition constitute 3% of the total FAP+ CAF, meaning 608 transitional Detox-iCAF among 4016 Detox-iCAF for a total of more than 18 000 FAP+ CAF isolated from human BC; and the cells involved in the indirect transition (Wound-myCAF) 6% (1107 cells). This underlines the inherent difficulty in capturing these transitional cells, which requires a highly resolute dataset. The BC atlas we built for our current study contains three times fewer FAP+ CAF compared to our enriched dataset, a necessary trade off when all cell types are sampled. Still, as requested, we have analyzed DPP4 and YAP1-TEAD-target gene density of expression in our BC atlas (shown in **Fig. 3B**). In this dataset, we observed a similar up-regulation of DPP4 in CAF transitioning between Detox-iCAF and ECM-myCAF, along with expression in a subset of ECM-myCAF fibroblasts (see below **Figure 7 for Reviewer**). Similarly, the pattern of YAP1-TEAD target genes mirrors the activation pattern from Detox-

iCAF to ECM-myCAF. Still, as expected, accurately distinguishing these two transitions in this dataset is challenging due to the limited number of cells available. This is why, and we hope the Reviewer will agree, we focus the *in silico* analysis on our enriched dataset (which remains the most resolutive one for FAP+ CAF in human BC) before the functional validations. Data are shown below on the **Figure 7 for Reviewer**.

Figure 7 for the Reviewer: Density of expression of DPP4 (A) and YAP1-TEAD-target genes (B) in the CAF populations from the human BC atlas we built in our study (BC atlas shown in Fig. 3B).

11. *The proposed role of YAP1 is interesting; however, it should be considered in the context that YAP1 likely plays a strong role in myCAFs.*

We thank the reviewer for his comment regarding the role of YAP1 in myCAF. YAP1 activity has been extensively studied in tumor cells in various cancer, in particular YAP1 up-regulation in cancer cells after crosstalk with CAF. To our knowledge, the role of YAP1 in CAF plasticity and differentiation in BC has been investigated at a much smaller extent and is much less known. While YAP1 has been identified as playing a role in myofibroblast differentiation in various fibrotic models with different etiologies⁸¹⁻⁸³, its implication in myofibroblastic differentiation in cancer has been less studied. Our study not only shows that YAP1 is essential for the differentiation of inflammatory CAFs (iCAF) into myCAF in BC but also reveals that iCAF can convert into ECM-myCAF through a DPP4-dependent YAP1-independent transition. This finding is crucial as it provides deeper insight into the functional heterogeneity and dynamics within the CAF populations in the human BC. Our findings highlight both the critical role of YAP1 in one transition but also the existence of alternative pathways, such as the DPP4-mediated transition, that operate independently of YAP1. While our study focuses on this particular aspect of YAP1 role, we acknowledge that further research is required to explore its broader implications in myCAFs and tumor biology as a whole. We have now highlighted this notion in the **Discussion p19**.

Text added in the Discussion, p19

While myofibroblast differentiation has been shown to involve YAP1 activation in various fibrotic models^{81, 82, 83}, its role in CAF plasticity is much less known in BC. Here, we found that the Detox-iCAF cluster can give rise to the activated myCAF state in the presence of cancer cells through two main paths: an indirect transition mediated by the YAP1-signaling pathway passing through the Wound-myCAF cluster and a DPP4-dependent direct transition between Detox-iCAF and ECM-myCAF clusters (See Model, Fig. 8).

12. *The language used in relation to Figure 6 comes through a little too strongly given the level of extrapolation to draw the correlations provided. While the results are of interest, the language could be toned down – eg. sentence on lines 520-521 as well as 521-524.*

We thank the Reviewer for this feedback on the language used in relation to the **New Fig. 7**. As requested, we have now revised the text to tone down the language, while still accurately transmitting the key findings of our study. We hope that these modifications address the concern raised by the Reviewer. They have been inserted **p18**.

Text added in the Results' section, p18

This analysis allowed us to identify a subgroup of DCIS patients with a potentially lower risk of invasive progression (**Fig. 7E**). Our findings thus suggest that FAP+ CAF heterogeneity might be an important factor in DCIS progression and provide preliminary insights for addressing the issue of overtreatment in DCIS, as patients with a low-risk of progression might benefit from therapeutic de-escalation.

13. Discussion of the current clinical work exploring FAP as a CAR-T target for solid tumors is warranted in this manuscript

We found the suggestion of the Reviewer very interesting, and we have now included a discussion on FAP as a potential CAR-T cell target, **p22**, as requested.

Text added in the Discussion, p22

Finally, FAP has been identified as an attractive target for CAR-T therapy because of its role in shaping the immunosuppressive TME, a major obstacle in the treatment of solid tumors. By targeting FAP+ CAF, CAR-T cells could disrupt tumor's immunosuppressive niches, ultimately enhancing immunotherapy efficacy. Still, FAP+ CAF are highly heterogenous and part of a complex network within the TME. As we show here, they also contribute to the formation of immuno-protective niches, such as the one composed of FAP+ Detox-iCAF and FOLR2+ TAM. Thus, a singular focus on eliminating FAP+ CAF may inadvertently disrupt these protective mechanisms and hinder the intended immunotherapeutic effect. In conclusion, combining multi-omic data in a spatial context enables us to provide new clues on FAP+ CAF identity, and how a given CAF cell state changes based on its neighboring cells.

Minor Comments

1. The first sentence of the abstract is too strong – This behavior of FAP+ CAFs is proposed, but there are certainly variations between cancer types and some assay dependence.

In the original sentence, we aimed to emphasize the established roles of FAP+ CAF as immunosuppressive and pro-metastatic components in the tumor microenvironment. However, we understand the Reviewer concern about the potential variations in CAF behaviors in different cancer types and the influence of assay-dependent factors. The revised sentence now underlines FAP+ CAF heterogeneity, while still highlighting their established roles in particular in breast cancer. We hope this modification better aligns with the complex nature of CAF in different contexts. To address this point of the Reviewer, we have thus revised the first sentence of the Abstract, **p2**, to reflect a more nuanced perspective.

Text added in the Abstract, p2:

Although heterogeneity of FAP+ Cancer-Associated Fibroblasts (CAF) has been described in breast cancer, their plasticity and spatial distribution remain poorly understood.

2. The selection of the name “EcoCellTypes” is unclear – would recommend something more specific to the cell types of interest

We thank the reviewer for this important feedback, we have now clarified the origin of the word EcoCellType in the Results' section, **p13**. As requested, we have also now attributed more specific names to each ECT, by considering the main cell types composing each ECT. We have indicated these names in the **New Fig. 4C**, and corresponding text, **p13-14**, making the description of these data more intelligible.

Text added in the Results' Section, p13-14

We next aimed to identify unique patterns of cell types that preferentially coexisted within specific niches. To do so, we applied hierarchical clustering on the mean abundance of cell types per niche. This allowed us to identify 10 co-localization patterns within the niches that

were shared across sections (**Fig. 4C**). We refer to these co-localization patterns as EcoCellTypes (ECT) which stands for ecosystem of cell types (#See Methods, #Niches and ECT identification) (**Fig. 4C**). ECT represented cell types and states co-existing across BC patients within the same tumor area (e.g. each area being a niche). Interestingly, some ECTs were found together in specific niches but not others, highlighting the interest of this approach to discover niche-specific colocalization pattern. Thus, each ECT contains a specific composition of cell types or states localized in close proximity, and we identified specific ECT enriched in different FAP+ CAF clusters.....

.....The “Detox-iCAF-enriched immuno-protective ECT4” highlighted the spatial co-occurrence of Detox-iCAF, ap-EC, Mo-DC, and FOLR2+ TAM, characterized by their localization in the peritumoral zone but also within the tumor bed, forming an intra-tumoral peritumor-like stroma. The “IL-iCAF-enriched normal stroma ECT5”, composed of IL-iCAF, Adipo-EC, and Contractile-CAP, encompassed cell types associated with the “Normal epithelial structures ECT6”, formed by normal epithelial and myoepithelial cells. Both ECT1 and ECT2 were enriched in immune cells. Indeed, the “IFN γ -iCAF-enriched immunosuppressive ECT1” contained immuno-suppressive cell types including FOXP3+ Treg or NKG2A+ NKreg, while the “Precursor immune cell ECT2” contained more precursor or cytotoxic cell states like SELL+ CD4+ and GZMK+ CD8+ T lymphocytes. ECT7-10 gathered cell types strongly associated with cancer cells. The “IFN $\alpha\beta$ -myCAF-enriched cancer cell ECT7” encompassed cancer cells, IFN $\alpha\beta$ -myCAF, and SPP1+ TAM. Interestingly, the “ECM- and TGF β -myCAF-enriched ECT9” also spatially overlapped with ECT7, revealing the proximity of cancer cells with ECM-myCAF, TGF β -myCAF, Angio-EC, and Ag-CAP. Finally, the “Wound-myCAF-enriched intratumoral stroma ECT10” brought together Wound-myCAF, ECM-CAP, and TREM2+ TAM.

3. Line 187 should likely read “in” rather than “on”

We thank the Reviewer for the careful reading of our manuscript. We have now corrected the Text accordingly, **p7**.

Text modified in the Results’ section, p7:

We confirmed that these fibroblasts were all positive for FAP, with a higher percentage of ANTXR1+ FAP+ myCAF clusters when expanded on plastic dishes and of ANTXR1- FAP+ iCAF clusters on collagen-coated dishes (**Fig. 1I** and **Supplementary Fig. 1E, F**), as observed in PDAC^{5, 63}.

4. Would consider an additional control for 1L – does the ECM increase with time even without cancer cells?

We thank the reviewer for this mention. As recommended, we have now analyzed the phenotype of Detox-iCAF fibroblasts at different timepoints without cancer cells. Interestingly, we confirmed that ECM-myCAF do not increase with time in absence of cancer cells, thereby indicating that cancer cells are essential for the transition from Detox-iCAF toward ECM-myCAF. These data are now added in **Supplementary Fig. 2D** and described **p8**.

Text added in the Results’ Section, p8

Importantly, maintenance of Detox-iCAF alone (**Supplementary Fig. 2D**) or co-culture with non-tumoral breast epithelial cells (MCF10A) (**Fig. 1O** and **Supplementary Fig. 2E**) did not induce Wound-myCAF or ECM-myCAF, showing that the presence of cancer cells is required to induce these clusters.

Text added in the Legend of the Supplementary Fig. 2, p60

(D) Bar blots showing the percentages of the distinct FAP+ CAF clusters after culture of primary Detox-iCAF alone, without cancer cells. Timepoints below each bar plot indicate the duration of the culture. Data are mean \pm SEM (n = 3).

Reviewer #4 (Remarks to the Author): *with expertise in breast cancer, omics*
In the manuscript entitled « Comprehensive spatial landscape and plasticity of immunosuppressive fibroblasts in breast cancer », Croizer et al combined scRNAseq and spatial transcriptomics to characterize the phenotype and function of FAP+ CAFs subsets. The authors identified 10 spatially distinct co-localization patterns of cluster of immune cells and CAFs and showed that immunosuppressive ECM-myCAF shift monocytes towards TREM2 macrophages in vitro, and cluster with SPP1+ TAM in situ, while immunoprotective Detox-iCAF shift monocytes towards FOLR2+ macrophages in vitro, and cluster with FOLR2+ TAMs in situ.

Overall, the study is very insightful and will bring new and important knowledge on TME niches within breast cancer, and beyond. The experimental plan is well-designed, the data are robust and main findings were validated in multiple datasets, with mechanistical experiments in vitro. In my opinion, the main findings of the current study is the identification of two clusters of paired subsets of fibroblast/macrophage, one immunosuppressive and one immunoprotective, especially that the authors showed mechanistically how distinct FAP+ CAF subset induce different macrophage phenotype. Further, recent papers extensively described these immunosuppressive TREM2 or SPP1 macrophage (Park & Merad, Nat Immunology 2023; Ruben & Pittet, Science 2023). Thus, I would suggest to edit the manuscript by focusing on these two pairs.

We are very grateful to the Reviewer for the positive assessment of our work and for considering that our study provides very insightful and important knowledge on TME niches in cancer. We greatly appreciated Reviewers' comments that we have now addressed and which improved our manuscript.

Major comments:

1. It is unclear why the authors investigate the effect of FAP+ susbset on outcome of DCIS since the main work was performed on invasive breast carcinoma. If the authors did not observe a prognostic effect of FAP+ subset on breast cancer outcome, I would suggest to investigate the prognostic value of ECMmyCAF/TREM2 TAM ratio and Detox-iCAF/FOLR2+ TAM in breast cancer publically available RNAseq datasets (all-comers), breast cancer subset (TNBC vs luminal and within luminal lobular vs ductal), or biopsies of TNBC pCR vs residual disease (I would expect that residual disease would be enriched in ECMmyCAF/TREM2).

We thank the reviewer for this comment. We have investigated the impact of FAP+ CAF clusters on DCIS because they are the most common precursor of invasive breast cancer (IBC). Moreover, the identification of DCIS patients with high-risk of recurrence (either in DCIS or in IBC) is pivotal for optimizing treatment decision. We do agree with the Reviewer that stromal heterogeneity and CAF composition observed in IBC can be key readouts of DCIS progression. We thus compared the FAP+ CAF cluster content between IBC molecular subtypes and evaluated their prognostic value, to address the Reviewer concern. To do so, we investigated the proportions of the different cell types and of EcoCellTypes (ECT) in the METABRIC cohort (Curtis, Nature, 2012) after deconvolution of available transcriptomic data using the BayesPrism algorithm. We performed deconvolution on 1234 patients, including 487 Lum A, 368 Lum B, 193 HER2 and 186 Basal-like TN BC.

At cellular level, Cox regression analyses revealed that higher fractions of cancer cells ($p = 0.0004$), IFN $\alpha\beta$ -myCAF ($p = 0.005$), FOXP3+ CD4+ Treg ($p = 0.0005$), SPP1+ TAM ($p = 0.003$) and TREM2+ TAM ($p = 0.04$) were associated with a decreased overall survival. Conversely, higher content in Detox-iCAF ($p = 0.007$), SELL+ CD4+ ($p = 0.016$), CD69+ CD4+ ($p = 0.03$) and FOLR2+ TAM ($p = 0.02$) was associated with a better prognosis. As suggested by the reviewer, we also examined the prognostic significance of the ECM-myCAF / TREM2 TAM ratio, and the Detox-iCAF / FOLR2+ TAM ratio, but none of these ratios showed any significant prognostic impact on overall survival.

As ECT recapitulate cell-type and cell-state communities, we also analyzed ECT prognostic values and highlighted significant differences in ECT composition among BC subtypes. Specifically, in terms of immune components, we found that Lum A BC exhibit a lower fraction of immuno-suppressive cells (ECT1) compared to LumB, HER2, and Basal-like TN BC. Conversely, they display a higher proportion of naive CD4 T cells, as well as precursor and cytotoxic CD8+ T lymphocytes (which constitute the ECT2) than the other BC subtypes. We also observed significant differences in the stromal compartment. Lum A BC show a higher content in ECT5 (composed of IL-iCAF, Adipo-EC, contractile-CAP) and in ECT9 (ECM-myCAF, TGF β -myCAF, Ag-CAP) compared to Lum B, HER2 and Basal-like TN BC. In contrast, ECT10 (Wound-myCAF, ECM-CAP and TREM2+ TAM) is enriched in Lum B and HER2 and particularly abundant in Basal-like TN BC, while ECT4 (Detox-iCAF, FOLR2+ TAM) accumulate in Lum A and Basal-like triple-negative BC compared to Lum B and HER2 BC. This analysis of the METABRIC cohort led us to look at the impact of tumor ECT composition on patient survival. Remarkably, ECT composition of BC tumors allowed us to stratify patients in 4 subgroups (C1 to C4) with different overall survival. Patients from the C2 subgroup, which is enriched in ECT1 (immuno-suppressive), ECT3 (plasma cells) and ECT10 (TREM2+ TAM, Wound-myCAF), are mainly composed of Basal-like TN (31.5%), Lum B (31.7%) and HER2 (23%) BC subtypes and show the worst survival, as expected. In contrast, patients from the C3 subgroup-enriched in ECT2 (immune precursors and effectors), ECT4 (Detox-iCAF, FOLR2+ TAM) and ECT5 (IL-iCAF, adipo-EC) are mainly of the Lum A subtype (69.6%) and show the best overall survival. The C1 and C4 subgroups are mainly composed of Luminal patients (Lum A and Lum B). While their 5-year survival is good and comparable to this one of the C3 subgroup, the prognosis of the C1 and C4 subgroups fall after 5 years. Compared to the C3 subgroup, this difference of survival can be explained -at least in part- by enrichment in Lum B BC and in ECT6-7 (Epithelial cells, IFN $\alpha\beta$ -myCAF, SPP1+ TAM) for the C4 subgroup and in ECT9 (Angio-EC, ECM-myCAF, TGF β -myCAF) for the C1 subgroup. These new data are now included in the **New Fig. 4D-F** described **p14**, its corresponding legend **p56** and the corresponding Methods, **p40-41**.

Text added in the Results section, p14

To investigate if the composition of these ECT differed between BC molecular subtypes, we performed deconvolution of transcriptomic data from the METABRIC cohort (487 Lum A, 368 Lum B, 193 HER2 and 186 Basal-like TN BC) using BayesPrism⁷⁴. In terms of immune components, we found that Lum A BC exhibited a lower fraction of immuno-suppressive cells (ECT1) compared to Lum B, HER2, and Basal-like TN BC (**Fig. 4D**). Conversely, Lum A BC displayed a higher proportion of naive CD4 T cells, as well as precursor and cytotoxic CD8+ T lymphocytes (which constituted the ECT2) compared to the other BC subtypes. We also observed significant differences in the stromal compartment: Lum A BC contained more ECT5 (composed of IL-iCAF, Adipo-EC, contractile-CAP) and ECT9 (ECM-myCAF, TGF β -myCAF, Ag-CAP) compared to Lum B, HER2 and Basal-like TN BC. In contrast, ECT10 (Wound-myCAF, ECM-CAP and TREM2+ TAM) was enriched in Lum B and HER2 and particularly abundant in Basal-like TN BC, while ECT4 (Detox-iCAF, FOLR2+ TAM) accumulated in Lum A (**Fig. 4D**). Remarkably, ECT composition in BC allowed us to stratify patients into 4 subgroups (C1 to C4) with different overall survival. (**Fig. 4E, F**). Patients in the C2 subgroup, with an enrichment in ECT1 (immuno-suppressive), ECT3 (plasma cells) and ECT10 (TREM2+ TAM, Wound-myCAF), were mainly composed of Basal-like TN (31.5%), Lum B (31.7%) and HER2 (23%) BC subtypes and showed the worst survival, as expected. In contrast, patients in the C3 subgroup -enriched in ECT2 (immune precursors, effectors), ECT4 (Detox, FOLR2+ TAM) and ECT5 (IL-iCAF, adipo-EC)- were mainly composed of the Lum A subtype (69.6%) and showed the best overall survival. The C1 and C4 subgroups were mainly composed of Luminal patients (Lum A and Lum B). While their 5-year survival was good and comparable to that of the C3 subgroup, the prognosis of the C1 and C4 subgroups fell after 5 years. Compared to the C3 subgroup, these survival differences could be explained in part by the C4 subgroup's enrichment in Lum B BC and ECT6-7 (Epithelial cells, IFN $\alpha\beta$ -myCAF, SPP1+ TAM) and the

C1 subgroup's abundance of ECT9 (Angio-EC, ECM-myCAF, TGF β -myCAF) (**Fig. 4E, F**). In conclusion, ECT were associated with the overall survival of BC patients, which is linked to ECT-enrichment in BC molecular subtypes.

Text added in the Legend of the Fig. 4D-F, p56

(D-F) Data from the METABRIC cohort (N = 1234 BC patients). **(D)** Proportions of each ECT in Lum A (N = 487), Lum B (N = 368), HER2 (N = 193) and Basal-like TN (N = 186) BC subtypes. P-values from Mann-Whitney test. **(E)** Heatmap and clustering of all BC samples (columns) from the METABRIC cohort showing 4 subgroups of patients (C1 = 263, C2 = 483, C3 = 227, C4 = 261) with different ECT enrichments. **(F) Left**, Kaplan–Meier curves showing overall survival of the 4 BC patient subgroups (C1–C4) stratified in the heatmap. P-value from Log-rank test. **Right**, Distribution of the BC molecular subtypes within the 4 subgroups (C1–C4) of BC patients.

Text added in the Methods, p40-41

Analysis of ECT composition in the METABRIC cohort: To analyze ECT enrichments in BC molecular subtypes, we used the BayesPrism algorithm⁷⁴ to deconvolute transcriptomic data from the METABRIC cohort. Normalized expression matrix, clinical information and PAM50 subtype classifications were obtained from METABRIC (https://www.cbioportal.org/study/summary?id=brca_metabric). Luminal A (N = 487), luminal B (N = 368), HER2 (N = 193) and Basal-like TN (N = 186) BC patients were conserved for the analysis. Raw count matrix of 73 426 high-quality cells from our BC atlas was used as input for prior information. Labels were derived from the annotation of 10 ECT described above. ECT6 and ECT7 were pooled to group all epithelial cell types. Mitochondrial and ribosomal protein coding genes were removed as these genes are expressed at high magnitude and not informative in distinguishing cell types. MALAT1 and genes from chrX and chrY were also removed following indication from BayesPrism's authors. To reduce batch effects and speed up computation, we performed deconvolution only on protein coding genes. Default parameters to control Gibbs sampling and optimization were used. Final estimation of cell type fraction in each bulk RNA-seq sample was recovered using the updated theta matrix and used for downstream analysis. Stratification of tumors was done by applying hierarchical clustering on the matrix of ECT fraction obtained using correlation distance and Ward.D2 method from *heatmap* R package. Differences in overall survival between the 4 subgroups of BC patients were assessed using Kaplan–Meier analysis and log-rank test statistics using the *survival* and *survminer* R packages.

Minor comments

1. *I found the abstract not easy to read. I would suggest that the authors remove the DPP4 and YAP1 mechanisms from the abstract, and rather focus on the two main niches: immunosuppressive ECM-myCAF/TREM2 macrophages and the immunoprotective Detox-iCAF/FOLR2 macrophages.*
2. *Similarly, I would suggest to edit the title and mention macrophages/fibroblast pairs*

We thank the reviewer for this feedback regarding the title and abstract of our paper. We appreciate your suggestion to focus on the immunosuppressive ECM-myCAF/TREM2 TAM and the immuno-protective Detox-iCAF/FOLR2 TAM in the abstract. It is indeed an important and interesting conclusion of our study and the inclusion of the immuno-protective Detox-iCAF/FOLR2 microenvironment was lacking in the abstract. We believe that the inclusion of the DPP4 and YAP1 mechanisms is essential to convey the full scope of our findings, especially regarding the plasticity and functional diversity of the FAP+ CAF clusters in breast cancer. Indeed, after the revision we have now strengthened our conclusions regarding CAF plasticity in breast cancer with additional assays with different cell lines, TGF β stimulation and TGFB knock down.

We have also strengthened our conclusions regarding CAF and immune cells crosstalk, therefore our study extends beyond fibroblast and macrophage interactions; it includes important insights into the roles of additional immune cells like T regulatory cells and NK cells. These elements are crucial to understand the full spatial landscape and cellular interactions in breast cancer, which is the core focus of our paper. To address the readability of the abstract, we have now revised it to ensure that the pivotal roles of DPP4 and YAP1, and the involvement of various cell types, including T reg and NK cells, are effectively communicated.

Regarding the title, we understand your suggestion to emphasize macrophage/fibroblast pairs. However, we believe that our current title, "Comprehensive Spatial Landscape and Plasticity of Immunosuppressive Fibroblasts in Breast Cancer," more accurately reflects the broad scope of our study, which encompasses a wider range of cellular interactions and mechanisms.

Finally, we now highlight the link between CAF clusters and macrophages in the schematic model in Figure 8, p58.

Text added in the Abstract, p2

In turn, ECM-myCAF polarize TREM2+ macrophages, regulatory NK and T cells to induce immunosuppressive EcoCellTypes, while Detox-iCAF are associated with FOLR2+ macrophages in an immuno-protective EcoCellType.

Sentences of the Figure 8 (schematic model)

p54: In addition, specific TAM are found in different FAP+ CAF cluster-enriched territories. While FOLR2+ TAM are close to Detox-iCAF, TREM2+ and SPP1+ TAM are enriched in ECM-myCAF, IFN $\alpha\beta$ -myCAF and TGF β -myCAF-enriched niches.

3. It would be nice to add histology images, H&E and IHC to illustrate these distinct niches from the morphological point of view

We thank the reviewer for this interesting proposition. As requested, we have now included histological images showing several niches. This enabled us to confirm that the niche composition totally matches with the cell types recognized by Pathologists (Pr. A. Vincent-Salomon and Dr. L. Djerroudi, authors of our manuscript). We have now included these results in the **Supplementary Fig 8C** and described in the Text, **p13**, with the corresponding legend of Supplementary Fig. 8C, **p60**.

Text added in the Results' Section, p13:

Using this unsupervised approach, we identified 11 spatial niches that could be visualized directly on sections (**Fig. 4B** and **Supplementary Fig. 8C**). These niches consisted of spots sharing similar cell type enrichment but exhibiting distinct spatial organization within the tissue.

Text added in the Legend of the Supplementary Fig. 8C, p60

(C) Section displayed in **Fig.3C**. Colors indicate the niches. Black boxes correspond to representative areas (Regions of Interest, ROI) for 4 selected niches shown at 2 different magnifications on the Bottom and on the Right. For each ROI: **Left:** Spots indicating the niche; Scale bar = 150 μm .: **Right:** Corresponding histological section of the niche; Scale bar = 25 μm .

REVIEWERS' COMMENTS

Reviewer #1 (Remarks to the Author):

I would like to thank the authors for the thorough response and revision based on my previous comments. I have no further concerns to raise and strongly support the publication of this manuscript.

Reviewer #2 (Remarks to the Author):

The authors have addressed all issues. Therefore, I think this manuscript can be accepted now.

Reviewer #3 (Remarks to the Author):

The authors have addressed my recommendations for revision.

Reviewer #4 (Remarks to the Author):

the authors addressed all the comments